# EMPIRICAL OR INVARIANT RISK MINIMIZATION?
# A SAMPLE COMPLEXITY PERSPECTIVE

**Kartik Ahuja**[†]**, Jun Wang**[*]**, Amit Dhurandhar**[†]**, Karthikeyan Shanmugam**[†]**, Kush R. Varshney**[†]
[†] IBM Research, TJ Watson Research Center, NY, [*] Rensselaer Polytechnique Institute

## ABSTRACT

Recently, invariant risk minimization (IRM) was proposed as a promising solution to address out-of-distribution (OOD) generalization. However, it is unclear when IRM should be preferred over the widely-employed empirical risk minimization (ERM) framework. In this work, we analyze both these frameworks from the perspective of sample complexity, thus taking a firm step towards answering this important question. We find that depending on the type of data generation mechanism, the two approaches might have very different finite sample and asymptotic behavior. For example, in the covariate shift setting we see that the two approaches not only arrive at the same asymptotic solution, but also have similar finite sample behavior with no clear winner. For other distribution shifts such as those involving confounders or anti-causal variables, however, the two approaches arrive at different asymptotic solutions where IRM is guaranteed to be *close* to the desired OOD solutions in the finite sample regime for polynomial generative models, while ERM is biased even asymptotically. We further investigate how different factors — the number of environments, complexity of the model, and IRM penalty weight — impact the sample complexity of IRM in relation to its distance from the OOD solutions.

## 1 INTRODUCTION

A recent study shows that models trained to detect COVID-19 from chest radiographs rely on spurious factors such as the source of the data rather than the lung pathology (DeGrave et al., 2020). This is just one of many alarming examples of spurious correlations failing to hold outside a specific training distribution. In one commonly cited example, Beery et al. (2018) trained a convolutional neural network (CNN) to classify camels from cows. In the training data, most pictures of the cows had green pastures, while most pictures of camels were in the desert. The CNN picked up the spurious correlation and associated green pastures with cows thus failing to classify cows on beaches.

Recently, Arjovsky et al. (2019) proposed a framework called invariant risk minimization (IRM) to address the problem of models inheriting spurious correlations. They showed that when data is gathered from multiple environments, one can learn to exploit invariant causal relationships, rather than relying on varying spurious relationships, thus learning robust predictors. More recent work suggests that empirical risk minimization (ERM) is still state-of-the-art on many problems requiring OOD generalization (Gulrajani & Lopez-Paz, 2020). This gives rise to a fundamental question: when is IRM better than ERM (and vice versa)? In this work, we seek to answer this question through a systematic comparison of the sample complexity of the two approaches under different types of train and test distributional mismatches.

The distribution shifts $\left(\mathbb{P}^{\text{train}}(X, Y) \neq \mathbb{P}^{\text{test}}(X, Y)\right)$ that we consider informally stated satisfy an *invariance condition* – there exists a representation $\Phi^*$ of the covariates such that $\mathbb{P}^{\text{train}}(Y|\Phi^*(X)) = \mathbb{P}^{\text{test}}(Y|\Phi^*(X)) = \mathbb{P}(Y|\Phi^*(X))$. A special case of this occurs when $\Phi^*$ is identity – $\mathbb{P}^{\text{train}}(X) \neq \mathbb{P}^{\text{test}}(X)$ but $\mathbb{P}^{\text{train}}(Y|X) = \mathbb{P}^{\text{test}}(Y|X)$ – such a shift is known as a *covariate-shift* (Gretton et al., 2009). In many other settings $\Phi^*$ may not be identity (denoted as I), examples include settings with confounders or anti-causal variables (Pearl, 2009) where covariates appear spuriously correlated with the label and $\mathbb{P}^{\text{train}}(Y|X) \neq \mathbb{P}^{\text{test}}(Y|X)$. We use causal Bayesian networks to illustrate these shifts in Figure 1. Suppose $X^e = [X_1^e, X_2^e]$ represents the image, where $X_1^e$ is the shape of the animal and $X_2^e$ is the background color, $Y^e$ is the label of the animal, and $e$ is the index of the environment/domain. In Figure 1a) $X_2^e$ is independent of $(Y^e, X_1^e)$, it represents the covariate shift

case ($\Phi^* = I$). In Figure 1b) $X_2^e$ is spuriously correlated with $Y^e$ through the confounder $\varepsilon^e$. In Figure 1c) $X_2^e$ is spuriously correlated with $Y^e$ as it is anti-causally related to $Y^e$. In both Figure 1b) and c) $\Phi^* \neq I$; $\Phi^*$ is a block diagonal matrix that selects $X_1^e$.

Our setup assumes we are given data from multiple training environments satisfying the invariance condition, i.e., $\mathbb{P}(Y|\Phi^*(X))$ is the same across all of them. Ideally, we want to learn and predict using $\mathbb{E}[Y|\Phi^*(X)]$; this predictor has a desirable OOD behavior as we show later where we prove min-max optimality with respect to (w.r.t.) unseen test distributions satisfying the invariance condition. Our goal is to analyze and compare ERM and IRM's ability to learn $\mathbb{E}[Y|\Phi^*(X)]$ from finite training data acquired from a fixed number of training environments. Our analysis has two parts.

**1) Covariate shift case ($\Phi^* = I$):** ERM and IRM achieve the same asymptotic solution $\mathbb{E}[Y|X]$. We prove (Proposition 4) that the sample complexity for both the methods is similar thus there is no clear winner between the two in the finite sample regime. For the setup in Figure 1a), both ERM and IRM learn a model that only uses $X_1^e$.

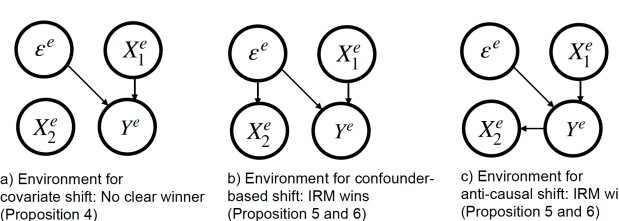

a) Environment for covariate shift: No clear winner (Proposition 4)

b) Environment for confounder-based shift: IRM wins (Proposition 5 and 6)

c) Environment for anti-causal shift: IRM wins (Proposition 5 and 6)

Figure 1: Causal Bayesian networks for different distribution shifts.

**2) Confounder/Anti-causal variable case ($\Phi^* \neq I$):** We consider a family of structural equation models (linear and polynomial) that may contain confounders and/or anti-causal variables. For the class of models we consider, the asymptotic solution of ERM is biased and not equal to the desired $\mathbb{E}[Y|\Phi^*(X)]$. We prove that IRM can learn a solution that is within $\mathcal{O}(\sqrt{\epsilon})$ distance from $\mathbb{E}[Y|\Phi^*(X)]$ with a sample complexity that increases as $\mathcal{O}(\frac{1}{\epsilon^2})$ and increases polynomially in the complexity of the model class (Proposition 5, 6); $\epsilon$ (defined later) is the slack in IRM constraints. For the setup in Figure 1b) and c), IRM gets close to only using $X_1^e$, while ERM even with infinite data (Proposition 17 in the supplement) continues to use $X_2^e$. We summarize the results in Table 1.

Arjovsky et al. (2019) proposed the colored MNIST (CMNIST) dataset; comparisons on it showed how ERM-based models exploit spurious factors (background color). The CMNIST dataset relied on anti-causal variables. Many supervised learning datasets may not contain anti-causal variables (e.g. human labeled images). Therefore, we propose and analyze three new variants of CMNIST in addition to the original one that map to different real-world settings: i) covariate shift based CMNIST (CS-CMNIST): relies on selection bias to induce spurious correlations, ii) confounded CMNIST (CF-CMNIST): relies on confounders to induce spurious correlations, iii) anti-causal CMNIST (AC-CMNIST): this is the original CMNIST proposed by Arjovsky et al. (2019), and iv) anti-causal and confounded (hybrid) CMNIST (HB-CMNIST): relies on confounders and anti-causal variables to induce spurious correlations. On the latter three datasets, which belong to the $\Phi^* \neq I$ class described above, IRM has a much better OOD behavior than ERM, which performs poorly regardless of the data size. However, IRM and ERM have a similar performance on CS-CMNIST with no clear winner. These results are consistent with our theory and are also validated in regression experiments.

## 2 RELATED WORKS

**IRM based works.** Following the original work IRM from Arjovsky et al. (2019), there have been several interesting works — (Teney et al., 2020; Krueger et al., 2020; Ahuja et al., 2020; Chang et al., 2020; Mahajan et al., 2020) is an incomplete representative list — that build new methods inpired from IRM to address the OOD generalization problem. Arjovsky et al. (2019) prove OOD guarantees for linear models with access to infinite data from finite environments. We generalize these results in several ways. We provide a first finite sample analysis of IRM. We characterize the impact of hypothesis class complexity, number of environments, weight of IRM penalty on the sample complexity and its distance from the OOD solution for linear and polynomial models.

**Theory of domain generalization and domain adaption.** Following the seminal works (Ben-David et al., 2007; 2010), there have been many interesting works — (Muandet et al., 2013; Ajakan et al., 2014; Zhao et al., 2019; Albuquerque et al., 2019; Li et al., 2017; Piratla et al., 2020; Matsuura & Harada, 2020; Deng et al., 2020; David et al., 2010; Pagnoni et al., 2018) is an incomplete representative list (see Redko et al. (2019) for further references) — that build the theory of domain adaptation and generalization and construct new methods based on it. While many of these works develop bounds on loss over the target domain using train data and unlabeled target data,

Table 1: Summary of (empirical) IRM vs. ERM for finite hypothesis class $\mathcal{H}_\Phi$. $\epsilon$: slack in IRM constraints, $\nu$: approximation w.r.t optimal risk, $\delta$: failure probability, $\mathcal{E}_{tr}$: set of training environments, $n$: data dimension, $p$: degree of the generative polynomial, $L, L'$: bound on loss & its gradients.

| Assumptions | Method | Sample complexity | OOD |
|---|---|---|---|
| *Covariate shift case:* $\mathbb{E}^e[Y^e\|X^e]$ is invariant (Proposition 4) | ERM | $\frac{8L^2}{\nu^2}\log\left(\frac{2\|\mathcal{H}_\Phi\|}{\delta}\right)$ | Yes |
| | IRM | $\max\left\{\frac{8L^2}{\nu^2}\log\left(\frac{4\|\mathcal{H}_\Phi\|}{\delta}\right), \frac{16L'^4}{\epsilon^2}\log\left(\frac{2}{\delta}\right)\right\}$ | Yes |
| *Confounder/Anti-causal variable case:* $\mathbb{E}^e[Y^e\|\Phi^*(X^e)]$ is invariant, Linear, Polynomial models, $\|\mathcal{E}_{tr}\| = \mathcal{O}(n^p)$ (Proposition 5, 6,17) | ERM | $\frac{8L^2}{\nu^2}\log\left(\frac{2\|\mathcal{H}_\Phi\|}{\delta}\right)$ | No |
| | IRM | $\frac{16L'^4}{\epsilon^2}\log\left(\frac{2\|\mathcal{H}_\Phi\|}{\delta}\right)$ | Yes |

some (Ben-David & Urner, 2012; David et al., 2010; Pagnoni et al., 2018) analyze the finite sample (PAC) guarantees for domain adaptation under covariate shifts. These works (Ben-David & Urner, 2012; David et al., 2010; Pagnoni et al., 2018) access unlabeled data from a target domain, which we do not. Instead, we have data from multiple training domains (as in domain generalization). In these works, the guarantees are w.r.t. a specific target domain, while we provide (for linear and polynomial models) worst-case guarantees w.r.t. all the unseen domains satisfying the invariance condition. Also, we consider a larger family of distribution shifts including covariate shifts. The above two categories are not exhaustive – e.g., there are some recent works that characterize how some inductive biases favor extrapolation Xu et al. (2021) and can be better for OOD generalization.

## 3 SAMPLE COMPLEXITY OF INVARIANT RISK MINIMIZATION

### 3.1 INVARIANT RISK MINIMIZATION

We start with some background on IRM (Arjovsky et al., 2019). Consider a dataset $D = \{D^e\}_{e \in \mathcal{E}_{tr}}$, which is a collection of datasets $D^e = \{(x_i^e, y_i^e, e)\}_{i=1}^{n_e}$ obtained from a set of training environments $\mathcal{E}_{tr}$, where $e$ is the index of the environment, $i$ is the index of the data point in the environment, $n_e$ is the number of points from environment, $x_i^e \in \mathcal{X} \subseteq \mathbb{R}^n$ is the feature value and $y_i^e \in \mathcal{Y} \subseteq \mathbb{R}$ is the corresponding label. Define a probability distribution $\{\pi^e\}_{e \in \mathcal{E}_{tr}}$, $\pi^e$ is the probability that a training data point is from environment $e$. Define a probability distribution of points conditional on environment $e$ as $\mathbb{P}^e$, $(X^e, Y^e) \sim \mathbb{P}^e$. Define the joint distribution $\bar{\mathbb{P}}$, $(X^e, Y^e, e) \sim \bar{\mathbb{P}}, d\bar{\mathbb{P}}(X^e, Y^e, e) = \pi^e d\mathbb{P}^e(X^e, Y^e)$. $D$ is a collection of i.i.d. samples from $\bar{\mathbb{P}}$. Define a predictor $f : \mathcal{X} \to \mathbb{R}$ and the space $\mathcal{F}$ of all the possible maps from $\mathcal{X} \to \mathbb{R}$. Define the risk achieved by $f$ in environment $e$ as $R^e(f) = \mathbb{E}^e\left[\ell\left(f(X^e), Y^e\right)\right]$, where $\ell$ is the loss, $f(X^e)$ is the predicted value, $Y^e$ is the corresponding label and $\mathbb{E}^e$ is the expectation conditional on environment $e$. The overall expected risk across the training environments is $R(f) = \sum_{e \in \mathcal{E}_{tr}} \pi^e R^e(f)$. We are interested in two settings: regression (square loss) and binary-classification (cross-entropy loss). In the main body, our focus is regression (square loss) and we mention wherever the results extend to binary-classification (cross-entropy). We discuss these extensions in the supplement.

**OOD generalization problem.** We want to construct a predictor $f$ that performs well across many unseen environments $\mathcal{E}_{all}$, where $\mathcal{E}_{all} \supseteq \mathcal{E}_{tr}$. For $o \in \mathcal{E}_{all} \backslash \mathcal{E}_{tr}$, the distribution $\mathbb{P}^o$ can be very different from the train environments. Next we state the OOD problem.

$$\min_{f \in \mathcal{F}} \max_{e \in \mathcal{E}_{all}} R^e(f) \tag{1}$$

The above problem is very challenging to solve since we only have access to data from training environments $\mathcal{E}_{tr}$ but are required to find the robust solution over all environments $\mathcal{E}_{all}$. Next, we make assumptions on $\mathcal{E}_{all}$ and characterize the optimal solution to equation 1.

**Assumption 1.** *Invariance condition. There exists a representation $\Phi^*$ that transforms $X^e$ to $Z^e = \Phi^*(X^e)$ and $\forall e, o \in \mathcal{E}_{all}, \forall z \in \Phi^*(\mathcal{X})$ satisfies $\mathbb{E}^e[Y^e|Z^e = z] = \mathbb{E}^o[Y^o|Z^o = z]$. Also, $\forall e \in \mathcal{E}_{all}, \forall z \in \Phi^*(\mathcal{X})$, $\mathsf{Var}^e[Y^e|Z^e = z] = \xi^2$, where $\mathsf{Var}^e$ is the conditional variance.*

The above assumption is inspired from causality (Pearl, 2009). $\Phi^*$ acts as the causal feature extractor and from the definition of causal features, it follows that $\mathbb{E}^e[Y^e|Z^e = z]$ does not vary across

environments. When a human labels a cow she uses $\Phi^*$ to extract causal features from the pixels to identify cow while ignoring the background. The first part of the above assumption encompasses a large class of distribution shifts including standard covariate shifts (Gretton et al., 2009). Covariate shift assumes $\forall e, o \in \mathcal{E}_{all}, \forall x \in \mathcal{X}$, $\mathbb{P}(Y^e | X^e = x)$ and $\mathbb{P}(Y^o | X^o = x)$ are equal thus implying $\mathbb{E}^e[Y^e | X^e = x] = \mathbb{E}^o[Y^o | X^o = x]$. Therefore, for covariate shifts, $\Phi^*$ is identity in Assumption 1. A simple instance illustrating Assumption 1 with $\Phi^* = \mathsf{I}$ is when $Y^e \leftarrow g(X^e) + \varepsilon^e$, where $\mathbb{E}^e[\varepsilon^e] = 0, \mathbb{E}^e[(\varepsilon^e)^2] = \sigma^2, \varepsilon^e \perp X^e$. Using Assumption 1, we define the invariant map $m : \Phi^*(\mathcal{X}) \to \mathbb{R}$ as follows

$$\forall z \in \Phi^*(\mathcal{X}), \ m(z) = \mathbb{E}^e[Y^e | Z^e = z], \text{ where } Z^e = \Phi^*(X^e) \tag{2}$$

**Assumption 2.** *Existence of an environment where the invariant representation is sufficient.* $\exists$ *an environment* $e \in \mathcal{E}_{all}$ *such that* $Y^e \perp X^e | Z^e$

Assumption 2 states there exists an environment where the information that $X^e$ has about $Y^e$ is also contained in $Z^e$. Define a composition $m \circ \Phi^*, \forall x \in \mathcal{X}, m \circ \Phi^*(x) = \mathbb{E}^e[Y^e | Z^e = \Phi^*(x)]$.

**Proposition 1.** *If $\ell$ is the square loss, and Assumptions 1 and 2 hold, then $m \circ \Phi^*$ solves the OOD problem (equation 1).*

The proofs of all the propositions are in the supplement. A similar result holds for the cross-entropy loss (discussion in supplement). For the rest of the paper, we focus on learning $m \circ \Phi^*$ as it solves the OOD problem. For covariate shifts $\Phi^* = \mathsf{I}$, $m(x) = \mathbb{E}^e[Y^e | X^e = x]$ is the OOD solution. In Arjovsky et al. (2019), a proof connecting $m \circ \Phi^*$ and OOD was not stated. Recently, in Koyama & Yamaguchi (2020), a result similar to Proposition 1 was shown but with a few differences. The authors assume conditional probabilities are invariant unlike our assumption that only requires conditional expectations and variances to be invariant. However, their result applies to more losses. $m \circ \Phi^*$ is the target we want to learn. Arjovsky et al. (2019) proposed IRM since standard min-max optimization over the training environments $\mathcal{E}_{tr}$ and ERM fail to learn $m \circ \Phi^*$ in many cases. The authors in Arjovsky et al. (2019) identify a crucial property of $m \circ \Phi^*$ and use it to define an object called invariant predictor that we define next.

**Invariant predictor and IRM optimization.** Define a representation map $\Phi : \mathcal{X} \to \mathcal{Z}$ from feature space to representation space $\mathcal{Z} \subseteq \mathbb{R}^q$. Define a classifier map, $w : \mathcal{Z} \to \mathbb{R}$ from representation space to real values. Define $\mathcal{H}_\Phi$ and $\mathcal{H}_w$ as the spaces of representations and classifiers respectively. A data representation $\Phi$ elicits an invariant predictor $w \circ \Phi$ across environments $e \in \mathcal{E}_{tr}$ if there is a classifier $w$ that achieves the minimum risk simultaneously for all the environments, i.e., $\forall e \in \mathcal{E}_{tr}, w \in \arg\min_{\bar{w} \in \mathcal{H}_w} R^e(\bar{w} \circ \Phi)$. Observe that if we we transform the data with representation $\Phi^*$ then $m$ will achieve the minimum risk simultaneously in all the environments. If $\Phi^* \in \mathcal{H}_\Phi$ and $m \in \mathcal{H}_w$, then $m \circ \Phi^*$ is an invariant predictor. IRM selects the invariant predictor with least sum risk across environments (results presented later can be adapted if invariant predictor was selected based on the worst-case risk over the environments as well) as follows:

$$\min_{\Phi \in \mathcal{H}_\Phi, w \in \mathcal{H}_w} R(w \circ \Phi) = \sum_{e \in \mathcal{E}_{tr}} \pi^e R^e(w \circ \Phi)$$
$$\text{s.t. } w \in \arg\min_{\bar{w} \in \mathcal{H}_w} R^e(\bar{w} \circ \Phi), \ \forall e \in \mathcal{E}_{tr} \tag{3}$$

From the above discussion we know $m \circ \Phi^*$ is a feasible solution to equation 3. It is also the ideal solution we want IRM to find since it solves equation 1. Later in Propositions 4, 5, and 6, we show that IRM actually solves equation equation 1. For the setups in Proposition 5, and 6, conventional ERM based approaches fail thus justifying the need for above formulation.

### 3.2 Sample Complexity of Gradient Constraint Formulation of IRM

In Arjovsky et al. (2019), a gradient constrained alternate (derived below in equation 4) to equation 3 was proposed, which focuses on linear and scalar classifiers ($\mathcal{Z} = \mathbb{R}$, $\Phi : \mathcal{X} \to \mathbb{R}$, $\mathcal{H}_w = \mathbb{R}$). In this case, the composite predictor $w \circ \Phi$ is a multiplication of $w$ and $\Phi$ written as $w \cdot \Phi$. (For binary-classification predictor's output $w \cdot \Phi(x)$ represents logits.) From the definition of invariant predictors and $\mathcal{H}_w = \mathbb{R}$ it follows that if $\forall \bar{w} \in \mathbb{R}$, $R^e(1 \cdot \Phi) \leq R^e(\bar{w} \cdot \Phi)$, then $\Phi$ is an invariant predictor. For square and cross-entropy losses, $R^e(w \cdot \Phi)$ is convex in $w$. Therefore, a gradient constraint $\|\nabla_{w|w=1.0} R^e(w \cdot \Phi)\| = 0$ is equivalent to the condition that $\forall \bar{w} \in \mathbb{R}$, $R^e(1 \cdot \Phi) \leq R^e(\bar{w} \cdot \Phi)$, which implies $\Phi$ is an invariant predictor. Recall that IRM aims is to search among invariant predictors and find one that minimizes the risk. We state this as a gradient constrained optimization as follows

$$\min_{\Phi \in \mathcal{H}_\Phi} R(\Phi)$$
$$\text{s.t. } \left\| \nabla_{w|w=1.0} R^e(w \cdot \Phi) \right\| = 0, \ \forall e \in \mathcal{E}_{tr} \tag{4}$$

We propose an $\epsilon$ approximation of the above with $\epsilon$ slack in the constraint. Define $R^{'}(\Phi) = \sum_{e \in \mathcal{E}_{tr}} \pi^e \| \nabla_{w|w=1.0} R^e(w \cdot \Phi) \|^2$ and a set $\mathcal{S}^{\mathsf{IV}}(\epsilon) = \{ \Phi \mid R^{'}(\Phi) \leq \epsilon, \Phi \in \mathcal{H}_\Phi \}$. Note that $R^{'}$ is very similar to the penalty defined in Arjovsky et al. (2019). The $\epsilon$ approximation of equation 4 is

$$\min_{\Phi \in \mathcal{S}^{\mathsf{IV}}(\epsilon)} R(\Phi) \tag{5}$$

If $\epsilon = 0$, then equation 4 and equation 5 are equivalent. In all the optimizations so far, the expectations are computed w.r.t. the distributions $\mathbb{P}^e$, which are unknown. Therefore, we develop an empirical version of equation 5 below (in equation 6) and call it empirical IRM (EIRM). We replace $R$ and $R^{'}$ with empirical estimators $\hat{R}$ and $\hat{R}^{'}$ respectively. For $R$ we use a simple plugin estimator (sample mean of loss across all the samples in $D$). For $R^{'}$ we construct a new estimator that enables the use of standard concentration inequalities. Define a set $\hat{\mathcal{S}}^{\mathsf{IV}}(\epsilon) = \{ \Phi \mid \hat{R}^{'}(\Phi) \leq \epsilon, \Phi \in \mathcal{H}_\Phi \}$.

$$\min_{\Phi \in \hat{\mathcal{S}}^{\mathsf{IV}}(\epsilon)} \hat{R}(\Phi) \tag{6}$$

If we replaced $\hat{\mathcal{S}}^{\mathsf{IV}}(\epsilon)$ with $\mathcal{H}_\Phi$ in equation 6, then we get the standard ERM. ERM aims to solve $\min_{\Phi \in \mathcal{H}_\Phi} \hat{R}(\Phi)$. The sample complexity analysis of ERM aims to understand the distance between the empirical solutions and the expected solutions as a function of the number of samples. Similarly, we seek to understand the relationship between solutions of equation 6 and equation 5.

**Assumption 3.** ***Bounded loss and bounded gradient of the loss.*** $\exists \ L < \infty$, $L' < \infty$ *such that* $\forall \Phi \in \mathcal{H}_\Phi, \forall x \in \mathcal{X}, \forall y \in \mathcal{Y}, |\ell(\Phi(x), y)| \leq L, |\frac{\partial \ell(w \cdot \Phi(x), y)}{\partial w}|_{w=1.0}| \leq L^{'}$.

If every $\Phi$ in the hypothesis class $\mathcal{H}_\Phi$ is bounded by $M$ and the label space $\mathcal{Y}$ is bounded, then for both square and cross-entropy loss, $\ell(\Phi(\cdot), \cdot)$ and $\frac{\partial \ell(w \cdot \Phi(\cdot), \cdot)}{\partial w}|_{w=1.0}$ are bounded. Define $\kappa = \min_{\Phi \in \mathcal{H}_\Phi} |R^{'}(\Phi) - \epsilon|$; $\kappa$ measures how close any penalty can get to the boundary $\epsilon$. $\kappa$ quantifies how good the finite sample approximation $\hat{R}^{'}$ need to be in order to get $\hat{\mathcal{S}}^{\mathsf{IV}}(\epsilon) = \mathcal{S}^{\mathsf{IV}}(\epsilon)$. Define $\nu$ to quantify the approximation w.r.t. optimal risk.

**Proposition 2.** *For every $\nu > 0, \epsilon > 0$ and $\delta \in (0, 1)$, if $\mathcal{H}_\Phi$ is a finite hypothesis class, Assumption 3 holds, $\kappa > 0$, and if the number of samples $|D|$ is greater than $\max \left\{ \frac{16 L^{'4}}{\kappa^2}, \frac{8 L^2}{\nu^2} \right\} \log \left( \frac{4|\mathcal{H}_\Phi|}{\delta} \right)$, then with a probability at least $1 - \delta$, every solution $\hat{\Phi}$ to EIRM (equation 6) is a $\nu$ approximation of IRM, i.e. $\hat{\Phi} \in \mathcal{S}^{\mathsf{IV}}(\epsilon)$, $R(\Phi^*) \leq R(\hat{\Phi}) \leq R(\Phi^*) + \nu$, where $\Phi^*$ is a solution to IRM (equation 5).*

**Proof Sketch.** The standard analysis in learning theory on ERM or regularized/constrained ERM typically relies on linearly separable loss functions. In such cases, we can use standard plug-in estimators and analyze their behavior using concentration inequalities. In our setting, $R^{'}$ is a weighted sum of squares of expectation and thus it is not linearly separable. We develop a new way of expressing $R^{'}$ that allows us to make it linearly separable. Next, in order to ensure $R(\Phi^*) \leq R(\hat{\Phi}) \leq R(\Phi^*) + \nu$ we first need to guarantee that the set of invariant predictors is exactly recovered, i.e., $\hat{\mathcal{S}}^{\mathsf{IV}}(\epsilon) = \mathcal{S}^{\mathsf{IV}}(\epsilon)$ (exact recovery is typically not required in existing constrained analysis such as Woodworth et al. (2017) Agarwal et al. (2018)). We show that if the number of samples grow as $\frac{1}{\kappa^2}$ even the closest points on either side of the boundary of the set $\mathcal{S}^{\mathsf{IV}}(\epsilon)$ are correctly discriminated, which guarantees exact recovery of $\mathcal{S}^{\mathsf{IV}}(\epsilon)$. Once the exact set is recovered, beyond this we use standard learning theory tools to ensure $R(\Phi^*) \leq R(\hat{\Phi}) \leq R(\Phi^*) + \nu$.

The above result holds for both square and cross-entropy loss. For ease of exposition, we use the standard setting of finite hypothesis class and extend all the results to infinite hypothesis classes in the supplement (summary of insights from the extension are in Section 3.3.2). Next, we state a standard result on ERM's sample complexity. Define a $\Phi^+$ such that $\Phi^+ \in \arg \min_{\Phi \in \mathcal{H}_\Phi} R(\Phi)$

**Proposition 3.** *(Shalev-Shwartz & Ben-David, 2014) For every $\nu > 0$ and $\delta \in (0, 1)$, if $\mathcal{H}_\Phi$ is a finite hypothesis class, Assumption 3 holds, and if the number of samples $|D|$ is greater than $\frac{8 L^2}{\nu^2} \log \left( \frac{2|\mathcal{H}_\Phi|}{\delta} \right)$, then with a probability at least $1 - \delta$, every solution $\Phi^\dagger$ to ERM is an $\nu$ approximation of expected risk minimization, i.e., $R(\Phi^+) \leq R(\Phi^\dagger) \leq R(\Phi^+) + \nu$.*

**Proposition 2 vs. 3** Since $\kappa \leq \epsilon$, the sample complexity of EIRM grows at least as $\mathcal{O}(\max\{\frac{1}{\epsilon^2}, \frac{1}{\nu^2}\})$. Let us look at the two terms inside $\max$- i) $\frac{1}{\nu^2}$ growth term is similar to ERM, it ensures $\nu$ approximate optimality in the overall risk $R$, ii) $\frac{1}{\epsilon^2}$ growth ensures the IRM penalty $R'$ is less than $\epsilon$. A direct comparison of sample complexities in Propositions 2 and 3 suggests that the sample complexity of EIRM is higher than ERM, which is not the complete picture. The two approaches may not converge to the same solutions and IRM may converge to a solution with better OOD behavior than one achieved by ERM. Therefore, a fair comparison is only possible when we also study the OOD properties of the solutions achieved by the two approaches, which is the subject of the next section.

### 3.3 OOD PERFORMANCE: ERM VS. IRM

We divide the comparisons based on distributional shift assumptions that decide whether ERM and IRM arrive at the same asymptotic solutions or not.

#### 3.3.1 COVARIATE SHIFT

**Assumption 4. *Invariance w.r.t. all the features.*** $\forall e, o \in \mathcal{E}_{all}$ and $\forall x \in \mathcal{X}$, $\mathbb{E}[Y^e|X^e = x] = \mathbb{E}[Y^o|X^o = x]$. $\forall e \in \mathcal{E}_{all}$, $X^e \sim \mathbb{P}_{X^e}^e$ and the support of $\mathbb{P}_{X^e}^e$ is equal to $\mathcal{X}$.

As stated earlier, the first part of the above assumption follows from standard covariate shift assumptions (Gretton et al., 2009) and is a special case of the first part of the Assumption 1 with $\Phi^*$ set to I. If $\Phi^* = I$, then $m$ (equation 2), which simplifies to $m(x) = \mathbb{E}[Y^e|X^e = x]$, solves the OOD problem equation 1. A generative model that satisfies the above Assumption 4 is given as

$$Y^e \leftarrow g(X^e) + \varepsilon^e, \ \mathbb{E}[\varepsilon^e] = 0, \ \varepsilon^e \perp X^e, \ \mathbb{E}[(\varepsilon^e)^2] = \sigma^2 \tag{7}$$

In the above model $X^e$ is the cause, $Y^e$ is the effect, and $g$ a general non-linear function (it satisfies Assumption 4 with $m = g$). Next, we compare ERM and IRM's ability to learn $m$ under covariate shifts. In Figure 1 a), we show Define $\tilde{\kappa} = \min_{\Phi_1, \Phi_2 \in \mathcal{H}_\Phi, \Phi_1 \neq \Phi_2} |R(\Phi_1) - R(\Phi_2)|$, which measures the minimum separation between the risks of any two distinct hypothesis in $\mathcal{H}_\Phi$.

**Proposition 4.** *Let $\ell$ be the square loss. For every $\nu > 0, \epsilon > 0$ and $\delta \in (0, 1)$, if $\mathcal{H}_\Phi$ is a finite hypothesis class, $m \in \mathcal{H}_\Phi$, Assumptions 3, 4 hold, and*

- *if the number of samples $|D|$ is greater than $\max\left\{\frac{8L^2}{\nu^2}\log(\frac{4|\mathcal{H}_\Phi|}{\delta}), \frac{16L'^4}{\epsilon^2}\log(\frac{2}{\delta})\right\}$, then with a probability at least $1 - \delta$, every solution $\hat{\Phi}$ to EIRM (equation 6) satisfies $R(m) \leq R(\hat{\Phi}) \leq R(m) + \nu$. If also $\nu < \tilde{\kappa}$, then $\hat{\Phi} = m$.*

- *if the number of samples $|D|$ is greater than $\frac{8L^2}{\nu^2}\log(\frac{2|\mathcal{H}_\Phi|}{\delta})$, then with a probability at least $1 - \delta$, every solution $\Phi^\dagger$ to ERM satisfies $R(m) \leq R(\Phi^\dagger) \leq R(m) + \nu$. If also $\nu < \tilde{\kappa}$, then $\Phi^\dagger = m$.*

**Implications of Proposition 4.** ERM and EIRM both asymptotically achieve the ideal OOD solution; the above proposition helps compare them in a finite sample regime. The second term inside the $\max$ for EIRM, $\frac{16L'^4}{\epsilon^2}\log(\frac{2}{\delta})$, does not depend on the size of the hypothesis class. Hence, for large hypothesis classes, the sample complexity of EIRM equals $\frac{8L^2}{\nu^2}\log(\frac{4|\mathcal{H}_\Phi|}{\delta})$. Consequently, the sample complexity of EIRM and ERM differs by a constant $\frac{8L^2}{\nu^2}\log(2)$. Thus we conclude, for large hypothesis classes, both ERM and EIRM have similar sample complexity. Next, we contrast the sample complexity of EIRM in Proposition 4, $\mathcal{O}(\frac{1}{\nu^2})$, to Proposition 2, $\mathcal{O}(\max\{\frac{1}{\epsilon^2}, \frac{1}{\nu^2}\})$; the additional covariate shift assumption in Proposition 4 helps get to a lower sample complexity of $\mathcal{O}(\frac{1}{\nu^2})$. In Proposition 4, we assumed square loss, but a similar result extends to cross-entropy loss as well.

#### 3.3.2 DISTRIBUTIONAL SHIFT WITH CONFOUNDERS AND (OR) ANTI-CAUSAL VARIABLES

In this section, we consider more general models than equation 7, which only contained cause $X^e$ and effect $Y^e$. We also allow confounders and anti-causal variables. However, we restrict $g$ to polynomials. We start with linear models from Arjovsky et al. (2019).

**Assumption 5.**

$$e \sim \mathsf{Categorical}(\{\pi^o\}_{o \in \mathcal{E}_{tr}}), \ \forall o \in \mathcal{E}_{tr}, \pi^o > 0$$
$$Y^e \leftarrow \gamma^\mathsf{T}(Z_1^e) + \varepsilon^e, \ \varepsilon^e \perp Z_1^e, \ \mathbb{E}[\varepsilon^e] = 0, \ \mathbb{E}[(\varepsilon^e)^2] = \sigma^2, \ |\varepsilon^e| \leq \varepsilon^{\mathsf{sup}} \tag{8}$$
$$X^e \leftarrow S(Z_1^e, Z_2^e)$$

*We assume that $Z_1$ component of $S$ is invertible, i.e. $\exists\, \tilde{S}$ such that $\tilde{S}(S(Z_1, Z_2)) = Z_1$, and $\gamma \neq 0$. $\forall e \in \mathcal{E}_{tr}, \pi^e \geq \frac{\pi^{\min}}{|\mathcal{E}_{tr}|} > 0$. Define $\Sigma^e = \mathbb{E}[X^e X^{e,\mathsf{T}}]$. $\forall e \in \mathcal{E}_{tr}, \Sigma^e$ is positive definite. The support of distribution of $Z^e = (Z_1^e, Z_2^e)$, $\mathbb{P}_{Z^e}^e$, is bounded and the norm of $S$, $\|S\| = \sigma_{\max}(S)$ (maximum singular value of $S$), is also bounded.*

In the above model, $Z_1^e$ is the cause of $X^e$ and $Y^e$ but may not be directly observed. $Z_2^e$ may be arbitrarily correlated with $Z_1^e$ and $\epsilon^e$. We observe a scrambled transformation of $(Z_1^e, Z_2^e)$ in $X^e$. If $Z_2^e$ is an effect of $Y^e$ ($Z_2^e \leftarrow Y^e + N^e$), then $Z_2^e$ is an anti-causal variable. If $H^e$ causes both $\varepsilon^e$ ($\varepsilon^e \leftarrow H^e + \bar{N}^e$) and $Z_2^e$ ($Z_2^e \leftarrow H^e + \tilde{N}^e$), then $H^e$ is a confounder. In both these cases, $Z_2^e$ is spuriously correlated with the label $Y^e$. Consequently, the standard ERM based models estimate $Z_2^e$ from $X^e$ and end up being biased w.r.t the desired OOD model, which does not use $Z_2^e$. If Assumptions 5, 2 hold, then a linear model $\tilde{S}^\mathsf{T}\gamma$ ($X^e \xrightarrow{\text{predict}} \gamma^\mathsf{T}\tilde{S}X^e = \gamma^\mathsf{T}Z_1^e$) solves the OOD problem in equation 1 ($\Phi^* = \tilde{S}$, $m = \gamma^\mathsf{T}$ in Proposition 1) and it relies only on $Z_1^e$. In the supplement (Proposition 17), we prove that ERM based models do not recover $\tilde{S}^\mathsf{T}\gamma$.

**Assumption 6.** *Linear general position.* *A set of training environments $\mathcal{E}_{tr}$ is said to lie in a linear general position of degree $r$ for some $r \in \mathbb{N}$ if $|\mathcal{E}_{tr}| > n - r + n/r$ and for all non-zero $x \in \mathbb{R}^n$*

$$\mathsf{dim}\Big(\mathsf{span}\Big\{\mathbb{E}^e[X^e X^{e,\mathsf{T}}]x - \mathbb{E}^e[X^e \varepsilon^e]\Big\}_{e \in \mathcal{E}_{tr}}\Big) > n - r \tag{9}$$

*where $\mathsf{span}$ is the linear span, $\mathsf{dim}$ is the dimension, and recall $n$ is dimension of $X^e$. This assumption checks for diversity in the environments and holds almost everywhere (Arjovsky et al., 2019).*

**Assumption 7.** *Inductive bias.* *$\mathcal{H}_\Phi$ is a finite set of linear models parametrized by $\Phi \in \mathbb{R}^n$ (output $\Phi^\mathsf{T}X^e$). $\tilde{S}^\mathsf{T}\gamma \in \mathcal{H}_\Phi$. $\exists\, \omega > 0, \Omega > 0$, s.t. $\forall \Phi \in \mathcal{H}_\Phi$, $\omega \leq \|\Phi\|^2 \leq \Omega$ & $2\omega \leq \|\tilde{S}^\mathsf{T}\gamma\|^2 \leq \frac{2}{3+2\sqrt{2}}\Omega$.*

Informally stated, the above assumption requires the OOD optimal predictor $\tilde{S}^\mathsf{T}\gamma$ to lie in the interior of the search space and not on the boundary. If Assumptions 5, 7 hold, then Assumption 3 holds. Hence, we can use the bounds $L$ and $L'$ on $\ell$ and $\frac{\partial\ell(w\cdot\Phi(\cdot),\cdot)}{\partial w}\big|_{w=1.0}$ respectively in our next result. Define the minimum eigenvalue across all $\Sigma^e$ as $\lambda_{\min} = \min_{e \in \mathcal{E}_{tr}} \lambda_{\min}(\Sigma_e)$, $\epsilon_{\mathsf{th}} = \frac{(24-16\sqrt{2})}{3}\frac{\pi^{\min}}{|\mathcal{E}_{tr}|}(\omega\lambda_{\min})^2$ and $\tau = \frac{1}{2\omega\lambda_{\min}}\sqrt{\frac{3|\mathcal{E}_{tr}|}{2\pi^{\min}}}$. Next, we analyze how EIRM learns $\tilde{S}^\mathsf{T}\gamma$.

**Proposition 5.** *Let $\ell$ be the square loss. For every $\epsilon \in (0, \epsilon_{\mathsf{th}})$ and $\delta \in (0, 1)$, if Assumptions 5, 6 (with $r = 1$), 7 hold and if the number of data points $|D|$ is greater than $\frac{16L'^4}{\epsilon^2}\log\big(\frac{2|\mathcal{H}_\Phi|}{\delta}\big)$, then with a probability at least $1 - \delta$, every solution $\hat{\Phi}$ to EIRM (equation 6) satisfies $\hat{\Phi} = (\tilde{S}^\mathsf{T}\gamma)\alpha$, where $\alpha \in \big[\frac{1}{1+\tau\sqrt{\epsilon}}, \frac{1}{1-\tau\sqrt{\epsilon}}\big]$.*

**Proof Sketch.** In learning theory it is common to analyze the concentration of empirical risks around the expected risks. In our case, we have a target ideal solution to equation 1 ($\tilde{S}^\mathsf{T}\gamma$) and we want our empirical solutions to concentrate around that. A direct finite sample approximation of equation 4 is hard to analyze. Therefore, we introduce an intermediate problem in equation 5 and then develop a finite sample approximation of it in equation 6. We first show that solving equation 5 leads to solutions in the neighborhood of the target. To show this we use the linear general position assumption. Next, we connect equation 6 and equation 5 using our new estimator for $R'$ and Hoeffding's inequality.

**Implications of Proposition 5. 1. Convergence rate of ERM vs. EIRM:** Recall that $\epsilon$ is the slack on IRM penalty $R'$. If $\epsilon$ is sufficiently small and the data grows as $\mathcal{O}(\frac{1}{\epsilon^2})$, every solution $\hat{\Phi}$ to EIRM (equation 6) is in $\sqrt{\epsilon}$ radius of the OOD solution, i.e., $\|\hat{\Phi} - \tilde{S}^\mathsf{T}\gamma\| = \mathcal{O}(\sqrt{\epsilon})$. We contrast these rates to ones in the covariate shift setting (Section 3.3.1). Let $\mathbb{E}[Y^e|X^e = x] = \Psi^\mathsf{T}x$. If the data grows as $\mathcal{O}(\frac{1}{\nu^2})$, then both ERM and EIRM solution converge to $\Psi$ as $\|\hat{\Phi} - \Psi\| = \mathcal{O}(\sqrt{\nu})$ (from Proposition 4). This shows that EIRM works in more settings (Proposition 4, 5) than ERM while matching the convergence rate of ERM.

**2. Sample complexity grows polynomially in data dimension to ensure OOD generalization:** Next, we set $\epsilon = \mu\epsilon_{\mathsf{th}}$ with $\mu \in [0, 1)$, and $|\mathcal{E}_{tr}| = 2n$ (satisfies Assumption 6 for $r = 1$). A simple manipulation of terms in Proposition 5 shows that sample complexity with quadratic growth in data dimension $n$, $\mathcal{O}\big(\frac{n^2}{\mu^2}\log\big(\frac{2|\mathcal{H}_\Phi|}{\delta}\big)\big)$, ensures $\hat{\Phi} = (\tilde{S}^\mathsf{T}\gamma)\alpha$ with $\alpha \in \big[\frac{1}{1+\sqrt{\mu}(\sqrt{2}-1)}, \frac{1}{1-\sqrt{\mu}(\sqrt{2}-1)}\big]$.

**3. Comparison with Proposition 2:** Lastly, we contrast sample complexity of EIRM in Proposition 5, $\mathcal{O}(\frac{1}{\epsilon^2})$, to Proposition 2, $\mathcal{O}(\max\{\frac{1}{\epsilon^2}, \frac{1}{\nu^2}\})$; the additional distributional assumptions in Proposition 5 help arrive at a lower sample complexity of $\mathcal{O}(\frac{1}{\epsilon^2})$. The bound in Proposition 2, $\mathcal{O}(\max\{\frac{1}{\epsilon^2}, \frac{1}{\nu^2}\})$, is larger than the one in Proposition 4, $\mathcal{O}(\frac{1}{\epsilon^2})$, and Proposition 5, $\mathcal{O}(\frac{1}{\epsilon^2})$, but is more general as it is agnostic to the distributional assumptions.

**A simple illustration summarizing Propositions 4, 5:** Set $S$ to identity in Assumption 5. Recall $Z_1^e$ and $Z_2^e$ from Assumption 5. Since $S$ is identity $X^e$ can be written as $[X_1^e, X_2^e]$, where $X_1^e = Z_1^e$ and $X_2^e = Z_2^e$. If $X_2^e \perp \varepsilon^e$, then $\mathbb{E}[Y^e|X^e]$ is invariant and Assumption 4 holds. This corresponds to the setup in Figure 1a). We can now use Proposition 4 and deduce that ERM and IRM have same sample complexity and end up learning the ideal model that only uses the causal features $X_1^e$. If $X_2^e \leftarrow \varepsilon^e + N^e$, then this corresponds to the setup Figure 1b), $X_1^e$ is the cause and $X_2^e$ is spuriously correlated with label $Y^e$ through the confounder $\varepsilon^e$. If $X_2^e \leftarrow Y^e + N^e$, then this corresponds to the setup in Figure 1c), $X_1^e$ is the cause and $X_2^e$ is anti-causally related to the label $Y^e$. In both these cases, the ideal OOD solution that solves equation 1 will only exploit $X_1^e$ to make predictions. From Propositions 5, it follows that IRM when fed with $\mathcal{O}(\frac{1}{\epsilon^2})$ samples, it is in $\sqrt{\epsilon}$ radius of the target OOD solution, while ERM is asymptotically biased and exploits $X_2^e$ (Proposition 17).

We define a polynomial version of the model in Assumption 5. We only need to change $Y^e \leftarrow \gamma^\mathsf{T}(Z_1^e) + \varepsilon^e$ to $Y^e \leftarrow \gamma^t \zeta_p(Z_1^e) + \varepsilon^e$. $\zeta_p$ is a polynomial feature map of degree $p$ defined as $\zeta_p : \mathbb{R}^c \to \mathbb{R}^{c'}$, where $c$ is the dimension of the input $Z_1^e$, $\zeta_p(W) = (W, W \otimes W, \ldots, (W \otimes W...\text{p times} \otimes W)) = ((W^{\otimes i})_{i=1}^p)$ and $\otimes$ is the Kronecker product. Also, $c' = \sum_{i=1}^p c^i$. Can we directly use the analysis from the linear case by transforming $X^e$ appropriately? No, we first need to find an appropriate transformation for the scrambling matrix $S$ that satisfies the conditions (invertibiltiy) in Assumption 5 while maintaining a linear relationship between transformations of $X^e$ and $Z^e$. We present the main result informally below (details are in the supplement).

**Proposition 6.** *(Informal statement) For sufficiently small $\epsilon$ and $\delta \in (0, 1)$, if Assumptions similar to Proposition 5 hold and $|D| \geq \frac{16L'^4}{\epsilon^2} \log(\frac{2|\mathcal{H}_\Phi|}{\delta})$, then with a probability at least $1 - \delta$, every solution $\hat{\Phi}$ to EIRM (equation 6) satisfies $\hat{\Phi} = \bar{\bar{S}}^\mathsf{T}\gamma(\alpha)$, where $\bar{\bar{S}}^\mathsf{T}\gamma$ is the OOD optimal model (defined in the supplement), $\alpha \in [\frac{1}{1+\tau\sqrt{\epsilon}}, \frac{1}{1-\tau\sqrt{\epsilon}}]$.*

**Insights from the polynomial case and infinite hypothesis case.** In the polynomial case, we adapt the linear general position Assumption 6, the number of environments $|\mathcal{E}_{tr}|$ are now required to grow as $\mathcal{O}(n^p)$. As a result, in the sample complexity analysis we discussed, we replace $n$ with $n^p$ to obtain that a sample complexity of $\mathcal{O}\left(\frac{n^{2p}}{\mu^2} \log\left(\frac{2|\mathcal{H}_\Phi|}{\delta}\right)\right)$ ensures $\hat{\Phi} = \bar{\bar{S}}^\mathsf{T}\gamma(\alpha)$ with $\alpha \in [\frac{1}{1+\sqrt{\mu}(\sqrt{2}-1)}, \frac{1}{1-\sqrt{\mu}(\sqrt{2}-1)}]$. In the infinite hypothesis case, the main change in the results is that we replace $|\mathcal{H}_\Phi|$ with an appropriate model complexity metric (Shalev-Shwartz & Ben-David, 2014). Consider Proposition 5, a sample complexity of $\mathcal{O}\left(\frac{n^3}{\mu^2} \log\left(\frac{n}{\mu}\right)\right)$ ensures $\hat{\Phi} = (\tilde{S}^\mathsf{T}\gamma)\alpha$ with $\alpha \in [\frac{1}{1+\sqrt{\mu}(\sqrt{2}-1)}, \frac{1}{1-\sqrt{\mu}(\sqrt{2}-1)}]$ in contrast to $\mathcal{O}\left(\frac{n^2}{\mu^2} \log\left(\frac{2|\mathcal{H}_\Phi|}{\delta}\right)\right)$ in the finite hypothesis case. We showed the benefits of IRM for polynomial models and other extensions (non-linear $S$) are future work. In the supplement, we provide a dialogue explaining how our work fits in the big picture.

## 4 Experiments

In this section, we discuss classification experiments (regression experiments with similar qualitative findings are in the supplement). We introduce three new variants of the colored MNIST (CMNIST) dataset in (Arjovsky et al., 2019). We divide the training data in MNIST digits into two environments ($e = 1, 2$) equally and the testing data in MNIST digits is assigned to another environment ($e = 3$). $X_g^e$: gray scale image of the digit, $Y_g^e$: label of the gray scale digit (digits $\geq 5$ have $Y_g^e = 1$ and digits $< 5$ have $Y_g^e = 0$). $X^e$: final colored image and $Y^e$: final label are generated as follows. Define Bernoulli variables $G, N, N^e$ that take a value 1 with probability $\theta$, $\beta$ and $\beta^e$ and 0 otherwise. Define a color variable $C^e$, where $C^e = 0$ is red and $C^e = 1$ is green. Let $\oplus$ denotes xor operation.

$$
\begin{aligned}
& Y_g^e \leftarrow \mathsf{L}(X_g^e), \mathsf{L} : \text{Human labeling,} \\
& Y^e \leftarrow \mathsf{L}(X_g^e) \oplus N, \text{ Corrupt the original labels with noise} \\
& C^e \leftarrow G(Y^e \oplus N^e) + (1 - G)(N \oplus N^e), \text{ Use } G \text{ to select b/w anti-causal or confounded} \\
& X^e \leftarrow T(X_g, C^e), T : \text{transformation to color the image}
\end{aligned}
\tag{10}
$$

If the probability $\theta = 1$, then $G = 1$ and that gives us back the original CMNIST in (Arjovsky et al., 2019), which we call anti-causal CMNIST (AC-CMNIST). If $\theta = 0$, then we get confounded colored MNIST (CF-CMNIST). If $0 < \theta < 1$, we get a hybrid dataset (HB-CMNIST). The above model in equation 10 has features of the model in Assumption 5, where $X_g^e$, $C^e$, L, $T$ take the role of $Z_1^e$, $Z_2^e$, $\gamma$, $S$. We set the noise $N$ parameter $\beta = 0.25$, and the parameter for $N^e$ in the three environments $[\beta^1, \beta^2, \beta^3] = [0.1, 0.2, 0.9]$. Color is spuriously correlated with the label; $\mathbb{P}(C^e = 1 | Y^e = 1)$ varies drastically in the three environments ($[0.9, 0.8, 0.1]$ for AC-CMNIST). In the variants of CMNIST we discussed, $\mathbb{P}(Y^e | X^e)$ varies across the environments. We now define a covariate shift based CMNIST (CS-CMNIST) ($\mathbb{P}(Y^e | X^e)$ is invariant). We use selection bias to induce spurious correlations. Generate a color $C^e$ uniformly at random. Select the pair $(X_g^e, C^e)$ with probability $1 - \psi^e$ if the label $Y_g^e$ and $C^e$ are the same, else select them with a probability $\psi^e$. If the pair is selected, then color the image $X^e \leftarrow T(X_g, C^e)$ and $Y^e \leftarrow Y_g^e$. Selection probability $\psi^e$ for the three environments are $[\psi^1, \psi^2, \psi^3] = [0.1, 0.2, 0.9]$. Due to the selection bias, color is spuriously correlated, $\mathbb{P}(C^e = 1 | Y^e = 1)$ varies drastically $[0.9, 0.8, 0.1]$. We provide the graphical models for the CMNIST variants and computations of $\mathbb{P}(C^e | Y^e)$, $\mathbb{P}(Y^e | X^e)$ in the supplement.

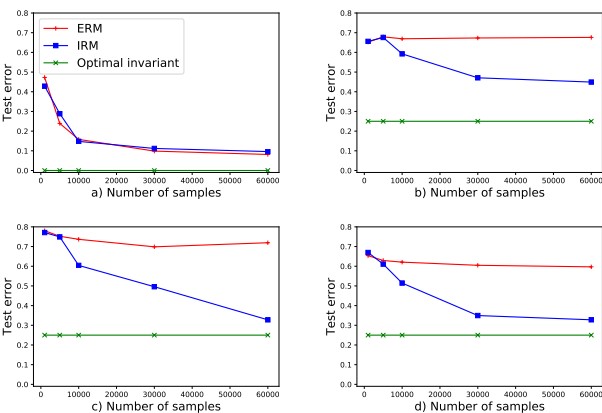

Figure 2: Comparisons: a) CS-CMNIST, b) CF-CMNIST, c) AC-CMNIST and d) HB-CMNIST.

## 4.1 RESULTS

We use the first two environments ($e = 1, 2$) to train and third environment ($e = 3$) to test. Other details of the training (models, hyperparameters, etc.) are in the supplement. For each of the above datasets, we run the experiments for different amounts of training data from $1000$ to up to $60000$ samples (10 trials for each data size). In Figure 2, we compare the models trained using IRM and ERM in terms of the classification error on the test environment $e = 3$ (a poor performance indicates model exploits the color) for varying number of train samples. We also provide the performance of the ideal hypothetical optimal invariant model. Observe that except for in the covariate shift setting where IRM and ERM are similar as seen in Figure 2a (as predicted from Proposition 4), IRM outperforms ERM in the remaining three datasets (as predicted from Proposition 5) as seen in Figure 2b-d. We further validate this claim through the regression experiments provided in the supplement. In CF-CMNIST, IRM achievs an error of $0.45$, which is much better than error of ERM ($0.7$) but is marginally better than a random guess. This suggests that confounder induced spurious correlations are harder to mitigate and may need more samples than needed in anti-causal case (AC-CMNIST).

## 5 CONCLUSION

We presented a sample complexity analysis of IRM to answer the question: when is IRM better than ERM (and vice-versa)? For distribution shifts such as the covariate shifts, we proved that both IRM and ERM have similar sample complexity and arrive at the desired OOD solution asymptotically. For distribution shifts involving confounders and (or) anti-causal variables and polynomial generative models, we proved that IRM is guaranteed to achieve the desired OOD solution while ERM can be asymptotically biased. We proposed new variants of original colored MNIST dataset from Arjovsky et al. (2019), which are more comprehensive and better capture how spurious correlations occur in reality. To the best of our knowledge, we believe this to be the first work that provides a rigorous characterization of impact of factors such as model complexity, number of environments on the sample complexity and its distance from the OOD solution in distribution shifts that go beyond covariate shifts.

## 6 ACKNOWLEDGEMENTS

This work was supported in part by the Rensselaer-IBM AI Research Collaboration (part of the IBM AI Horizons Network).

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

## 7 SUPPLEMENT

### 7.1 DIALOGUE ON IRM CONTINUED

[*In the original IRM paper by Arjovsky et al. (2019), we left two graduate students* ERIC *and* IRMA *strolling along the Parisian streets on a warm summer evening. A lot has transpired since then.* ERIC *is now holed up inside his room at Cité Universitaire and* IRMA *has returned to her parents' home in Provence. The two are talking through a Zoom call.*]

- ERIC: After reading Arjovsky et al. (2019) and Gulrajani & Lopez-Paz (2020), I was not sure of how to understand the settings where IRM is beneficial over ERM and vice-versa? From Gulrajani & Lopez-Paz (2020), I understood that ERM continues to be the state-of-the-art if sufficient care is taken in performing model selection.

- IRMA: But after reading this manuscript I understand that there are settings where no matter how one selects the model, ERM is bound to be at a disadvantage.

- ERIC: What you say seems to contradict my understanding of Gulrajani & Lopez-Paz (2020).

- IRMA: Actually, they ...

[IRMA*'s audio and video freeze for a minute, leaving* ERIC *to ponder this conundrum for a little bit on his own.* IRMA *stops her video and continues only with audio and the conversation is able to resume.*]

- IRMA: Sorry, how much did you hear?

- ERIC: You were just starting to explain why ERM can be at a disadvantage, but I didn't hear anything after that.

- IRMA: Okay, let me start my explanation again. There is no contradiction. The current manuscript says that in covariate shift settings, there maybe no clear winner. Many of the datasets considered in Gulrajani & Lopez-Paz (2020) are perhaps similar to covariate shift settings. Also, there could be another reason why IRM did not outperform ERM in Gulrajani & Lopez-Paz (2020) and I will come to it in a bit.

- ERIC: Yes, you are right! The datasets are human labeled images if I recall correctly. In these datasets it's safe to assume $\mathbb{P}(Y^e|X^e)$ is close to invariant across domains (as long as the cohort of humans in the different domains are not very different). What happens when $\mathbb{P}(Y^e|X^e)$ varies a lot?

- IRMA: Yes, when $\mathbb{P}(Y^e|X^e)$ varies a lot, IRM can be at an advantage over ERM. In this manuscript, I learned that when data generation involves confounders or anti-causal variables, it is possible to show that IRM methods converge to desirable OOD solutions with good convergence rates.

- ERIC: Interesting! I should read that. The experiment on colored MNIST by Arjovsky et al. (2019) contained anti-causal variables right? Is that the reason why IRM performed better than ERM?

- IRMA: Yes, you are right. In the current manuscript, the authors also develop other variants of colored MNIST (covariate shift based, confounder based, anti-causal based, and hybrid of confounder and anti-causal). IRM retains its advantage on all but the first dataset. I liked that IRM has advantage over ERM in confounded datasets as I believe many real datasets can have confounders generating spurious correlations.

- ERIC: Wow! These different variants of CMNIST sound exciting. Going back to the covariate shift case, do you think that we can perhaps come up with a finer criterion to say when IRM is better than ERM and vice-versa?

- IRMA: Yes, actually I have been thinking about this problem and would be happy to share my initial thoughts. Identifying a representation $\Phi^*$ that leads to invariant conditional distributions is crucial to the success of IRM. A subtle factor that is implicitly assumed is representations obtained from multiple domains should overlap.

- ERIC: Can you clarify what you mean by overlap?

- IRMA: Sure! Consider the dataset from domain 1, say photographs of birds, and domain 2, say sketches of birds. Imagine I have access to the oracle representation $\Phi^*$. Now I pass the data from domain 1 and 2 through $\Phi^*$ to get the representations. If these representations from the two domains live in very different parts of the representation space, then we cannot hope that IRM will offer any advantage. This is the other explanation, which I was going to come to, why IRM did not outperform ERM in Gulrajani & Lopez-Paz (2020).

- ERIC: How about when there is a complete overlap?

- IRMA: Yes, if there is a strong or complete overlap in the representations from the two domains, then it is possible that IRM can help. In fact I believe in such settings, if the ERM models trained on the two domains disagree a lot, then that can be a strong indication towards the models heavily exploiting spurious correlations. In such cases, I believe IRM can again offer some advantage.

- ERIC: I will try to put these ideas down on paper and meet you again for a discussion.
- IRMA: Great! I truly hope that it can be in person at the café in Palais-Royal where we first started this conversation. So long for now!

[*The students end their call.*]

## 7.2 SUPPLEMENTARY MATERIALS FOR EXPERIMENTS

In this section, we cover the supplementary materials for the experiments. The code to reproduce the results presented in this work can be found at `https://github.com/IBM/OoD`.

### 7.2.1 CLASSIFICATION

We first describe the model and other training details.

**Choice of $\mathcal{H}_\Phi$ and other training details** We use the same architecture for both ERM and IRM. We choose the architecture that was used in Arjovsky et al. (2019): 2 layer MLP. The first two layer consists of 390 hidden nodes, and the output layer has two nodes (for the two classes). We use ReLU activation in each layer, a regularization weight of 0.0011 is used for each layer. We use a learning rate of 4.9e-4, batch size of 512 for both ERM and IRM. We use 1000 gradient steps for IRM. As was done in the original IRM work (Arjovsky et al., 2019), we use a threshold on steps (190) after which a large penalty is imposed for violating the IRM constraint. We use the train domain validation set procedure described in Gulrajani & Lopez-Paz (2020) to select the penalty value from the set $\{1e4, 3.3e4, 6.6e4, 1e5\}$ (with 4:1 train-validation split). With the same learning rate, we observed ERM was slower at learning than IRM. To ensure ERM always converges we set the number of epochs to a very high value $100$ ($118k$ steps).

**A. Covariate shift based for CMNIST**

We provide the generative model for CS-CMNIST below.

$$
\begin{aligned}
Y_g^e &\leftarrow \mathsf{L}(X_g^e),\ C^e \leftarrow \mathsf{Uniform}(\{0,1\}) \\
U^e &\leftarrow \mathsf{Bernoulli}\Big(\big((C^e \oplus Y_g^e)\psi^e + \big(1 - (C^e \oplus Y_g^e)\big)\big(1 - \psi^e\big)\big)\Big) \\
X^e &\leftarrow T(X^g, C^e)|U^e = 1, Y^e \leftarrow Y_g^e|U^e = 1
\end{aligned}
\tag{11}
$$

**A.1. Compute $\mathbb{P}(C^e|Y^e)$ and $\mathbb{P}(Y^e|X^e)$.**

We compute $\mathbb{P}(C^e|Y^e)$ and $\mathbb{P}(Y^e|X^e)$ for the covariate shift based CMNIST described by equation 11. $\mathbb{P}(C^e|Y^e)$ helps us understand how the spurious correlations vary across the environments. $\mathbb{P}(Y^e|X^e)$ helps us understand if the covariate shift condition is satisfied or not. Compute the probability $\mathbb{P}(C^e|Y^e = 1) = \mathbb{P}(C^e|Y_g^e = 1, U^e = 1)$ as follows.

$$
\begin{aligned}
\mathbb{P}(C^e = 1|Y_g^e = 1, U^e = 1) &= \frac{\mathbb{P}(C^e = 1, Y_g^e = 1|U^e = 1)}{\mathbb{P}(C^e = 1, Y_g^e = 1|U^e = 1) + \mathbb{P}(C^e = 0, Y_g^e = 1|U^e = 1)} \\
\mathbb{P}(C^e = 1, Y_g^e = 1|U^e = 1) &= \frac{\mathbb{P}(C^e = 1, Y_g^e = 1, U^e = 1)}{\sum_{a,b} \mathbb{P}(C^e = a, Y_g^e = b, U^e = 1)} = \frac{1}{2}(1 - \psi^e) \\
\mathbb{P}(C^e = 0, Y_g^e = 1|U^e = 1) &= \frac{\mathbb{P}(C^e = 0, Y_g^e = 1, U^e = 1)}{\sum_{a,b} \mathbb{P}(C^e = a, Y_g^e = b, U^e = 1)} = \frac{1}{2}\psi^e \\
\mathbb{P}(C^e = 1|Y_g^e = 1, U^e = 1) &= (1 - \psi^e)
\end{aligned}
\tag{12}
$$

From the above simplification we gather that $\mathbb{P}(C^e = 1|Y^e = 1)$ is 0.9, 0.8 and 0.1 in the three environments. Define $p_l = P(Y_g^e = 1|X_g^e)$. In MNIST data it is reasonable to assume deterministic labeling, i.e. $p_l = 1$ or $p_l = 0$. From the way the data is constructed we can assume that $T$ is invertible, i.e. from colored image $X^e$ we can get back the grayscale image and the color $X_g^e, C^e$. We now move to computing $\mathbb{P}(Y^e|X^e)$. We assume $Y_g^e = 1$ in the simplifcation below.

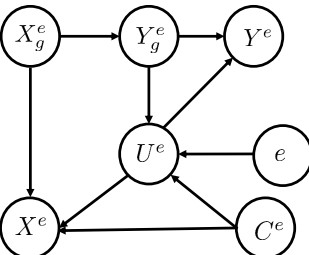

$X_g^e$   grayscale image   $U^e$   selection variable

$Y_g^e$   digit label   $e$   environment index

$C^e$   color index   $X^e$   colored image

$Y^e$   final label

Figure 3: Graphical model for CS-CMNIST

$$\mathbb{P}(Y_g^e = 1, C^e = 0, U^e = 1|X_g^e) = \mathbb{P}(Y_g^e = 1|X_g^e)\mathbb{P}(C^e = 0)\mathbb{P}(U^e = 1|Y_g^e = 1, C^e = 0)$$
$$= 0.5p_l\psi_e$$
$$\mathbb{P}(Y_g^e = 0, C^e = 0, U^e = 1|X_g^e) = \mathbb{P}(Y_g^e = 0|X_g^e)\mathbb{P}(C^e = 0)\mathbb{P}(U^e = 1|Y_g^e = 1, C^e = 0)$$
$$= 0.5(1 - p_l)(1 - \psi_e)$$
$$\mathbb{P}(Y^e = 1|X^e) = \mathbb{P}(Y_g^e = 1|X_g^e, C^e = 0, U^e = 1) = \qquad (13)$$
$$\frac{\mathbb{P}(Y_g^e = 1, C^e = 0, U^e = 1|X_g^e)}{\mathbb{P}(Y_g^e = 1, C^e = 0, U^e = 1|X_g^e) + \mathbb{P}(Y_g^e = 0, C^e = 0, U^e = 1|X_g^e)}$$
$$\mathbb{P}(Y^e = 1|X^e) = \frac{p_l\psi_e}{2p_l\psi_e + 1 - p_l - \psi_e}$$

Observe that under standard assumption of deterministic labeling $p_l = 1$, $\mathbb{P}(Y^e = 1|X^e)$ remains invariant and for high values of $p_l$ it is relatively stable across the environments.

**A.2 Graphical model for covariate shift based CMNIST.** In Figure 3, we provide the graphical model for covariate shift based CMNIST described in equation 11.

**A.3. Results with numerical values and standard errors.** In Table 2, we provide the numerical values for the results showed in the Figure 2 a) along with the standard errors.

**B. Colored MNIST with anti-causal variables (AC-CMNIST)**

**B.1. Compute $\mathbb{P}(C^e|Y^e)$ and $\mathbb{P}(Y^e|X^e)$.** We first compute $\mathbb{P}(C^e|Y^e)$. $\mathbb{P}(C^e = 1|Y^e = 1) = \mathbb{P}(Y^e \oplus N^e = 1|Y^e = 1) = \mathbb{P}(N^e = 0|Y^e = 1) = \beta^e$. Therefore, $\mathbb{P}(C^e = 1|Y^e = 1)$ is $0.9, 0.8$ and $0.1$ in environments $1, 2$, and $3$ respectively. Next, we compute $\mathbb{P}(Y^e|X^e)$. We assume $Y_g^e = 1$ in the simplfication below. We also assume deteministic labeling. In the simplfication that follows we use $\beta = 0.25$.

| Method | Number of samples | Test error |
|--------|-------------------|------------|
| ERM | 1000 | $47.27 \pm 0.72$ |
| IRM | 1000 | $42.84 \pm 0.94$ |
| ERM | 5000 | $23.91 \pm 1.23$ |
| IRM | 5000 | $28.85 \pm 5.21$ |
| ERM | 10000 | $15.80 \pm 0.63$ |
| IRM | 10000 | $14.80 \pm 0.30$ |
| ERM | 30000 | $9.86 \pm 0.32$ |
| IRM | 30000 | $11.20 \pm 0.64$ |
| ERM | 60000 | $8.15 \pm 0.37$ |
| IRM | 60000 | $9.62 \pm 0.67$ |

Table 2: Comparison of ERM vs IRM: CS-CMNIST

$$
\begin{aligned}
&\mathbb{P}(Y^e = 1, C^e = 0|X_g^e) = \\
&\mathbb{P}(Y^e = 1, C^e = 0|X_g^e) = \mathbb{P}(Y^e = 1|Y_g^e = 1)\mathbb{P}(C^e = 0|Y^e = 1) = 0.75\beta^e \\
&\mathbb{P}(Y^e = 0, C^e = 0|X_g^e) = \\
&\mathbb{P}(Y^e = 1, C^e = 0|X_g^e) = \mathbb{P}(Y^e = 0|Y_g^e = 1)\mathbb{P}(C^e = 0|Y^e = 0) = 0.25(1 - \beta^e) \\
&\mathbb{P}(Y^e = 1|X^e) = \mathbb{P}(Y^e = 1|X_g^e, C^e = 0) = \frac{3\beta^e}{2\beta^e + 1} \\
&\mathbb{P}(Y^e = 1|X^e) = 0.25 \quad \text{(For environment 1)} \\
&\mathbb{P}(Y^e = 1|X^e) = 0.428 \quad \text{(For environment 2)} \\
&\mathbb{P}(Y^e = 1|X^e) = 0.964 \quad \text{(For environment 2)}
\end{aligned}
\tag{14}
$$

**B.2 Graphical model for anti-causal CMNIST** In Figure 4, we provide the graphical model for AC-CMNIST described in equation 10 (for $G = 1$).

**B.3. Results with numerical values and standard errors.** In Table 3, we provide the numerical values for the results showed in the Figure 2 c) along with the standard errors.

## C. Colored MNIST with confounded variables (CF-CMNIST)

**C.1. Compute $\mathbb{P}(C^e|Y^e)$ and $\mathbb{P}(Y^e|X^e)$.** We start by computing $\mathbb{P}(C^e|Y^e)$. Recall that $C^e = (N \oplus N^e)$. In the simplfication that follows we use $\beta = 0.25$.

$$
\mathbb{P}(N = 0|Y^e = 1) = \frac{\mathbb{P}(N = 0, Y^e = 1)}{\mathbb{P}(N = 0, Y^e = 1) + \mathbb{P}(N = 1, Y^e = 1)} = \frac{0.75 * 0.5}{0.75 * 0.5 + 0.25 * 0.5} = 0.75
\tag{15}
$$

We use the above to compute $\mathbb{P}(C^e = 0|Y^e = 1) = \mathbb{P}(N = 0, N^e = 0|Y^e = 1) + \mathbb{P}(N = 1, N^e = 1|Y^e = 1) = (1 - \beta^e)0.75 + 0.25\beta^e = 0.75 - 0.5\beta^e$. For environment 1, 2, 3 the above probability $\mathbb{P}(C^e = 0|Y^e = 1)$ is 0.7, 0.65 and 0.30 respectively. Next, we compute the probability $\mathbb{P}(Y^e|X^e)$. Suppose $Y_g^e = 1$ for the calculation below. We also assume deterministic labeling.

| Method | Number of samples | Test error |
|--------|-------------------|------------|
| ERM | 1000 | $78.04 \pm 0.70$ |
| IRM | 1000 | $77.12 \pm 1.00$ |
| ERM | 5000 | $75.23 \pm 1.04$ |
| IRM | 5000 | $74.89 \pm 1.08$ |
| ERM | 10000 | $74.68 \pm 1.23$ |
| IRM | 10000 | $60.42 \pm 1.72$ |
| ERM | 30000 | $69.87 \pm 0.39$ |
| IRM | 30000 | $49.61 \pm 2.57$ |
| ERM | 60000 | $71.96 \pm 0.55$ |
| IRM | 60000 | $32.78 \pm 2.70$ |

Table 3: Comparison of ERM vs IRM: AC-CMNIST

$$
\begin{aligned}
&\mathbb{P}(Y^e = 1, C^e = 1|X_g^e) = \\
&\mathbb{P}(Y^e = 1, C^e = 1|X_g^e) = \mathbb{P}(Y^e = 1|Y_g^e = 1)\mathbb{P}(C^e = 1|N = 0) = 0.75\beta^e \\
&\mathbb{P}(Y^e = 0, C^e = 1|X_g^e) = \\
&\mathbb{P}(Y^e = 1, C^e = 1|X_g^e) = \mathbb{P}(Y^e = 0|Y_g^e = 1)\mathbb{P}(C^e = 1|N = 1) = 0.25(1 - \beta^e) \\
&\mathbb{P}(Y^e = 1|X^e) = \mathbb{P}(Y^e = 1|X_g^e, C^e = 1) = \frac{3\beta^e}{2\beta^e + 1} \\
&\mathbb{P}(Y^e = 1|X^e) = 0.25 \quad \text{(For environment 1)} \\
&\mathbb{P}(Y^e = 1|X^e) = 0.428 \quad \text{(For environment 2)} \\
&\mathbb{P}(Y^e = 1|X^e) = 0.964 \quad \text{(For environment 2)}
\end{aligned}
\tag{16}
$$

**C.2 Graphical model for confounded CMNIST.** In Figure 4, we provide the graphical model for confounded CMNIST described in equation 10 (for $G = 0$).

**C.3. Results with numerical values and standard errors.** In Table 4, we provide the numerical values for the results showed in the Figure 2 b) along with the standard errors.

**D. Colored MNIST with anti-causal variables and confounded variables (HB-CMNIST)**

**D.1. Compute $\mathbb{P}(C^e|Y^e)$ and $\mathbb{P}(Y^e|X^e)$.**

We start by computing $\mathbb{P}(C^e|Y^e)$. Recall that $C^e = G(Y^e \oplus N^e) + (1 - G)(N \oplus N^e)$. $G = 1$ with probability $\theta$ and 0 otherwise.

$$
\mathbb{P}(C^e = 1|Y^e = 1) = \frac{\mathbb{P}(C^e = 1, Y^e = 1)}{\mathbb{P}(C^e = 1, Y^e = 1) + \mathbb{P}(C^e = 0, Y^e = 1)}
\tag{17}
$$

$$
\mathbb{P}(C^e = 1, Y^e = 1) = \theta(1 - \beta^e) + (1 - \theta)(0.25 + 0.5\beta^e)
\tag{18}
$$
$$
\mathbb{P}(C^e = 0, Y^e = 1) = \theta(\beta^e) + (1 - \theta)(0.75 - 0.25\beta^e)
\tag{19}
$$

We used $\theta = 0.8$ in the experiments. $\mathbb{P}(Y^e|X^e)$ can be computed on the same lines as was shown for anti-causal and confounded model and it varies significantly across the environments.

**D.2 Graphical model for confounded CMNIST.**

In Figure 4, we provide the graphical model for confounded CMNIST described in equation 10 (for $0 < \theta < 1$).

**D.3. Results with numerical values and standard errors.** In Table 5, we provide the numerical values for the results showed in the Figure 2 d) along with the standard errors.

| Method | Number of samples | Test error |
|--------|-------------------|------------|
| ERM | 1000 | $65.21 \pm 0.64$ |
| IRM | 1000 | $65.60 \pm 0.53$ |
| ERM | 5000 | $67.91 \pm 0.40$ |
| IRM | 5000 | $67.59 \pm 1.14$ |
| ERM | 10000 | $66.92 \pm 0.30$ |
| IRM | 10000 | $59.26 \pm 1.62$ |
| ERM | 30000 | $67.32 \pm 0.28$ |
| IRM | 30000 | $47.01 \pm 1.37$ |
| ERM | 60000 | $67.62 \pm 0.31$ |
| IRM | 60000 | $44.92 \pm 0.84$ |

Table 4: Comparison of ERM vs IRM: CF-CMNIST

| Method | Number of samples | Test error |
|--------|-------------------|------------|
| ERM | 1000 | $65.38 \pm 0.56$ |
| IRM | 1000 | $66.98 \pm 0.61$ |
| ERM | 5000 | $62.92 \pm 0.99$ |
| IRM | 5000 | $61.09 \pm 1.74$ |
| ERM | 10000 | $62.08 \pm 1.19$ |
| IRM | 10000 | $51.46 \pm 1.22$ |
| ERM | 30000 | $60.48 \pm 0.42$ |
| IRM | 30000 | $34.96 \pm 1.47$ |
| ERM | 60000 | $59.70 \pm 0.72$ |
| IRM | 60000 | $32.80 \pm 0.55$ |

Table 5: Comparison of ERM vs IRM: HB-CMNIST

### 7.2.2 REGRESSION

We use the same structure for the generative model as described by Arjovsky et al. (2019). We work with the four different variants on the same lines as CMNIST (covariate shift based, confounded, anti-causal, hybrid). The comparisons in Arjovsky et al. (2019) were for anti-causal and hybrid models. The general model is written as

$$
\begin{aligned}
H^e &\leftarrow \mathcal{N}(0, \sigma_e^2 \mathsf{I}_s) \\
X_1^e &\leftarrow \mathcal{N}(0, \sigma_e^2 \mathsf{I}_s) + W_{h \to 1} H^e \\
Y^e &\leftarrow W_{1 \to y}^e X_1^e + \mathcal{N}(0, \sigma_e^2) + W_{h \to y} H^e \\
X_2^e &\leftarrow W_{y \to 2} Y^e + \mathcal{N}(0, \mathsf{I}_s) + W_{h \to 2} H^e
\end{aligned}
\tag{20}
$$

$H^e$ is the hidden confounder, $X^e = [X_1^e, X_2^e]$ is the observed covariate vector, $Y^e$ is the label. Different $W$'s correspond to the weight vectors that multiply with the covariates and the confounders. The four datasets differ in the weight $W$ vectors and we describe them below. $\sigma_e$ is environment

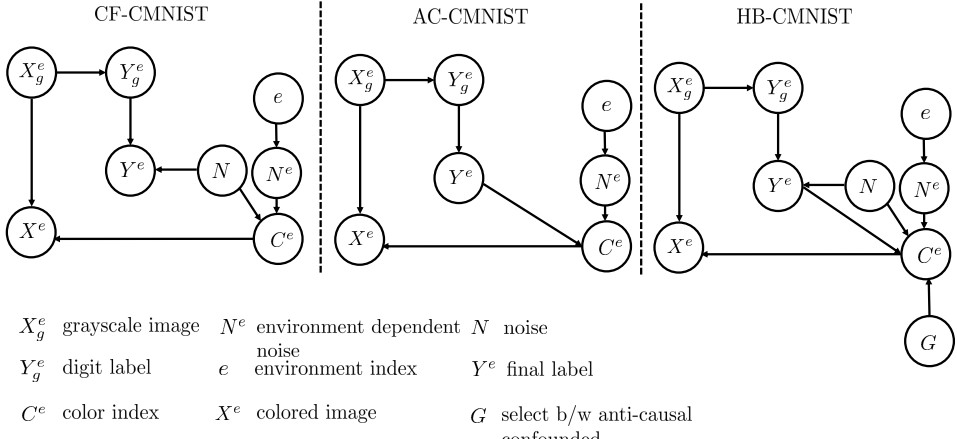

Figure 4: Graphical model for CF-CMNIST, AC-CMNIST, HB-CMNIST

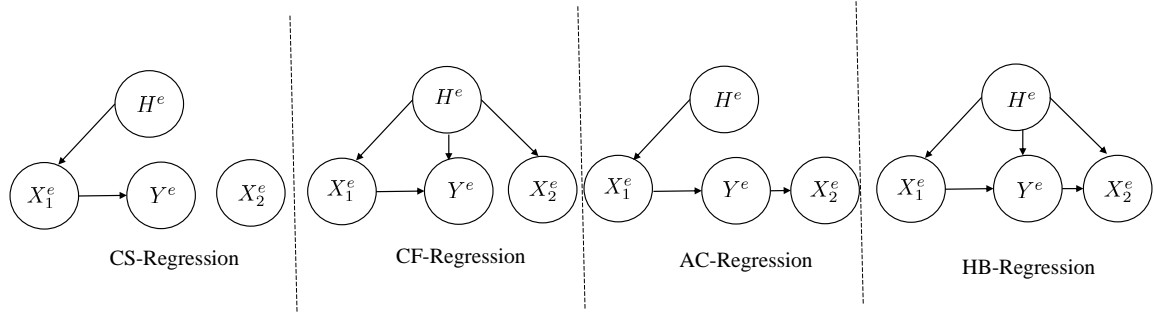

Figure 5: Graphical models for the different regression datasets

dependent standard deviation, we use two training environments with $\sigma_1 = 0.2$ and $\sigma_2 = 2.0$ respectively and both environments .

- Covariate shift case (CS-regresion): In this case, we fix $W_{h \to 2}$, $W_{h \to y}$, and $W_{y \to 2}$ to zero and we draw each entry of $W_{1 \to y}$ from $\frac{1}{s}\mathcal{N}(0, 1)$ and set $W_{h \to 1}$ to identity.

- Confounded variable case (CF-regression): Set $W_{y \to 2}$ to zero and we draw each entry in $W_{1 \to y}, W_{h \to 1}, W_{h \to 2}, W_{h \to y}$ from $\frac{1}{s}\mathcal{N}(0, 1)$.

- Anti-causal variable case (AC-regression): Set $W_{h \to y}$,$W_{h \to 1}$ and $W_{h \to 2}$ to zero and draw each entry of $W_{1 \to y}$ and $W_{y \to 2}$ from $\frac{1}{s}\mathcal{N}(0, 1)$.

- Hybrid confounded and anti-causal variable case (HB-regression): Draw each $W_{h \to y}$,$W_{h \to 1}$, $W_{h \to 2}$, $W_{1 \to y}$ and $W_{y \to 2}$ from $\frac{1}{s}\mathcal{N}(0, 1)$.

We also present the graphical models for the four types of models used in Figure 5.

### 7.2.3 CHOICE OF $\mathcal{H}_\Phi$ AND OTHER TRAINING DETAILS

We use a linear model that takes as input $X^e$. For ERM we use standard linear regression from sklearn. For IRM, we use 50k gradient steps with learning rate 1e-3, batch size is equal to the size of the training data. We use the train domain validation set procedure described by Gulrajani & Lopez-Paz (2020) to select the penalty value from the set $\{0, 1e-5, 1e-4, 1e-3, 1e-2, 1e-1\}$ (with 4:1 train-validation split). We average the results over 25 trials.

### 7.2.4 RESULTS

We discuss results for the case when the length of the covariate vector $X^e$ is 10. The desired optimal invariant predictor is $W^* = [W_{1 \to y}, 0]$. We will compare ERM and IRM in terms of the model estimation error, i.e., the distance between the model estimated by the method $\hat{W}$ and the true model given as $\|\hat{W} - W^*\|^2$. In Figures 6, 7, 8, 9, we compare the model estimation error vs. the number of samples when the model. In these comparisons, we see that consistent with the classification experiments and predictions from Proposition 4 in the covariate shift case (See Figure 6) there is no clear winner between the two approaches. There are gains from using IRM in the other cases (Figures 7, 8, 9). However, in the confounder case in Figure 7, the gains from IRM appear in the low sample regime but are not there in the high sample regime. This is because for this setup the asymptotic bias of ERM is also very small. In addition to the Figures 6, 7, 8, 9, we provide the tables (see Table 6, 7, 8, 9) with the numerical values for the mean model estimation (and the standard error) error shown in the figures.

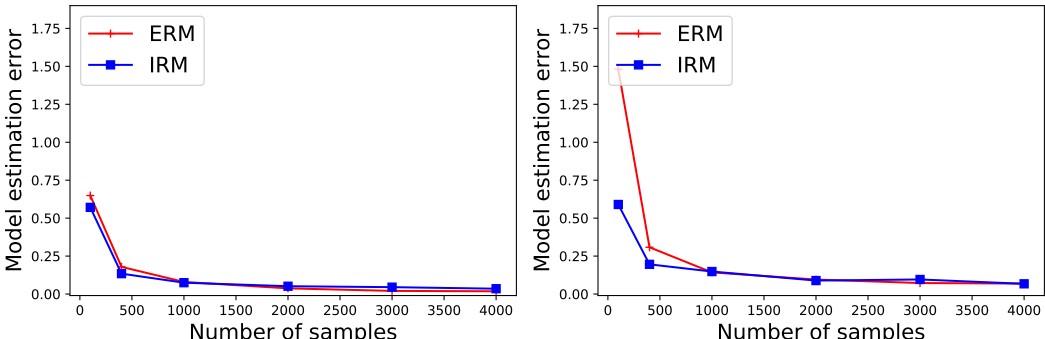

Figure 6: Comparisons: $n = 10$ CS-regression     Figure 7: Comparisons: $n = 10$ CF-regression

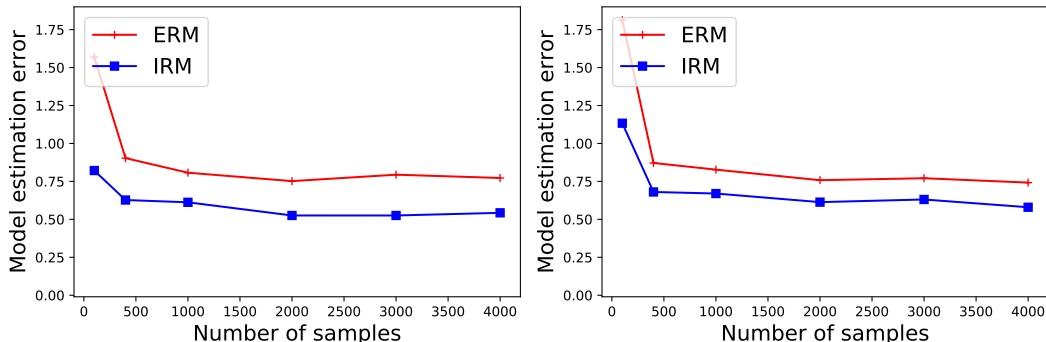

Figure 8: Comparisons: $n = 10$ AC-regression     Figure 9: Comparisons: $n = 10$ HB-regression

| Method | Number of samples | Model estimation error |
|--------|-------------------|------------------------|
| ERM | 50 | $0.65 \pm 0.06$ |
| IRM | 50 | $0.57 \pm 0.10$ |
| ERM | 200 | $0.18 \pm 0.02$ |
| IRM | 200 | $0.13 \pm 0.02$ |
| ERM | 500 | $0.08 \pm 0.005$ |
| IRM | 500 | $0.08 \pm 0.014$ |
| ERM | 1000 | $0.037 \pm 0.004$ |
| IRM | 1000 | $0.051 \pm 0.007$ |
| ERM | 1500 | $0.021 \pm 0.002$ |
| IRM | 1500 | $0.045 \pm 0.007$ |
| ERM | 2000 | $0.018 \pm 0.002$ |
| IRM | 2000 | $0.035 \pm 0.006$ |

Table 6: Comparison of ERM vs IRM: $n = 10$ CS-regression

| Method | Number of samples | Model estimation error |
|--------|-------------------|------------------------|
| ERM | 50 | $1.48 \pm 0.15$ |
| IRM | 50 | $0.59 \pm 0.10$ |
| ERM | 200 | $0.30 \pm 0.03$ |
| IRM | 200 | $0.20 \pm 0.03$ |
| ERM | 500 | $0.14 \pm 0.01$ |
| IRM | 500 | $0.15 \pm 0.02$ |
| ERM | 1000 | $0.10 \pm 0.02$ |
| IRM | 1000 | $0.09 \pm 0.01$ |
| ERM | 1500 | $0.07 \pm 0.01$ |
| IRM | 1500 | $0.10 \pm 0.01$ |
| ERM | 2000 | $0.07 \pm 0.01$ |
| IRM | 2000 | $0.07 \pm 0.01$ |

Table 7: Comparison of ERM vs IRM: $n = 10$ CF-regression

### 7.3 PROOFS FOR THE PROPOSITIONS

In the results to follow, we will rely on Hoeffding's inequality. We restate the inequality below for convenience.

**Lemma 1.** *(Hoeffding's inequality). Let $\theta_1, \ldots \theta_m$ be a sequence of i.i.d. random variables and assume that for all $i$, $\mathbb{E}[\theta_i] = \mu$ and $\mathbb{P}[a \leq \theta_i \leq b] = 1$. Then, for any $\epsilon > 0$*

$$\mathbb{P}\left[\frac{1}{m}|\sum_{i=1}^{m} \theta_i - \mu| > \epsilon\right] \leq 2\exp(-2\frac{m\epsilon^2}{(b-a)^2})$$

We restate the propositions from the main body. Next, we prove Proposition 1 from the main body of the manuscript.

| Method | Number of samples | Model estimation error |
|--------|-------------------|------------------------|
| ERM | 50 | $1.57 \pm 0.18$ |
| IRM | 50 | $0.82 \pm 0.20$ |
| ERM | 200 | $0.90 \pm 0.08$ |
| IRM | 200 | $0.63 \pm 0.08$ |
| ERM | 500 | $0.81 \pm 0.08$ |
| IRM | 500 | $0.61 \pm 0.05$ |
| ERM | 1000 | $0.75 \pm 0.07$ |
| IRM | 1000 | $0.53 \pm 0.04$ |
| ERM | 1500 | $0.79 \pm 0.07$ |
| IRM | 1500 | $0.53 \pm 0.05$ |
| ERM | 2000 | $0.77 \pm 0.08$ |
| IRM | 2000 | $0.54 \pm 0.04$ |

Table 8: Comparison of ERM vs IRM: $n = 10$ AC-regression

| Method | Number of samples | Model estimation error |
|--------|-------------------|------------------------|
| ERM | 50 | $1.81 \pm 0.19$ |
| IRM | 50 | $1.13 \pm 0.18$ |
| ERM | 200 | $0.87 \pm 0.09$ |
| IRM | 200 | $0.68 \pm 0.10$ |
| ERM | 500 | $0.83 \pm 0.09$ |
| IRM | 500 | $0.67 \pm 0.07$ |
| ERM | 1000 | $0.76 \pm 0.07$ |
| IRM | 1000 | $0.61 \pm 0.05$ |
| ERM | 1500 | $0.77 \pm 0.07$ |
| IRM | 1500 | $0.63 \pm 0.05$ |
| ERM | 2000 | $0.74 \pm 0.07$ |
| IRM | 2000 | $0.58 \pm 0.05$ |

Table 9: Comparison of ERM vs IRM: $n = 10$ HB-regression

**Proposition 7.** *If $\ell$ is square loss, and Assumptions 1,2 hold, then $m \circ \Phi^*$ solves the OOD problem (equation 1).*

*Proof.* Define a predictor $w \circ \Phi^*$, where $\Phi^*$ is defined in Assumption 1. Let us simplify the expression for the risk for this predictor $R^e(w \circ \Phi^*)$ using square loss for $\ell$. Recall $Z^e = \Phi^*(X^e)$

$$R^e(w \circ \Phi^*) = \mathbb{E}^e\left[\left(Y^e - \mathbb{E}^e\left[Y^e|Z^e\right] + \mathbb{E}^e\left[Y^e|Z^e\right] - (w \circ \Phi^*)(X^e)\right)^2\right]$$

$$= \mathbb{E}^e\left[\left(Y^e - \mathbb{E}^e\left[Y^e|Z^e\right]\right)^2\right] + \mathbb{E}^e\left[\left(\mathbb{E}^e\left[Y^e|Z^e\right] - (w \circ \Phi^*)(X^e)\right)^2\right] +$$

$$2\mathbb{E}^e\left[\left(Y^e - \mathbb{E}^e\left[Y^e|Z^e\right]\right)\left(\mathbb{E}^e\left[Y^e|Z^e\right] - (w \circ \Phi^*)(X^e)\right)\right]$$

$$= \mathbb{E}^e\left[\left(Y^e - m(Z^e)\right)^2\right] + \mathbb{E}^e\left[\left(m(Z^e) - w(Z^e)\right)^2\right] + 2\mathbb{E}^e\left[\left(Y^e - m(Z^e)\right)\left(m(Z^e) - w(Z^e)\right)\right]$$

$$= \mathbb{E}^e\left[\left(Y^e - m(Z^e)\right)^2\right] + \mathbb{E}^e\left[\left(m(Z^e) - w(Z^e)\right)^2\right]$$

$$= \xi^2 + \mathbb{E}^e\left[\left(m(Z^e) - w(Z^e)\right)^2\right]$$

$$(21)$$

In the above simplification in equation 21, we use the following equation 22 and equation 23, which rely on the law of total expectation.

$$\mathbb{E}^e\left[\left(Y^e - m(Z^e)\right)\left(m(Z^e) - w(Z^e)\right)\right] = \mathbb{E}^e\left[\mathbb{E}^e\left[\left(Y^e - m(Z^e)\right)\left(m(Z^e) - w(Z^e)\right)\Big|Z^e\right]\right]$$

$$= \mathbb{E}^e\left[\left(\mathbb{E}^e\left[Y^e|Z^e\right] - m(Z^e)\right)\left(m(Z^e) - w(Z^e)\right)\right] = 0$$

$$(22)$$

$$\mathbb{E}^e\left[\left(Y^e - m(Z^e)\right)^2\right] = \mathbb{E}^e\left[\mathbb{E}^e\left[\left(Y^e - m(Z^e)\right)^2\Big|Z^e\right]\right] = \mathbb{E}^e\left[\mathsf{Var}^e\left[Y^e|Z^e\right]\right] = \xi^2 \qquad (23)$$

In the last equality in equation 23, we use the Assumption 1 and obtain $\xi^2$. Therefore, substituting $w = m$ in equation 21 achieves a risk of $\xi^2$ for all the environments.

$$\forall e \in \mathcal{E}_{all}, \; R^e(m \circ \Phi^*) = \xi^2 \qquad (24)$$

Consider the environment $q$ that satisfies Assumption 2. Let us simplify the expression for the risk achieved by every predictor $f \in \mathcal{F}$ in the environment $q$ following the steps similar to equation 21.

$$R^q(f) = \mathbb{E}^q\left[\left(Y^q - \mathbb{E}^q[Y^q|Z^q]\right)^2\right] +$$

$$\mathbb{E}^q\left[\left(\mathbb{E}^q[Y^q|Z^q] - f(X^q)\right)^2\right] + 2\mathbb{E}^q\left[\left(Y^q - \mathbb{E}^q[Y^q|Z^q]\right)\left(\mathbb{E}^q[Y^q|Z^q] - f(X^q)\right)\right]$$

$$= \mathbb{E}^q\left[\left(Y^q - m(Z^q)\right)^2\right] + \mathbb{E}^q\left[\left(m(Z^q) - f(X^q)\right)^2\right] + 2\mathbb{E}^q\left[\left(Y^q - m(Z^q)\right)\left(m(Z^q) - f(X^q)\right)\right]$$

$$= \mathbb{E}^q\left[\left(Y^q - m(Z^q)\right)^2\right] + \mathbb{E}^q\left[\left(m(Z^q) - f(X^q)\right)^2\right]$$

$$= \xi^2 + \mathbb{E}^q\left[\left(m(Z^q) - f(X^q)\right)^2\right]$$

$$(25)$$

In the above simplification in equation 25, we use the following equation 26

$$\mathbb{E}^q\left[\left(Y^q - m(Z^q)\right)\left(m(Z^q) - f(X^q)\right)\right] = \mathbb{E}^q\left[\mathbb{E}^e\left[\left(Y^e - m(Z^q)\right)\left(m(Z^q) - f(X^q)\right)\Big|Z^q\right]\right]$$

$$= \mathbb{E}^q\left[\left(\mathbb{E}^q\left[Y^q|Z^q\right] - m(Z^q)\right)\left(m(Z^q) - f(X^q)\right)\right] = 0 \text{ (Last equality follows from the Assumption 2)}$$

$$(26)$$

Therefore, for environment $q$ satisfying Assumption 2 from equation 25, it follows that for all $f \in \mathcal{F}$, $R^q(f) \geq \xi^2$. Therefore, we can write that

$$\forall f \in \mathcal{F}, \ \max_{e \in \mathcal{E}_{all}} R^e(f) \geq R^q(f) \geq \xi^2 \tag{27}$$

Therefore, from equation 27 it directly follows that

$$\min_{f \in \mathcal{F}} \max_{e \in \mathcal{E}_{all}} R^e(f) \geq \xi^2 \tag{28}$$

We showed in equation 24, $R^e(m \circ \Phi^*) = \xi^2$ for all the environments.

Hence, $f = m \circ \Phi^*$ achieves the RHS of equation 28. This completes the proof. $\qquad \square$

Some of the proofs that we describe next take a few intermediate steps to build. Here we give a brief preview of the key ingredients that we developed to build these propositions.

- In Proposition 2, our goal is to carry out a sample complexity analysis of EIRM in the same spirit as ERM. However, there are two key challenges that we are faced with - i) the IRM penalty $R^{'}$ is not separable (as it is composed of terms involving squares of expectations) and ii) unlike ERM, IRM is a constrained optimization problem. To deal with i), we develop an estimator in the next section that allows us to re-express IRM penalty in a separable fashion. To deal with ii), we define a parameter $\kappa$ that measures the minimum separation between IRM penalty and $\epsilon$. We show that as long as this separation for all the predictors in the hypothesis class is positive, then we can rely on $\kappa$-representative property from Shalev-Shwartz & Ben-David (2014) applied to the new estimator that we build to show that the set of empirical invariant predictors $\hat{\mathcal{S}}^{\mathsf{IV}}(\epsilon)$ are the same as exact invariant predictors $\mathcal{S}^{\mathsf{IV}}(\epsilon)$.

- In Proposition 5, our goal is to show that approximate OOD can be achieved by IRM in the finite sample regime. This result builds on the infinite sample result from Arjovsky et al. (2019). In Theorem 9 in Arjovsky et al. (2019), it was shown that for linear models (defined in Assumption 5) obeying linear general position 6, if the gradient constraints in the exact gradient constraint IRM equation 4 are satisfied, then the OOD solution is achieved. We extend this result to show that if the constraints in the approximate $\epsilon$ penalty based IRM in equation 5 are satisfied, then we are guaranteed to be in the $\sqrt{\epsilon}$ neighborhood of the OOD solution. Note that this result is again in the infinite sample regime as it proves the approximation for solutions of the problem equation 5, which involves expectations w.r.t true distributions. Next, we exploit similar tools that we introduced to prove Proposition 2 to also prove the finite sample extension.

- In later sections, we show the generalizations to infinite hypothesis classes. In particular, we focus on parametric model families that are Lipschitz continuous. The extension to infinite hypothesis classes is based on carefully exploiting the covering number based techniques Shalev-Shwartz & Ben-David (2014) for the IRM penalty estimator that we introduced. We also provide generalizations of the results for linear models to polynomial models. To arrive at these results, we exploit some standard properties of tensor products.

### 7.3.1 EMPIRICAL ESTIMATOR OF $R^{'}$

Next, we define an estimator for $R^{'}$. We first simplify $R^{'}$ as follows.

Observe that

$$\nabla_{w|w=1.0} R^e(w \cdot \Phi) = \frac{\partial \mathbb{E}^e\big[\ell(w \cdot \Phi(X^e), Y^e)\big]}{\partial w}\Big|_{w=1.0} = \mathbb{E}^e\left[\frac{\partial \ell(w \cdot \Phi(X^e), Y^e)}{\partial w}\Big|_{w=1.0}\right]$$

and

$$\|\nabla_{w|w=1.0} R^e(w \cdot \Phi)\|^2 = \left(\frac{\partial \mathbb{E}^e\big[\ell(w \cdot \Phi(X^e), Y^e)\big]}{\partial w}\Big|_{w=1.0}\right)^2 = \left(\mathbb{E}^e\left[\frac{\partial \ell(w \cdot \Phi(X^e), Y^e)}{\partial w}\Big|_{w=1.0}\right]\right)^2 \tag{29}$$

In the above simplification, we used Leibniz integral rule and take the derivative inside the expectation.

Also we can write $\mathbb{E}[X]^2 = \mathbb{E}[AB]$, where $A$ and $B$ are independent and identical random variables with same distribution as $X$. Therefore, we consider two independent data points $(X^e, Y^e) \sim \mathbb{P}^e$ and $(\tilde{X}^e, \tilde{Y}^e) \sim \mathbb{P}^e$.

$$\|\nabla_{w|w=1.0} R^e(w \cdot \Phi)\|^2 = \mathbb{E}^e \left[ \left( \frac{\partial \ell(w \cdot \Phi(X^e), Y^e)}{\partial w} \Big|_{w=1.0} \right) \left( \frac{\partial \ell(w \cdot \Phi(\tilde{X}^e), \tilde{Y}^e)}{\partial w} \Big|_{w=1.0} \right) \right] \quad (30)$$

In the above the expectation $\mathbb{E}^e$ is taken over the joint distribution over pairs of distributions of pairs $(X^e, Y^e), (\tilde{X}^e, \tilde{Y}^e)$ from the same environment $e$.

We write

$$
\begin{aligned}
R^{'}(\Phi) &= \sum_{e \in \mathcal{E}_{tr}} \pi^e \|\nabla_{w|w=1.0} R^e(w \cdot \Phi)\|^2 \\
&= \sum_{e \in \mathcal{E}_{tr}} \pi^e \mathbb{E}^e \left[ \left( \frac{\partial \ell(w \cdot \Phi(X^e), Y^e)}{\partial w} \Big|_{w=1.0} \right) \left( \frac{\partial \ell(w \cdot \Phi(\tilde{X}^e), \tilde{Y}^e)}{\partial w} \Big|_{w=1.0} \right) \right]
\end{aligned}
\quad (31)
$$

In the above simplification, we used equation 30. Define a joint distribution $\tilde{\mathbb{P}}$ over the tuple $(e, (X^e, Y^e), (\tilde{X}^e, \tilde{Y}^e))$, where $e \sim \{\pi^o\}_{o \in \mathcal{E}_{tr}}$, $(X^e, Y^e) \sim \mathbb{P}^e$ and $(\tilde{X}^e, \tilde{Y}^e) \sim \mathbb{P}^e$. Also,

$$\tilde{\mathbb{P}}\Big( (e, (X^e, Y^e), (\tilde{X}^e, \tilde{Y}^e)) \Big) = \pi^e \mathbb{P}^e(X^e, Y^e) \mathbb{P}^e(\tilde{X}^e, \tilde{Y}^e) \quad (32)$$

We rewrite the above expression equation 31 in terms of an expectation w.r.t $\tilde{\mathbb{P}}$, which we represent as $\tilde{\mathbb{E}}$ follows

$$R^{'}(\Phi) = \tilde{\mathbb{E}} \left[ \left( \frac{\partial \ell(w \cdot \Phi(X^e), Y^e)}{\partial w} \Big|_{w=1.0} \right) \left( \frac{\partial \ell(w \cdot \Phi(\tilde{X}^e), \tilde{Y}^e)}{\partial w} \Big|_{w=1.0} \right) \right] \quad (33)$$

Define

$$\ell^{'}\Big( h, ((X^e, Y^e), (\tilde{X}^e, \tilde{Y}^e)) \Big) = \left( \frac{\partial \ell(w \cdot \Phi(X^e), Y^e)}{\partial w} \Big|_{w=1.0} \right) \left( \frac{\partial \ell(w \cdot \Phi(\tilde{X}^e), \tilde{Y}^e)}{\partial w} \Big|_{w=1.0} \right) \quad (34)$$

Substitute equation 34 in equation 33 to obtain

$$R^{'}(\Phi) = \tilde{\mathbb{E}} \left[ \ell^{'}\Big( h, ((X^e, Y^e), (\tilde{X}^e, \tilde{Y}^e)) \Big) \right] \quad (35)$$

We construct a simple estimator $\hat{R}^{'}(\Phi)$ by pairing the data points in each environment. For simplicity assume that each environment has even number of points. In environment $e$, which has $n_e$ points we construct $\frac{n_e}{2}$ pairs. Define a set of such pairs as

$$\tilde{D} = \{\{(x^e_{2i-1}, y^e_{2i-1}), (x^e_{2i}, y^e_{2i})\}_{i=1}^{\frac{n_e}{2}}\}_{e \in \mathcal{E}_{tr}} \quad (36)$$

$$\hat{R}^{'}(\Phi) = \frac{2}{|D|} \sum_{e \in \mathcal{E}_{tr}} \sum_{i=1}^{\frac{n_e}{2}} \ell^{'}\Big( h, ((X^e_{2i-1}, Y^e_{2i-1}), (\tilde{X}^e_{2i}, \tilde{Y}^e_{2i})) \Big) \quad (37)$$

There can be other estimators of $R^{'}(\Phi)$ where we separately estimate each term $\|\nabla_{w|w=1.0} R^e(w.\Phi)\|^2$ and $\pi^e$ in the summation. We rely on the above estimator equation 37 as its separability allows us to use standard concentration inequalities, e.g., Hoeffding's inequality.

### 7.3.2 $\epsilon$-REPRESENTATIVE TRAINING SET FOR $R$ AND $R^{'}$

We use the definition of $\epsilon$-representative sample from Shalev-Shwartz & Ben-David (2014) and state it appropriately for both $R$ and $R^{'}$.

**Definition 1.** *A training set $S$ is called $\epsilon$-representative (w.r.t. domain $\mathcal{Z}$, hypothesis $\mathcal{H}$, loss $\ell$ and distribution $\mathcal{D}$) if*

$$\forall h \in \mathcal{H}, |\hat{R}(h) - R(h)| \leq \epsilon \tag{38}$$

*where $R(h) = \mathbb{E}_{\mathcal{D}}[\ell(h(X), Y]$ and $(X, Y) \sim \mathcal{D}$.*

Following the above definition, we apply it to the set of points $\tilde{D}$ defined above in equation 36. $\tilde{D}$ is called $\epsilon$-representative w.r.t. domain $\mathcal{X}$, hypothesis $\mathcal{H}_{\Phi}$, loss $\ell^{'}$ (equation 33) and distribution $\tilde{\mathbb{P}}$ (equation 32) if

$$\forall \Phi \in \mathcal{H}_{\Phi}, |\hat{R}^{'}(\Phi) - R^{'}(\Phi)| \leq \epsilon \tag{39}$$

where $R^{'}(\Phi) = \tilde{\mathbb{E}}\left[\ell^{'}\left(h, \left((X^e, Y^e), (\tilde{X}^e, \tilde{Y}^e)\right)\right)\right]$ (from equation 34) and $(e, (X^e, Y^e), (\tilde{X}^e, \tilde{Y}^e)) \sim \tilde{\mathbb{P}}$.

Recall the definition of $\kappa$, $\kappa = \min_{\Phi \in \mathcal{H}_{\Phi}} |R^{'}(\Phi) - \epsilon|$. Next, we show that if $\tilde{D}$ is $\frac{\kappa}{2}$-representative w.r.t $\mathcal{X}$, $\mathcal{H}_{\Phi}$, loss $\ell^{'}$, distribution $\tilde{\mathbb{P}}$ then the set of invariant predictors in equation 6 ($\hat{\mathcal{S}}^{\mathsf{IV}}(\epsilon)$ ) and the set of invariant predictors in equation 5 ($\mathcal{S}^{\mathsf{IV}}(\epsilon)$) are equal.

**Lemma 2.** *If $\kappa > 0$ and $\tilde{D}$ is $\frac{\kappa}{2}$-representative w.r.t $\mathcal{X}$, $\mathcal{H}_{\Phi}$, loss $\ell^{'}$ and distribution $\tilde{\mathbb{P}}$ , then $\hat{\mathcal{S}}^{\mathsf{IV}}(\epsilon) = \mathcal{S}^{\mathsf{IV}}(\epsilon)$.*

*Proof.* First we show $\mathcal{S}^{\mathsf{IV}}(\epsilon) \subseteq \hat{\mathcal{S}}^{\mathsf{IV}}(\epsilon)$. From the definition of $\kappa$, $\kappa = \min_{\Phi \in \mathcal{H}_{\Phi}} |R^{'}(\Phi) - \epsilon|$ it follows that $\forall \Phi \in \mathcal{H}_{\Phi}$

$$|R^{'}(\Phi) - \epsilon| \geq \kappa \implies R^{'}(\Phi) \geq \epsilon + \kappa \text{ or } R^{'}(\Phi) \leq \epsilon - \kappa \tag{40}$$

Consider any $\Phi$ in $\mathcal{S}^{\mathsf{IV}}(\epsilon)$.

$$R^{'}(\Phi) \leq \epsilon \tag{41}$$

Given the definition of $\kappa$ and equation 40, we obtain

$$R^{'}(\Phi) \leq \epsilon \implies R^{'}(\Phi) \leq \epsilon - \kappa \tag{42}$$

Therefore, $\mathcal{S}^{\mathsf{IV}}(\epsilon) \subseteq \mathcal{S}^{\mathsf{IV}}(\epsilon - \kappa)$. Also, it follows from the definition of the set $\mathcal{S}^{\mathsf{IV}}(\epsilon)$ that $\mathcal{S}^{\mathsf{IV}}(\epsilon - \kappa) \subseteq \mathcal{S}^{\mathsf{IV}}(\epsilon)$. Hence,

$$\mathcal{S}^{\mathsf{IV}}(\epsilon) = \mathcal{S}^{\mathsf{IV}}(\epsilon - \kappa) \tag{43}$$

Consider any $\Phi$ in $\mathcal{S}^{\mathsf{IV}}(\epsilon)$

$$\begin{aligned} R^{'}(\Phi) &\leq \epsilon - \kappa \text{ (From equation 43)} \\ R^{'}(\Phi) - \hat{R}^{'}(\Phi) + \hat{R}^{'}(\Phi) &\leq \epsilon - \kappa \\ \hat{R}^{'}(\Phi) &\leq \epsilon - \kappa + |R^{'}(\Phi) - \hat{R}^{'}(\Phi)| \end{aligned} \tag{44}$$

From the definition of $\frac{\kappa}{2}$-representativeness it follows that $|R^{'}(\Phi) - \hat{R}^{'}(\Phi)| \leq \frac{\kappa}{2}$ and substituting this in equation 44 we get

$$\hat{R}^{'}(\Phi) \leq \epsilon - \kappa/2 \implies \hat{R}^{'}(\Phi) \leq \epsilon \implies \mathcal{S}^{\mathsf{IV}}(\epsilon) \subseteq \hat{\mathcal{S}}^{\mathsf{IV}}(\epsilon) \tag{45}$$

Next we show $\hat{\mathcal{S}}^{\mathsf{IV}}(\epsilon) \subseteq \mathcal{S}^{\mathsf{IV}}(\epsilon)$.

Consider $\Phi \in \hat{\mathcal{S}}^{\mathsf{IV}}(\epsilon)$

$$\hat{R}^{'}(\Phi) \leq \epsilon$$
$$\hat{R}^{'}(\Phi) - R^{'}(\Phi) + R^{'}(\Phi^{'}) \leq \epsilon \qquad (46)$$
$$R^{'}(\Phi) \leq \epsilon + |\hat{R}^{'}(\Phi) - R^{'}(\Phi)|$$

From the definition of $\frac{\kappa}{2}$-representativeness it follows that $|R^{'}(\Phi) - \hat{R}^{'}(\Phi)| \leq \frac{\kappa}{2}$ and substituting this in equation 46 we get

$$R^{'}(\Phi) \leq \epsilon + \frac{\kappa}{2} \qquad (47)$$

From equation 40, it follows that $R^{'}(\Phi) \leq \epsilon + \frac{\kappa}{2} \implies R^{'}(\Phi) \leq \epsilon$. Therefore, $\Phi \in \mathcal{S}^{\mathsf{IV}}(\epsilon)$. This proves the second part $\hat{\mathcal{S}}^{\mathsf{IV}}(\epsilon) \subseteq \mathcal{S}^{\mathsf{IV}}(\epsilon)$ and completes the proof.

$\square$

**Lemma 3.** *If $\kappa > 0$, $D$ is $\frac{\nu}{2}$-representative w.r.t $\mathcal{X}$, $\mathcal{H}_\Phi$, loss $\ell$ and distribution $\bar{\mathbb{P}}$ (joint distribution over $(e, X^e, Y^e)$ defined in Section 3.1) and and $\tilde{D}$ is $\frac{\kappa}{2}$-representative w.r.t $\mathcal{X}$, $\mathcal{H}_\Phi$, loss $\ell^{'}$ and distribution $\tilde{\mathbb{P}}$, then every solution $\hat{\Phi}$ to EIRM (equation 6) satisfies $\hat{\Phi}$ is in $\mathcal{S}^{\mathsf{IV}}(\epsilon)$ and $R(\Phi^*) \leq R(\hat{\Phi}) \leq R(\Phi^*) + \epsilon$, where $\Phi^*$ is the solution of IRM in equation 5.*

*Proof.* Given the condition in the above lemma, we are able to use the previous Lemma 2 to deduce that $\hat{\mathcal{S}}^{\mathsf{IV}}(\epsilon) = \mathcal{S}^{\mathsf{IV}}(\epsilon)$. This makes the set of predictors satisfying the constraints in EIRM equation 6 and IRM equation 5 the same.

$\Phi^*$ solves equation 5 and $\hat{\Phi}$ solves equation 6. From $\frac{\nu}{2}$-representativeness we know that $R(\hat{\Phi}) - \frac{\nu}{2} \leq \hat{R}(\hat{\Phi})$. From the optimality of $\hat{\Phi}$ we know that $\hat{R}(\hat{\Phi}) \leq \hat{R}(\Phi^*)$ ($\Phi^* \in \hat{\mathcal{S}}^{\mathsf{IV}}(\epsilon) = \mathcal{S}^{\mathsf{IV}}(\epsilon)$). Moreover, from $\frac{\nu}{2}$-representativeness we know that $\hat{R}(\Phi^*) \leq R(\Phi^*) + \frac{\nu}{2}$. We combine these conditions as follows.

$$R(\hat{\Phi}) - \frac{\nu}{2} \leq \hat{R}(\hat{\Phi}) \leq \hat{R}(\Phi^*) \leq R(\Phi^*) + \frac{\nu}{2} \qquad (48)$$

Comparing the first and third inequality in the above equations we get $R(\hat{\Phi}) \leq R(\Phi^*) + \nu$. From the optimality of $\Phi^*$ over the set $\mathcal{S}^{\mathsf{IV}}(\epsilon)$ and since $\hat{\Phi} \in \mathcal{S}^{\mathsf{IV}}(\epsilon)$ it follows that $R(\Phi^*) \leq R(\hat{\Phi})$. Hence, $R(\Phi^*) \leq R(\hat{\Phi}) \leq R(\Phi^*) + \nu$. This completes the proof.

$\square$

Next, we prove Proposition 2 from the main body of the manuscript.

**Proposition 8.** *For every $\nu > 0, \epsilon > 0$ and $\delta \in (0, 1)$, if $\mathcal{H}_\Phi$ is a finite hypothesis class, Assumption 3 holds, $\kappa > 0$, and if the number of samples $|D|$ is greater than $\max\left\{\frac{16L^{'4}}{\kappa^2}, \frac{8L^2}{\nu^2}\right\} \log\left(\frac{4|\mathcal{H}_\Phi|}{\delta}\right)$, then with a probability at least $1 - \delta$, every solution $\hat{\Phi}$ of EIRM (equation 6) is a $\nu$ approximation of IRM, i.e. $\hat{\Phi} \in \mathcal{S}^{\mathsf{IV}}(\epsilon)$, $R(\Phi^*) \leq R(\hat{\Phi}) \leq R(\Phi^*) + \nu$, where $\Phi^*$ is a solution to IRM (equation 5).*

*Proof.* From Lemma 3, we know that if $D$ is $\frac{\nu}{2}$-representative w.r.t $\mathcal{X}$, $\mathcal{H}_\Phi$, loss $\ell$ and distribution $\bar{\mathbb{P}}$ and if $\tilde{D}$ is $\frac{\kappa}{2}$-representative $\mathcal{X}$, $\mathcal{H}_\Phi$, loss $\ell^{'}$ and distribution $\tilde{\mathbb{P}}$, then the the claim in the above Proposition is true.

Define an event $A$: $D$ is $\frac{\nu}{2}$-representative w.r.t $\mathcal{X}$, $\mathcal{H}_\Phi$, loss $\ell$ and distribution $\bar{\mathbb{P}}$.

Define an event $B$: $\tilde{D}$ is $\frac{\kappa}{2}$-representative $\mathcal{X}$, $\mathcal{H}_\Phi$, loss $\ell^{'}$ and distribution $\tilde{\mathbb{P}}$.

Define success as $A \cap B$. Next, we show that if $|D|$ is greater than $\max\left\{\frac{16L^{'4}}{\kappa^2}, \frac{8L^2}{\nu^2}\right\} \log\left(\frac{4|\mathcal{H}_\Phi|}{\delta}\right)$, then $\mathbb{P}(A \cap B)$ occurs with a probability at least $1 - \delta$. $P(A \cap B) = 1 - P(A^c \cup B^c) \geq 1 - P(A^c) -$

$P(B^c)$. If we bound $P(A^c) \leq \frac{\delta}{2}$ and $P(B^c) \leq \frac{\delta}{2}$, then we know the probability of success is at least $1 - \delta$.

We write

$$\mathbb{P}(A) = \mathbb{P}\Big(\Big\{D : \forall h \in \mathcal{H}_\Phi, |\hat{R}(h) - R(h)| \leq \frac{\nu}{2}\Big\}\Big) = 1 - \mathbb{P}\Big(\Big\{D : \exists h \in \mathcal{H}_\Phi, |\hat{R}(h) - R(h)| > \frac{\nu}{2}\Big\}\Big)$$

$$\begin{aligned}
\mathbb{P}\Big(\Big\{D : \exists h \in \mathcal{H}_\Phi, |\hat{R}(h) - R(h)| > \frac{\nu}{2}\Big\}\Big) &= \mathbb{P}\Big(\bigcup_{h \in \mathcal{H}_\Phi} \Big\{D : |\hat{R}(h) - R(h)| > \frac{\nu}{2}\Big\}\Big) \\
&\leq \sum_{h \in \mathcal{H}_\Phi} \mathbb{P}\Big(\Big\{D : |\hat{R}(h) - R(h)| > \frac{\nu}{2}\Big\}\Big)
\end{aligned} \tag{49}$$

The loss function is bounded $|\ell(\Phi(\cdot), \cdot)| \leq L$. From Hoeffding's inequality in Lemma 1 it follows that

$$\mathbb{P}\Big(\Big\{D : |\hat{R}(h) - R(h)| > \frac{\nu}{2}\Big\}\Big) \leq 2\exp\Big(-\frac{|D|\nu^2}{8L^2}\Big) \tag{50}$$

Using this expression equation 50 in equation 49, we get

$$2|\mathcal{H}_\Phi| \exp\Big(-\frac{|D|\nu^2}{8L^2}\Big) \leq \frac{\delta}{2} \implies |D| \geq \frac{8L^2}{\nu^2} \log\Big(\frac{4|\mathcal{H}_\Phi|}{\delta}\Big) \tag{51}$$

$$\mathbb{P}(B) = \mathbb{P}\Big(\Big\{\tilde{D} : \forall h \in \mathcal{H}_\Phi, |\hat{R}'(h) - R'(h)| \leq \frac{\kappa}{2}\Big\}\Big) = 1 - \mathbb{P}\Big(\Big\{\tilde{D} : \exists h \in \mathcal{H}_\Phi, |\hat{R}'(h) - R'(h)| > \frac{\kappa}{2}\Big\}\Big)$$

$$\begin{aligned}
\mathbb{P}\Big(\Big\{\tilde{D} : \exists h \in \mathcal{H}_\Phi, |\hat{R}'(h) - R'(h)| > \frac{\kappa}{2}\Big\}\Big) &= \mathbb{P}\Big(\bigcup_{h \in \mathcal{H}_\Phi} \Big\{\tilde{D} : |\hat{R}'(h) - R'(h)| > \frac{\kappa}{2}\Big\}\Big) \\
&\leq \sum_{h \in \mathcal{H}_\Phi} \mathbb{P}\Big(\Big\{\tilde{D} : |\hat{R}'(h) - R'(h)| > \frac{\kappa}{2}\Big\}\Big)
\end{aligned} \tag{52}$$

The gradient of loss function is bounded $|\frac{\partial \ell(h(\cdot), \cdot)}{\partial w}|_{w=1.0}| \leq L'$. From the definition of $\ell'(h(\cdot), \cdot)$ in equation 34, we can infer that $|\ell'(h(\cdot), \cdot)| \leq L'^2$. Recall that $R'(h) = \tilde{\mathbb{E}}\Big[\ell'\Big(h, \big((X^e, Y^e), (\tilde{X}^e, \tilde{Y}^e)\big)\Big)\Big]$

From Hoeffding's inequality in Lemma 1 it follows that

$$\mathbb{P}\Big(\Big\{\tilde{D} : |\hat{R}'(h) - R'(h)| > \frac{\kappa}{2}\Big\}\Big) \leq 2\exp\Big(-\frac{|\tilde{D}|\kappa^2}{8L'^4}\Big) = 2\exp\Big(-\frac{|D|\kappa^2}{16L'^4}\Big) \tag{53}$$

Using the above equation 53 in equation 52 we get

$$2|\mathcal{H}_\Phi| \exp\Big(-\frac{|D|\kappa^2}{16L'^4}\Big) \leq \frac{\delta}{2} \implies |D| \geq \frac{16L'^4}{\kappa^2} \log\Big(\frac{4|\mathcal{H}_\Phi|}{\delta}\Big) \tag{54}$$

Combining the two conditions in equation 51 and equation 51 we get that if

$$|D| \geq \max\Big\{\frac{16L'^4}{\kappa^2}, \frac{8L^2}{\nu^2}\Big\} \log\Big(\frac{4|\mathcal{H}_\Phi|}{\delta}\Big)$$

then with probability at least $1 - \delta$ event $A \cap B$ occurs.

$\square$

### 7.3.3 PROPERTY OF LEAST SQUARES OPTIMAL SOLUTIONS

We first remind ourselves of a simple property of least squares minimization. Consider the least squares minimization setting, where $R(h) = \mathbb{E}[(Y - h(X))^2]$.

$$
\begin{aligned}
\mathbb{E}[(Y - h(X))^2] &= \mathbb{E}\Big[\big(Y - E[Y|X] + \mathbb{E}[Y|X] - h(X)\big)^2\Big] \\
&= \mathbb{E}_X\Big[\mathbb{E}\big[(Y - E[Y|X])^2|X\big]\Big] + \mathbb{E}_X\Big[\big(\mathbb{E}[Y|X] - h(X)\big)^2\Big] \\
&= \mathbb{E}_X\Big[\mathsf{Var}\big[Y|X\big]\Big] + \mathbb{E}_X\Big[\big(\mathbb{E}[Y|X] - h(X)\big)^2\Big]
\end{aligned}
\tag{55}
$$

In the above simplification, we use the law of total expectation. Both the terms in the above equations are always greater than or equal to zero. The first term does not depend on $h$, which implies the minimization can focus on second term only.

$$
\min_h R(h) = \mathbb{E}_X\Big[\mathsf{Var}\big[Y|X\big]\Big] + \min_h \mathbb{E}_X\Big[\big(\mathbb{E}[Y|X] - h(X)\big)^2\Big]
\tag{56}
$$

Assume that $\mathbb{P}$ has full support over $\mathcal{X}$. Define $\forall x \in \mathcal{X}, h^*(x) = \mathbb{E}[Y|X = x]$. $\forall h, R(h) \geq \mathbb{E}_X\Big[\mathsf{Var}\big[Y|X\big]\Big]$. Since $R(h^*) = \mathbb{E}_X\Big[\mathsf{Var}\big[Y|X\big]\Big]$. Therefore,

$$
h^* \in \arg\min_h \mathbb{E}_X\Big[\big(\mathbb{E}[Y|X] - h(X)\big)^2\Big]
\tag{57}
$$

Moreover, we conclude that $h^*$ is the unique minimizer. Observe that $\mathbb{E}_X\Big[\big(\mathbb{E}[Y|X] - h(X)\big)^2\Big]$ is zero for $h = h^*$. From Theorem 1.6.6 in (Ash et al., 2000), it follows that any other minimizer is same as $h^*$ except over a set of measure zero.

Recall the definition of $m(x) = \mathbb{E}[Y^e|X^e = x]$

**Lemma 4.** *Let $\ell$ be the square loss. If Assumption 4 holds and $m \in \mathcal{H}_\Phi$, then $m$ uniquely solves expected risk minimization $m \in \arg\min_{\Phi \in \mathcal{H}_\Phi} R(\Phi)$ and also uniquely solves IRM (equation 5).*

*Proof.* $R(\Phi) = \sum_{e \in \mathcal{E}_{tr}} \pi^e R^e(\Phi)$. From Assumption 4 and the observation in equation 57, it follows that the unique optimal solution to expected risk minimization for each $R^e$ is $m$. Therefore, $m$ also minimizes the weighted combination $R$.

To show the latter part of the Lemma, if we can show that $m \in \mathcal{S}^{\mathsf{IV}}(\epsilon)$, then the rest of the proof follows from the previous part as we already showed $m$ is a minimizer among all the functions in $\mathcal{H}_\Phi$ and $\mathcal{S}^{\mathsf{IV}}(\epsilon) \subseteq \mathcal{H}_\Phi$.

Suppose $m \notin \mathcal{S}^{\mathsf{IV}}(\epsilon)$. This implies there exists at least one environment for which $\|\nabla_{w|w=1.0} R^e(w \cdot m)\|^2 > 0 \implies \nabla_{w|w=1.0} R^e(w \cdot m) \neq 0$. As a result $\exists w$ in the neighborhood of $w = 1.0$ where $R^e(w \cdot m) < R^e(m)$ (if such a point does not exist and all the points in the neigbohood of $w = 1.0$ are greater than or equal to $R^e(m)$ that would make $\nabla_{w|w=1.0} R^e(w \cdot m) = 0$, which would be a contradiction). Therefore, $R^e(w \cdot m) < R^e(m)$. However, this is a contradiction as we know that $m$ is the unique optimizer for each environment. Hence, $m \notin \mathcal{S}^{\mathsf{IV}}(\epsilon)$ cannot be true and thus $m \in \mathcal{S}^{\mathsf{IV}}(\epsilon)$. This completes the proof. $\square$

Next, we prove Proposition 4 from the main body of the manuscript.

**Proposition 9.** *Let $\ell$ be the square loss. For every $\nu > 0, \epsilon > 0$ and $\delta \in (0, 1)$, if $\mathcal{H}_\Phi$ is a finite hypothesis class, $m \in \mathcal{H}_\Phi$, Assumptions 3, 4 hold, and*

- *if the number of samples $|D|$ is greater than $\max\big\{\frac{8L^2}{\nu^2} \log(\frac{4|\mathcal{H}_\Phi|}{\delta}), \frac{16L'^4}{\epsilon^2} \log(\frac{2}{\delta})\big\}$, then with a probability at least $1 - \delta$, every solution $\hat{\Phi}$ to EIRM (equation 6) satisfies $R(m) \leq R(\hat{\Phi}) \leq R(m) + \nu$. If also $\nu < \tilde{\kappa}$, then $\Phi^\dagger = m$.*

- *if the number of samples $|D|$ is greater than $\frac{8L^2}{\nu^2} \log(\frac{2|\mathcal{H}_\Phi|}{\delta})$, then with a probability at least $1 - \delta$, every solution $\Phi^\dagger$ to ERM satisfies $R(m) \leq R(\Phi^\dagger) \leq R(m) + \nu$. If also $\nu < \tilde{\kappa}$, then $\Phi^\dagger = m$.*

*Proof.* We first cover the second part of the Proposition. From Proposition 3, we know that the output of ERM will satisfy $R(\Phi^+) \leq R(\Phi^\dagger) \leq R(\Phi^+) + \nu$. In this case from Lemma 4, it follows that $\Phi^+ = m$. From the definition of $\tilde{\kappa}$ and the fact that $\nu < \tilde{\kappa}$ implies that $\Phi^\dagger = m$. We now move to the first part of the Proposition.

For EIRM we will derive a tighter bound on sample complexity than the one in Proposition 2 since we can now use the Assumption 4. Observe that $\forall e \in \mathcal{E}_{tr}$, $\nabla_{w|w=1.0} R^e(w \cdot m) = 0$ (see the proof of Lemma 4). Therefore, $R^{'}(m) = 0$.

Define an event $A$: $D$ is $\frac{\nu}{2}$-representative w.r.t $\mathcal{X}$, $\mathcal{H}_\Phi$, loss $\ell$ and distribution $\bar{\mathbb{P}}$.

Define an event $B$: $\tilde{D}$ is such that $|\hat{R}^{'}(m) - R^{'}(m)| \leq \frac{\epsilon}{2}$. Since $R^{'}(m) = 0$, $|\hat{R}^{'}(m) - R^{'}(m)| \leq \frac{\epsilon}{2} \implies |\hat{R}^{'}(m)| \leq \frac{\epsilon}{2} \implies \hat{R}^{'}(m) \leq \frac{\epsilon}{2}$. Therefore, $m \in \hat{\mathcal{S}}^{\mathsf{IV}}(\epsilon)$.

If $A \cap B$ occurs, then $R(m) \leq R(\hat{\Phi}) \leq R(m) + \nu$; we justify claim next. Suppose $\hat{\Phi}$ solves equation 6. If event $A$ occurs, then from $\frac{\nu}{2}$-representative condition we know that $R(\hat{\Phi}) - \frac{\nu}{2} \leq \hat{R}(\hat{\Phi})$. From optimality of $\hat{\Phi}$ it follows that $\hat{R}(\hat{\Phi}) \leq \hat{R}(m)$ (event $B \implies m \in \hat{\mathcal{S}}^{\mathsf{IV}}(\epsilon)$). Moreover, from $\frac{\nu}{2}$-representative property, we conclude that $\hat{R}(m) \leq R(m) + \frac{\nu}{2}$. We combine these conditions as follows.

$$R(\hat{\Phi}) - \frac{\nu}{2} \leq \hat{R}(\hat{\Phi}) \leq \hat{R}(m) \leq R(m) + \frac{\nu}{2} \tag{58}$$

From the above we have $R(m) \leq R(\hat{\Phi}) \leq R(m) + \nu$. Recall the definition of $\tilde{\kappa}$ and since $\nu < \tilde{\kappa} \implies \hat{\Phi} = m$.

Next, we bound the probability of success. $P(A \cap B) = 1 - P(A^c \cup B^c) \geq 1 - P(A^c) - P(B^c)$. If we can bound $P(A^c) \leq \frac{\delta}{2}$ and $P(B^c) \leq \frac{\delta}{2}$, then we know the probability of success is at least $1 - \delta$.

We write

$$\mathbb{P}(A) = \mathbb{P}\Big(\Big\{D : \forall h \in \mathcal{H}_\Phi, |\hat{R}(h) - R(h)| \leq \frac{\nu}{2}\Big\}\Big) = 1 - \mathbb{P}\Big(\Big\{D : \exists h \in \mathcal{H}_\Phi, |\hat{R}(h) - R(h)| > \frac{\nu}{2}\Big\}\Big)$$

From equation 51 if the condition

$$|D| \geq \frac{8L^2}{\nu^2} \log\Big(\frac{4|\mathcal{H}_\Phi|}{\delta}\Big) \tag{59}$$

is true, then is true, then event $A^c$ occurs with probability at most $\frac{\delta}{2}$.

We write $\mathbb{P}(B) = \mathbb{P}\Big(\Big\{\tilde{D} :, |\hat{R}^{'}(m) - R^{'}(m)| \leq \frac{\epsilon}{2}\Big\}\Big) = 1 - \mathbb{P}\Big(\Big\{\tilde{D} : |\hat{R}^{'}(m) - R^{'}(m)| > \frac{\epsilon}{2}\Big\}\Big)$. The gradient of loss function is bounded $|\frac{\partial \ell(\Phi(\cdot), \cdot)}{\partial w}|_{w=1.0}| \leq L^{'}$. From Hoeffding's inequality in Lemma 1 it follows that

$$\mathbb{P}\Big(\Big\{\tilde{D} : |\hat{R}^{'}(h) - R^{'}(h)| > \frac{\epsilon}{2}\Big\}\Big) \leq 2\exp(-\frac{|D|\epsilon^2}{16L^{'4}}) \tag{60}$$

We bound the above equation 60 by $\frac{\delta}{2}$ to get

$$2\exp(-\frac{|D|\epsilon^2}{16L^{'4}}) \leq \frac{\delta}{2} \implies |D| \geq \frac{16L^{'4}}{\epsilon^2} \log(\frac{4}{\delta}) \tag{61}$$

Combining the two conditions equation 59 and equation 61,

$$|D| \geq \max\Big\{\frac{8L^2}{\nu^2} \log(\frac{4|\mathcal{H}_\Phi|}{\delta}), \frac{16L^{'4}}{\epsilon^2} \log(\frac{4}{\delta})\Big\}$$

This ensures $P(A \cap B) \geq 1 - \delta$. This completes the proof.

$\square$

Before stating the proof of Proposition 5, we will prove an intermediate proposition. For clarity, we will restate the result (Theorem 9 from Arjovsky et al. (2019)) next.

**Proposition 10.** *(Theorem 9 Arjovsky et al. (2019)) If Assumptions 5 and 6 (with $r = 1$) hold and let $\Phi \in \mathbb{R}^{n \times 1}$ ($\Phi \neq 0$), then*

$$\Phi^\mathsf{T} \mathbb{E}^e [X^e X^{e,\mathsf{T}}] \Phi = \Phi^\mathsf{T} \mathbb{E}^e [X^e Y^e] \tag{62}$$

*holds for all $e \in \mathcal{E}_{tr}$ iff $\Phi = \tilde{S}^\mathsf{T} \gamma$.*

Next, we propose an $\epsilon$-approximation of Proposition 10.

Define $\epsilon_0 = \frac{\pi^{\min}}{|\mathcal{E}_{tr}|} (\omega \lambda_{\min})^2 (12 - 8\sqrt{2})$.

**Proposition 11.** *Let $\ell$ be the square loss. If Assumptions 5, 6 (with $r = 1$) and 7 hold , then for all $0 < \epsilon < \epsilon_0$, the solution $\Phi$*

$$R'(\Phi) \leq \epsilon \tag{63}$$

*satisfies $\Phi = \tilde{S}^\mathsf{T} \gamma(\alpha)$, where $\alpha \in \left[ \frac{1}{1 + \frac{1}{2\omega \lambda_{\min}} \sqrt{\frac{\epsilon |\mathcal{E}_{tr}|}{\pi^{\min}}}}, \frac{1}{1 - \frac{1}{2\omega \lambda_{\min}} \sqrt{\frac{\epsilon |\mathcal{E}_{tr}|}{\pi^{\min}}}} \right]$*

*Proof.* Let us start by simplifying $\nabla_{w|w=1.0} R^e(w \cdot \Phi)$, using square loss for $\ell$ and linear representation $\Phi \in \mathbb{R}^{n \times 1}$.

$$\begin{aligned}
\nabla_{w|w=1.0} R^e(w \cdot \Phi) &= \frac{\partial \mathbb{E}^e \left[ \left( Y^e - w \cdot \Phi^\mathsf{T} X^e \right)^2 \right]}{\partial w} \bigg|_{w=1.0} = 2\mathbb{E}^e \left[ (\Phi^\mathsf{T} X^e)^2 \right] - 2\mathbb{E}^e \left[ \Phi^\mathsf{T} X^e Y^e \right] \\
&= 2\Phi^\mathsf{T} \mathbb{E}^e \left[ X^e X^{e,\mathsf{T}} \right] \Phi - 2\Phi^\mathsf{T} \mathbb{E}^e \left[ Y^e X^e \right]
\end{aligned} \tag{64}$$

Plug the above equation 64 in the condition $R'(\Phi) \leq \epsilon$ to get

$$R'(\Phi) = \sum_e \pi^e \| \nabla_{w|w=1.0} R^e(w \cdot \Phi) \|^2 = 4 \sum_e \pi^e \left( \Phi^\mathsf{T} \mathbb{E}^e \left[ X^e X^{e,\mathsf{T}} \right] \Phi - \Phi^\mathsf{T} \mathbb{E}^e \left[ X^e Y^e \right] \right)^2 \leq \epsilon \tag{65}$$

From the bound on $\pi^e$ in Assumption 5 it follows that for each $e \in \mathcal{E}_{tr}$

$$\begin{aligned}
\left( \Phi^\mathsf{T} \mathbb{E}^e \left[ X^e X^{e,\mathsf{T}} \right] \Phi - \Phi^\mathsf{T} \mathbb{E}^e \left[ X^e Y^e \right] \right)^2 &\leq \frac{\epsilon |\mathcal{E}_{tr}|}{4\pi^{\min}} \\
\left| \Phi^\mathsf{T} \mathbb{E}^e \left[ X^e X^{e,\mathsf{T}} \right] \Phi - \Phi^\mathsf{T} \mathbb{E}^e \left[ X^e Y^e \right] \right| &\leq \sqrt{\frac{\epsilon |\mathcal{E}_{tr}|}{4\pi^{\min}}}
\end{aligned} \tag{66}$$

Note that if the condition above equation 66 is not true, then the preceeding condition in equation 65 cannot be true as contribution from one term in the summation itself will exceed $\epsilon$. In the above equation 66, we are using the positive square root since RHS has to be greater than or equal to zero.

We compute the second derivative of loss w.r.t $w$

$$\begin{aligned}
\nabla_w^2 R^e(w \cdot \Phi) &= \frac{\partial^2 \mathbb{E}^e \left[ \left( Y^e - \Phi^\mathsf{T} X^e w \right)^2 \right]}{\partial w^2} = \frac{\partial \left( 2w \mathbb{E}^e \left[ (\Phi^\mathsf{T} X^e)^2 \right] - 2\mathbb{E}^e \left[ \Phi^\mathsf{T} X^e Y^e \right] \right)}{\partial w} \\
&= 2\Phi^\mathsf{T} \mathbb{E}^e \left[ X^e X^{e,\mathsf{T}} \right] \Phi \\
&= 2\Phi^\mathsf{T} \Sigma^e \Phi
\end{aligned} \tag{67}$$

Since $\Sigma^e$ is symmetric, we can use the eigenvalue decomposition of $\Sigma^e = U\Lambda U^\mathsf{T}$ in equation 67 to get $\Phi^\mathsf{T} \Sigma^e \Phi = \Phi^\mathsf{T} U \Lambda U^\mathsf{T} \Phi$. Substitute $\tilde{\Phi} = U^\mathsf{T} \Phi$ to get $\Phi^\mathsf{T} \Sigma^e \Phi = \tilde{\Phi}^\mathsf{T} \Lambda \tilde{\Phi} \geq \lambda_{\min}(\Sigma_e) \| \tilde{\Phi} \|^2 = \lambda_{\min}(\Sigma_e) \Phi^\mathsf{T} U U^\mathsf{T} \Phi = \lambda_{\min}(\Sigma_e) \| \Phi \|^2$. From Assumption 5, we have $\lambda_{\min}(\Sigma_e) \geq \lambda_{\min}$ and from Assumption 7 we have $\| \Phi \|^2 \geq \omega$. Therefore, we can deduce that $2\Phi^\mathsf{T} \Sigma^e \Phi \geq 2\lambda_{\min} \omega$. Therefore, the second derivative defined in equation 67 is always greater than or equal to $2\lambda_{\min} \omega$.

$$\nabla_w^2 R^e(w \cdot \Phi) \geq 2\lambda_{\min} \omega > 0 \tag{68}$$

Let $\epsilon^{'} = 2\sqrt{\frac{\epsilon|\mathcal{E}_{tr}|}{4\pi^{\min}}} = \sqrt{\frac{\epsilon|\mathcal{E}_{tr}|}{\pi^{\min}}}$. We rewrite equation 66 in terms of $\nabla_{w|w=1.0}R^e(w \cdot \Phi)$ and $\epsilon^{'}$ to get

$$
\begin{aligned}
|\nabla_{w|w=1.0}R^e(w \cdot \Phi)| &\leq \epsilon^{'} \\
-\epsilon^{'} \leq \nabla_{w|w=1.0}R^e(w \cdot \Phi) &\leq \epsilon^{'}
\end{aligned}
\tag{69}
$$

Since the second derivative (equation 68) is strictly positive and larger than or equal to $\lambda_{\min}\omega \implies \exists \; w^e$ in the neighborhood of $w = 1.0$ at which $\nabla_{w|w=w^e}R^e(w \cdot \Phi) = 0$. This holds for all the environments in $\mathcal{E}_{tr}$. Define

$$
c(w) = \nabla_w R^e(w \cdot \Phi)
$$

Also, define $c^{'}(w) = \frac{\partial c(w)}{\partial w}$. Also, $c^{'}(w) = \nabla_w^2 R^e(w \cdot \Phi)$. Suppose $\nabla_{w|w=1.0}R^e(w \cdot \Phi) = c(1) < 0$. Since for all $w$, $c'(w) > 0$ (from equation 68), $\exists \; w^e > 1$, where $c(w^e) = 0$. Using fundamental theorem of calculus, we write

$$
c(w) - c(1) = \int_1^w c^{'}(u)du \geq 2\lambda_{\min}\omega(w - 1)
$$

Substituting $w = w^e$ in the above

$$
\begin{aligned}
c(w^e) - c(1) &\geq 2\lambda_{\min}\omega(w^e - 1) \\
-c(1) &\geq 2\lambda_{\min}\omega(w^e - 1)
\end{aligned}
$$

$$
w^e \leq 1 - \frac{c(1)}{2\lambda_{\min}\omega} \leq 1 + \frac{\epsilon^{'}}{2\lambda_{\min}\omega} = 1 + \frac{\sqrt{\frac{\epsilon|\mathcal{E}_{tr}|}{\pi^{\min}}}}{2\lambda_{\min}\omega}
\tag{70}
$$

Suppose $\nabla_{w,w=1.0}R^e(w.\Phi) = c(1) > 0$. Using fundamental theorem of calculus we can write

$$
c(1) - c(w) = \int_w^1 c^{'}(u)du \geq \lambda_{\min}\omega(1 - w)
$$

Substituting $w = w^e$ in the above

$$
\begin{aligned}
c(1) - c(w^e) &\geq 2\lambda_{\min}\omega(1 - w^e) \\
c(1) &\geq 2\lambda_{\min}\omega(1 - w^e)
\end{aligned}
$$

$$
w^e \geq 1 - \frac{c(1)}{2\lambda_{\min}\omega} \geq 1 - \frac{\epsilon^{'}}{2\lambda_{\min}\omega} = 1 - \frac{\sqrt{\frac{\epsilon|\mathcal{E}_{tr}|}{\pi^{\min}}}}{2\lambda_{\min}\omega}
\tag{71}
$$

Define $\eta_1 = \frac{\frac{1}{2\omega\lambda_{\min}}\sqrt{\frac{\epsilon|\mathcal{E}_{tr}|}{\pi^{\min}}}}{1 - \frac{1}{2\omega\lambda_{\min}}\sqrt{\frac{\epsilon|\mathcal{E}_{tr}|}{\pi^{\min}}}}$, $\eta_2 = \frac{-\frac{1}{2\omega\lambda_{\min}}\sqrt{\frac{\epsilon|\mathcal{E}_{tr}|}{\pi^{\min}}}}{1 + \frac{1}{2\omega\lambda_{\min}}\sqrt{\frac{\epsilon|\mathcal{E}_{tr}|}{\pi^{\min}}}}$. Combining equation 70 and equation 71 and using the definition of $\eta_1$ and $\eta_2$, we can conclude that $w^e \in [\frac{1}{1+\eta_1}, \frac{1}{1+\eta_2}]$. If we reparamterize $w^e = \frac{1}{1+\eta^e}$, then $\eta^e \in [\eta_2, \eta_1]$. We expand this condition $\nabla_{w|w=w^e}R^e(w \cdot \Phi) = 0$.

$$
\begin{aligned}
\frac{\partial}{\partial w}\mathbb{E}^e\big[(Y^e - \Phi^{\mathsf{T}}X^e w)^2\big] &= 2w\mathbb{E}^e\big[(\Phi^{\mathsf{T}}X^e)^2\big] - 2\mathbb{E}^e\big[\Phi^{\mathsf{T}}X^eY^e\big] \\
&= 2\Phi^{\mathsf{T}}\Big[\mathbb{E}^e\big[X^eX^{e,\mathsf{T}}\big]\Phi w^e - \mathbb{E}^e\big[X^eY^e\big]\Big] \\
&= 2\Phi^{\mathsf{T}}\Big[\mathbb{E}^e\big[X^eX^{e,\mathsf{T}}\big]\Phi w^e - \mathbb{E}^e\big[X^eX^{e,\mathsf{T}}\big]\tilde{S}^{\mathsf{T}}\gamma - \mathbb{E}^e\big[X^e\varepsilon^e\big]\Big] \\
&= 2\Phi^{\mathsf{T}}\Big[\mathbb{E}^e\big[X^eX^{e,\mathsf{T}}\big]\big(\Phi w^e - \tilde{S}^{\mathsf{T}}\gamma\big) - \mathbb{E}^e\big[X^e\varepsilon^e\big]\Big] \\
&= 2\Phi^{\mathsf{T}}\Big[\mathbb{E}^e\big[X^eX^{e,\mathsf{T}}\big]\big(\Phi w^e - \tilde{S}^{\mathsf{T}}\gamma\big) - \mathbb{E}^e\big[X^e\varepsilon^e\big]\Big] = 0 \\
&= 2\Phi^{\mathsf{T}}\Big[\mathbb{E}^e\big[X^eX^{e,\mathsf{T}}\big]\big(\Phi\frac{1}{1+\eta^e} - \tilde{S}^{\mathsf{T}}\gamma\big) - \mathbb{E}^e\big[X^e\varepsilon^e\big]\Big] = 0
\end{aligned}
\tag{72}
$$

Assume that $\Phi\frac{1}{1+\eta^e} \neq \tilde{S}^{\mathsf{T}}\gamma$, for all $e \in \mathcal{E}_{tr}$. If this assumption is not true and $\Phi\frac{1}{1+\eta^e} = \tilde{S}^{\mathsf{T}}\gamma$ for some $e \in \mathcal{E}_{tr}$, then it already establishes the claim we set out to prove in this Proposition (since $\eta^e \in [\eta_2, \eta_1]$).

Define $q^e = \left[ \mathbb{E}^e[X^e X^{e,\mathsf{T}}]\left( \Phi\frac{1}{1+\eta^e} - \tilde{S}^{\mathsf{T}}\gamma \right) - \mathbb{E}^e[X^e \varepsilon^e] \right]$. From the Assumption 6 and since $\Phi\frac{1}{1+\eta^e} \neq \tilde{S}^{\mathsf{T}}\gamma$, we know $\dim\left( \mathrm{span}\left\{ q^e \right\} \right) > n - 1$.

From rank-nullity theorem we know dimension of kernel space of $\Phi$ (rank of $\Phi$ is 1) is $n - 1$. From equation 72 it follows that $q^e$ is in kernel space of $\Phi$. Therefore, $\dim(\mathrm{Ker}(\Phi)) = n - 1 \implies \dim\left( \mathrm{span}\left\{ q^e \right\} \right) \leq n - 1$ which leads to a contradiction.

Therefore, $\Phi\frac{1}{1+\eta^e} = \tilde{S}^{\mathsf{T}}\gamma$ at least for one environment.

If $\Phi = \tilde{S}^{\mathsf{T}}\gamma(1 + \eta^e)$, then $\|\Phi\|^2 = \|\tilde{S}^{\mathsf{T}}\gamma\|^2(1 + \eta^e)^2 \geq \|\tilde{S}^{\mathsf{T}}\gamma\|^2(1 + \eta_2)^2 \geq \|\tilde{S}^{\mathsf{T}}\gamma\|^2\frac{1}{2} \geq \omega$. In this simplification, we use $(1 + \eta_2)^2 \geq \frac{1}{\left(1 + \frac{1}{2\omega\lambda_{\min}}\sqrt{\frac{\epsilon|\mathcal{E}_{tr}|}{\pi^{\min}}}\right)^2} \geq \frac{1}{\left(1 + \frac{1}{2\omega\lambda_{\min}}\sqrt{\frac{\epsilon_0|\mathcal{E}_{tr}|}{\pi^{\min}}}\right)^2} = \frac{1}{\left(1 + \sqrt{3 - 2\sqrt{2}}\right)^2} = \frac{1}{2}$ and $\|\tilde{S}^{\mathsf{T}}\gamma\|^2 \geq 2\omega$ (from Assumption 7). This ensures that for any solution of the form $\Phi\frac{1}{1+\eta^e}$ the assumption $\|\Phi\|^2 \geq \omega$ is automatically satisfied.

If $\Phi = \tilde{S}^{\mathsf{T}}\gamma(1 + \eta^e)$, then $\|\Phi\|^2 = \|\tilde{S}^{\mathsf{T}}\gamma\|^2(1 + \eta^e)^2 \leq \|\tilde{S}^{\mathsf{T}}\gamma\|^2(1 + \eta_1)^2 \leq \|\tilde{S}^{\mathsf{T}}\gamma\|^2\left(\frac{3 + 2\sqrt{2}}{2}\right) \leq \Omega$. In the above simplification, we use $(1 + \eta_1)^2 \leq \frac{1}{\left(1 - \frac{1}{2\omega\lambda_{\min}}\sqrt{\frac{\epsilon|\mathcal{E}_{tr}|}{\pi^{\min}}}\right)^2} \leq \frac{1}{\left(1 - \frac{1}{2\omega\lambda_{\min}}\sqrt{\frac{\epsilon_0|\mathcal{E}_{tr}|}{\pi^{\min}}}\right)^2} = \frac{1}{\left(1 - \sqrt{3 - 2\sqrt{2}}\right)^2} = \frac{3 + 2\sqrt{2}}{2}$ and $\|\tilde{S}^{\mathsf{T}}\gamma\|^2 \leq \frac{2}{3 + 2\sqrt{2}}\Omega$ (from Assumption 7). This ensures that for any solution of the form $\Phi\frac{1}{1+\eta^e}$ the assumption $\|\Phi\|^2 \leq \Omega$ is automatically satisfied.

The entire proof so far has characterized the property of $\Phi$ that satisfies $R'(\Phi) \leq \epsilon$. But how do we know such a $\Phi$ exists. Recall $\tilde{S}^t\gamma \in \mathcal{H}_\Phi$. For each environment $e \in \mathcal{E}_{tr}$

$$
\begin{aligned}
\nabla_{w|w=1.0}R^e(w \cdot \Phi) &= \Phi^{\mathsf{T}}\mathbb{E}^e[X^e X^{e,\mathsf{T}}]\Phi - \Phi^{\mathsf{T}}\mathbb{E}^e[X^e Y^e] \\
&= \gamma^{\mathsf{T}}\tilde{S}\mathbb{E}^e[X^e X^{e,\mathsf{T}}]\tilde{S}^{\mathsf{T}}\gamma - \gamma^{\mathsf{T}}\tilde{S}\mathbb{E}^e[X^e Y^e] \\
&= \gamma^{\mathsf{T}}\mathbb{E}^e[Z_1^e Z_1^{e,\mathsf{T}}]\gamma - \gamma^{\mathsf{T}}\mathbb{E}^e[Z_1^e Y^e] \\
&= \gamma^{\mathsf{T}}\mathbb{E}^e[Z_1^e Z_1^{e,\mathsf{T}}]\gamma - \gamma^{\mathsf{T}}\mathbb{E}^e[Z_1^e Z_1^{e,\mathsf{T}}]\gamma - \gamma^{\mathsf{T}}\mathbb{E}^e[Z_1^e \varepsilon^e] \\
&= 0
\end{aligned}
\tag{73}
$$

We use Assumption 5 in the simplification above in equation 73 Therefore, $R'(\tilde{S}^t\gamma) = 0$ and as a result the existence of a $\Phi$ that satisfies $R'(\Phi) \leq \epsilon$ is guaranteed. This completes the proof.

$\square$

Before proving Proposition 5, we first establish that Assumptions 5 and 7 are sufficient to ensure that Assumption 3 holds and we can thus use bounds $L$ and $L'$ defined in Assumption 3.

**Proving Assumption 3 for square loss $\ell$ from Assumptions 5, 7.** From Assumption 5, we have

$$
\begin{aligned}
Z^e &= (Z_1^e, Z_2^e) \\
X^e &= SZ^e \\
\|X^e\| &\leq \|S\|\|Z^e\|
\end{aligned}
\tag{74}
$$

From Assumption 5, $\|S\|$ is bounded and $\|Z^e\|$ is bounded $\implies \|X^e\|$ is bounded as well (from equation 74). Therefore, there exists $X^{\mathsf{sup}} < \infty$, s.t. $\|X^e\| \leq X^{\mathsf{sup}}$.

$$Y^e = (\tilde{S}^{\mathsf{T}}\gamma)^{\mathsf{T}}X^e + \varepsilon^e$$
$$|Y^e| \le \|\tilde{S}^{\mathsf{T}}\gamma\|\|X^e\| + |\varepsilon^e| \implies \tag{75}$$
$$|Y^e| \le \sqrt{\frac{2}{3+2\sqrt{2}}}\Omega X^{\mathsf{sup}} + \varepsilon^{\mathsf{sup}}$$

In the last step of equation 75, we use $\|\tilde{S}^{\mathsf{T}}\gamma\|^2 \le \frac{2}{3+2\sqrt{2}}\Omega$ (Assumption 7), $\|X^e\| \le X^{\mathsf{sup}}$ derived above (equation 74), and $|\varepsilon^e| \le \varepsilon^{\mathsf{sup}}$ (Assumption 5). Therefore, $Y^e$ is bounded and there exists a $K$ such that $|Y^e| \le K \le \sqrt{\Omega}X^{\mathsf{sup}} + \varepsilon^{\mathsf{sup}}$. Therefore, $\forall \Phi \in \mathcal{H}_\Phi$ and for all $X^e, Y^e$ sampled from the model in Assumption 5 we have

$$\ell(\Phi(X^e), Y^e) = (Y^e - \Phi^{\mathsf{T}}X^e)^2 \le (K + \sqrt{\Omega}X^{\mathsf{sup}})^2 \tag{76}$$

$$\left|\frac{\partial \ell(w \cdot \Phi^{\mathsf{T}}X, Y)}{\partial w}\Big|_{w=1.0}\right| = \left|\Phi^{\mathsf{T}}X(\Phi^{\mathsf{T}}X - Y)\right| \le (\sqrt{\Omega}X^{\mathsf{sup}})(\sqrt{\Omega}X^{\mathsf{sup}} + K) \tag{77}$$

From equation 76, we conclude that $\ell(\Phi(\cdot), \cdot)$ is bounded and there exists an $L$ such that $|\ell(\Phi(\cdot), \cdot)| \le L \le (K + \sqrt{\Omega}X^{\mathsf{sup}})^2$. From equation 77, we conclude that $\frac{\partial \ell(w \cdot \Phi^{\mathsf{T}}X, Y)}{\partial w}\Big|_{w=1.0}$ is bounded and there exists an $L'$ such that $\left|\frac{\ell(w \cdot \Phi(\cdot), \cdot)}{\partial w}\Big|_{w=1.0}\right| \le L' \le (\sqrt{\Omega}X^{\mathsf{sup}})(\sqrt{\Omega}X^{\mathsf{sup}} + K)$.

Define $\epsilon_{\mathsf{th}} = \frac{24-16\sqrt{2}}{3}\frac{\pi^{\mathsf{min}}}{|\mathcal{E}_{tr}|}(\omega\lambda_{\mathsf{min}})^2$ and $\tau = \frac{1}{2\omega\lambda_{\mathsf{min}}}\sqrt{\frac{3|\mathcal{E}_{tr}|}{2\pi^{\mathsf{min}}}}$

Next, we prove Proposition 5 from the main body of the manuscript.

**Proposition 12.** *Let $\ell$ be the square loss. For every $\epsilon \in (0, \epsilon_{\mathsf{th}})$ and $\delta \in (0, 1)$, if Assumptions 5, 6 (with $r = 1$), 7 hold and if the number of data points $|D|$ is greater than $\frac{16L'^4}{\epsilon^2}\log\left(\frac{2|\mathcal{H}_\Phi|}{\delta}\right)$, then with a probability at least $1 - \delta$, every solution $\hat{\Phi}$ to EIRM (equation 6) satisfies $\hat{\Phi} = (\tilde{S}^{\mathsf{T}}\gamma)\alpha$, where $\alpha \in [\frac{1}{1+\tau\sqrt{\epsilon}}, \frac{1}{1-\tau\sqrt{\epsilon}}]$.*

*Proof.* Define an event $A$: $\{\tilde{D} : \forall \Phi \in \mathcal{H}_\Phi, |\hat{R}'(\Phi) - R'(\Phi)| \le \frac{\epsilon}{2}\}$. If event $A$ happens, then

$$\hat{R}'(\Phi) \le \epsilon$$
$$\hat{R}'(\Phi) - R'(\Phi) + R'(\Phi) \le \epsilon$$
$$R'(\Phi) \le \epsilon + |\hat{R}'(\Phi) - R'(\Phi)| \tag{78}$$
$$R'(\Phi) \le \frac{3\epsilon}{2}$$

If event $A$ happens, then every solution $\hat{\Phi}$ to EIRM (equation 6) satisfies equation 78. From equation 78, we see that we should substitute $\epsilon$ with $\frac{3\epsilon}{2}$ and use Proposition 11. If the Assumptions 5, 6 (with $r = 1$), 7 hold and event $A$ happens, then for all $0 < \frac{3}{2}\epsilon < \epsilon_0$ the output of EIRM equation 6 $\hat{\Phi} = \tilde{S}^t(\alpha)$, where $\alpha \in [\frac{1}{1+\frac{1}{2\omega\lambda_{\mathsf{min}}}\sqrt{\frac{3\epsilon|\mathcal{E}_{tr}|}{2\pi^{\mathsf{min}}}}}, \frac{1}{1-\frac{1}{2\omega\lambda_{\mathsf{min}}}\sqrt{\frac{3\epsilon|\mathcal{E}_{tr}|}{2\pi^{\mathsf{min}}}}}] = [\frac{1}{1+\tau\sqrt{\epsilon}}, \frac{1}{1-\tau\sqrt{\epsilon}}]$. Note that $\frac{3}{2}\epsilon < \epsilon_0 \implies \epsilon < \epsilon_{\mathsf{th}}$. Hence, all that remains to be shown is that event $A$ occurs with a probability at least $1 - \delta$. Next, we show that if $|D| \ge \frac{16L'^4}{\epsilon^2}\log(\frac{2|\mathcal{H}_\Phi|}{\delta})$, then with probability $1 - \delta$ event $A$ happens.

Now we need to bound the probability $\mathbb{P}(A)$. We will find an upper bound on the failure probability using Hoeffding's inequality (Lemma 1) and the bound $L'$ (derived in equation 77) as follows. We redo the same analysis as was done in equation 54 for reader's convenience.

$$\mathbb{P}(A) = 1 - \mathbb{P}\Big(\Big\{\tilde{D} : \exists \Phi \in \mathcal{H}_\Phi, |\hat{R}'(\Phi) - R'(\Phi)| > \frac{\epsilon}{2}\Big\}\Big) = \mathbb{P}\Big(\bigcup_{\Phi \in \mathcal{H}_\Phi} \Big\{\tilde{D} : |\hat{R}'(\Phi) - R'(\Phi)| > \frac{\epsilon}{2}\Big\}\Big)$$

$$\mathbb{P}\Big(\bigcup_{\Phi \in \mathcal{H}_\Phi} \Big\{\tilde{D} : |\hat{R}'(\Phi) - R'(\Phi)| > \frac{\epsilon}{2}\Big\}\Big) \leq \sum_{\Phi \in \mathcal{H}_\Phi} \mathbb{P}\Big(\{\tilde{D} : |\hat{R}'(\Phi) - R'(\Phi)| > \frac{\epsilon}{2}\}\Big) \leq 2|\mathcal{H}_\Phi| e^{-\frac{\epsilon^2 |D|}{16 L'^4}}$$

$$2|\mathcal{H}_\Phi| e^{-\frac{\epsilon^2 |D|}{16 L'^4}} \leq \delta \implies \mathbb{P}(A^c) \leq \delta$$

$$|D| \geq \frac{16 L'^4}{\epsilon^2} \log\Big(\frac{2|\mathcal{H}_\Phi|}{\delta}\Big) \implies \mathbb{P}(A^c) \leq \delta$$

$$(79)$$

Hence, we know that if $|D| \geq \frac{16 L'^4}{\epsilon^2} \log\Big(\frac{2|\mathcal{H}_\Phi|}{\delta}\Big)$, then with probability $1 - \delta$ event $A$ happens.

The proof characterized the property of $\hat{\Phi}$ that satisfies $\hat{R}'(\hat{\Phi}) \leq \epsilon$. But how do we know such a $\hat{\Phi}$ exists; we show that a $\hat{\Phi}$ always exists. Consider a $\Phi$ that satisfiess the following.

$$\begin{aligned} R'(\Phi) &\leq \frac{\epsilon}{2} \\ \hat{R}'(\Phi) - \hat{R}'(\Phi) + R'(\Phi) &\leq \frac{\epsilon}{2} \\ \hat{R}'(\Phi) &\leq \frac{\epsilon}{2} + |\hat{R}'(\Phi) - R'(\Phi)| \\ \hat{R}'(\Phi) &\leq \epsilon \text{ (follows from event } A) \end{aligned}$$

$$(80)$$

From equation 73 in the proof of Proposition 11, we know that $R'(\tilde{\mathcal{S}}^\mathsf{T}\gamma) = 0$. From equation 80, $\tilde{\mathcal{S}}^\mathsf{T}\gamma \in \mathcal{H}_\Phi$ satisfies $\hat{R}'(\Phi) \leq \epsilon$.

$$\square$$

Since equation 6 is a constrained optimization, a penalty based version (IRMv1) was proposed by Arjovsky et al. (2019) that minimizes $\hat{R}(\Phi) + \lambda \hat{R}'(\Phi)$. Both Propositions 4 and Proposition 5 can be extended to IRMv1. Below we show the sample complexity analysis for IRMv1 applied to the setting assumed in Proposition 5. Sample complexity analysis of IRMv1 below shows that distance to the OOD solution decays as $\mathcal{O}(\sqrt{1/\lambda})$. Define $\lambda_{\mathsf{th}} = \max\{5\sigma^2/3\epsilon_{\mathsf{th}}, 1\}$

**Corollary 1.** *For every $\delta \in (0, 1)$, $\lambda > \lambda_{\mathsf{th}}$, if Assumptions 5, 6 (with $r = 1$), 7 hold, and $|D| \geq \max\big\{\frac{64 L'^4 \lambda^2}{25\sigma^4} \log\big(\frac{4|\mathcal{H}_\Phi|}{\delta}\big), \frac{32 L^2 \lambda^2}{25\sigma^4} \log(\frac{4}{\delta})\big\}$, then with a probability at least $1 - \delta$ every solution $\hat{\Phi}$ of IRMv1 satisfies $\hat{\Phi} = (\tilde{\mathcal{S}}^\mathsf{T}\gamma)\alpha$, where $\alpha \in \big[\frac{1}{1 + \tau\sigma\sqrt{\frac{5}{3\lambda}}}, \frac{1}{1 - \tau\sigma\sqrt{\frac{5}{3\lambda}}}\big]$.*

*Proof.* The empirical version of IRMv1 minimizes

$$\hat{R}(\Phi) + \lambda \hat{R}'(\Phi)$$

Let us compute the risk achieved by the ideal invariant predictor $\tilde{\mathcal{S}}^\mathsf{T}\gamma$.

$$R(\tilde{\mathcal{S}}^\mathsf{T}\gamma) = \sum_{e \in \mathcal{E}_{tr}} \pi^e \mathbb{E}^e\Big[\big(Y^e - \gamma^\mathsf{T}\tilde{\mathcal{S}} X^e\big)^2\Big] = \sum_{e \in \mathcal{E}_{tr}} \pi^e \mathbb{E}^e\Big[\big(Y^e - \gamma^\mathsf{T} Z_1^e\big)^2\Big] = \sum_{e \in \mathcal{E}_{tr}} \pi^e \mathbb{E}^e\big[(\varepsilon^e)^2\big] = \sigma^2$$

$$(81)$$

Define event $A$: $\Big\{D : |\hat{R}(\tilde{\mathcal{S}}^\mathsf{T}\gamma) - R(\tilde{\mathcal{S}}^\mathsf{T}\gamma)| \leq \frac{\epsilon}{2}\Big\}$.

If event $A$ holds, then

$$\hat{R}(\tilde{\mathcal{S}}^\mathsf{T}\gamma) \leq \sigma^2 + \frac{\epsilon}{2}$$

$$(82)$$

Define event $B$: $\Big\{\tilde{D} : \forall \Phi \in \mathcal{H}_\Phi, |\hat{R}'(\Phi) - R'(\Phi)| \leq \frac{\epsilon}{2}\Big\}$.

If event $B$ holds, then $|\hat{R}'\big(\tilde{\mathcal{S}}^{\mathsf{T}}\gamma\big) - R'\big(\tilde{\mathcal{S}}^{\mathsf{T}}\gamma\big)| \leq \frac{\epsilon}{2}$ and if we plug in $R'\big(\tilde{\mathcal{S}}^{\mathsf{T}}\gamma\big) = 0$ (from equation 73), then

$$\hat{R}'\big(\tilde{\mathcal{S}}^{\mathsf{T}}\gamma\big) \leq \frac{\epsilon}{2} \tag{83}$$

Define success as $A \cap B$. If event $A \cap B$ occurs, then from equation 82 and equation 83 the following is true for any solution $\Phi$ of IRMv1

$$\hat{R}(\Phi) + \lambda \hat{R}'(\Phi) \leq \hat{R}\big(\tilde{\mathcal{S}}^{\mathsf{T}}\gamma\big) + \lambda \hat{R}'\big(\tilde{\mathcal{S}}^{\mathsf{T}}\gamma\big) \leq \sigma^2 + \frac{\epsilon}{2} + \lambda\frac{\epsilon}{2} \tag{84}$$

Since $\hat{R}(\Phi) \geq 0$ it follows that

$$\hat{R}'(\Phi) \leq \frac{\sigma^2 + \frac{\epsilon}{2}}{\lambda} + \frac{\epsilon}{2} \implies \hat{R}'(\Phi) - R'(\Phi) + R'(\Phi) \leq \frac{\sigma^2 + \frac{\epsilon}{2}}{\lambda} + \frac{\epsilon}{2} \implies R'(\Phi) \leq \frac{\sigma^2 + \frac{\epsilon}{2}}{\lambda} + \epsilon \tag{85}$$

In the last implication in the above equation 85, we use the condition that event $B$ occurs. Let $\epsilon = \frac{\sigma^2}{\lambda}$ and substitute in the above equation 85 to get

$$R'(\Phi) \leq \frac{2\sigma^2}{\lambda} + \frac{\sigma^2}{2\lambda^2} \leq \frac{5\sigma^2}{2\lambda}(\text{since } \lambda \geq 1, \ \frac{\sigma^2}{2\lambda^2} \leq \frac{\sigma^2}{2\lambda}) \tag{86}$$

Recall Proposition 11 and see $\frac{5\sigma^2}{2\lambda}$ takes the role of $\epsilon$. If $\frac{5\sigma^2}{2\lambda} \leq \epsilon_0 \implies \lambda \geq \frac{5\sigma^2}{3\epsilon_{\text{th}}}$, then the condition in the Proposition 11 is true, then every solution of IRMv1 is $\tilde{\mathcal{S}}^{\mathsf{T}}\gamma(\alpha)$, where $\alpha \in [\frac{1}{1+\frac{1}{\omega\lambda_{\min}}\sqrt{\frac{5\sigma^2|\mathcal{E}_{tr}|}{2\lambda\pi^{\min}}}}, \frac{1}{1-\frac{1}{\omega\lambda_{\min}}\sqrt{\frac{5\sigma^2|\mathcal{E}_{tr}|}{2\lambda\pi^{\min}}}}] = [\frac{1}{1+\tau\sigma\sqrt{\frac{5}{3\lambda}}}, \frac{1}{1-\tau\sigma\sqrt{\frac{5}{3\lambda}}}]$

We arrived at the above result assuming $A \cap B$ occurs. We will now show that if $|D| \geq \max\{\frac{16L'^4\lambda^2}{\sigma^4}\log(\frac{4|\mathcal{H}_\Phi|}{\delta}), \frac{8L^2\lambda^2}{\sigma^4}\log(\frac{4}{\delta})\}$, then with a probability at least $1 - \delta$, $A \cap B$ occurs.

We write $\mathbb{P}(A) = \mathbb{P}\Big(\big\{D : |\hat{R}(\tilde{\mathcal{S}}^{\mathsf{T}}\gamma) - R(\tilde{\mathcal{S}}^{\mathsf{T}}\gamma)| \leq \frac{\epsilon}{2}\big\}\Big) = 1 - \mathbb{P}\Big(\big\{D : |\hat{R}(\tilde{\mathcal{S}}^{\mathsf{T}}\gamma) - R(\tilde{\mathcal{S}}^{\mathsf{T}}\gamma)| > \frac{\epsilon}{2}\big\}\Big)$. From Hoeffding's inequality in Lemma 1 and 3 it follows that

$$\mathbb{P}\Big(\big\{D : |\hat{R}(\tilde{\mathcal{S}}^{\mathsf{T}}\gamma) - R(\tilde{\mathcal{S}}^{\mathsf{T}}\gamma)| > \frac{\epsilon}{2}\big\}\Big) \leq 2\exp\Big(-\frac{|D|\epsilon^2}{8L^2}\Big)$$

$$2\exp\Big(-\frac{|D|\epsilon^2}{8L^2}\Big) \leq \frac{\delta}{2} \implies \mathbb{P}(A^c) \leq \frac{\delta}{2}$$

$$|D| \geq \frac{8L^2}{\epsilon^2}\log\Big(\frac{4}{\delta}\Big) \implies \mathbb{P}(A^c) \leq \frac{\delta}{2} \tag{87}$$

Next, we will show next that if $|D| \geq \frac{16L'^4}{\epsilon^2}\log(\frac{4|\mathcal{H}_\Phi|}{\delta})$, then with probability at least $1 - \frac{\delta}{2}$ event B happens.

Now we need to bound the probability $\mathbb{P}(B)$. We will find an upper bound on the failure probability using Hoeffding's inequality and 3 as follows

$$\mathbb{P}(B) = 1 - \mathbb{P}\Big(\big\{\tilde{D} : \exists\Phi \in \mathcal{H}_\Phi, |\hat{R}'(\Phi) - R'(\Phi)| > \frac{\epsilon}{2}\big\}\Big) = \mathbb{P}\Big(\bigcup_{\Phi\in\mathcal{H}_\Phi}\big\{\tilde{D} : |\hat{R}'(\Phi) - R'(\Phi)| > \frac{\epsilon}{2}\big\}\Big)$$

$$\mathbb{P}\Big(\bigcup_{\Phi\in\mathcal{H}_\Phi}\big\{\tilde{D} : |\hat{R}'(\Phi) - R'(\Phi)| > \frac{\epsilon}{2}\big\}\Big) \leq \sum_{\Phi\in\mathcal{H}_\Phi}\mathbb{P}\Big(\big\{\tilde{D} : |\hat{R}'(\Phi) - R'(\Phi)| > \frac{\epsilon}{2}\big\}\Big) \leq 2|\mathcal{H}_\Phi|e^{-\frac{\epsilon^2|D|}{16L'^4}}$$

$$2|\mathcal{H}_\Phi|e^{-\frac{\epsilon^2|D|}{16L'^4}} \leq \frac{\delta}{2} \implies \mathbb{P}(B^c) \leq \frac{\delta}{2}$$

$$|D| \geq \frac{16L'^4}{\epsilon^2}\log\Big(\frac{4|\mathcal{H}_\Phi|}{\delta}\Big) \implies \mathbb{P}(B^c) \leq \frac{\delta}{2} \tag{88}$$

Hence, we know that if $|D| \geq \frac{16L'^4}{\epsilon^2} \log(\frac{4|\mathcal{H}_\Phi|}{\delta})$, then with a probability at least $1 - \frac{\delta}{2}$ event $B$ happens.

Combining equation 87 and equation 88 we get if

$$|D| \geq \max \left\{ \frac{16L'^4}{\epsilon^2} \log \left( \frac{4|\mathcal{H}_\Phi|}{\delta} \right), \frac{8L^2}{\epsilon^2} \log \left( \frac{4}{\delta} \right) \right\}$$

then $A \cap B$ occurs with at least $1 - \delta$ probability. Substitute $\epsilon = \frac{5\sigma^2}{2\lambda}$ and this completes the proof.

$\square$

## 7.4 EXTENSIONS: POLYNOMIAL MODELS, INFINITE HYPOTHESIS CLASSES, BINARY-CLASSIFICATION

### 7.4.1 POLYNOMIAL MODEL

In the main body of the work, we did not cover the polynomial model in detail due to space limitations. In this section, we study the polynomial model and show how the results for the linear model can be generalized to this case. We first begin by stating the model itself.

**Assumption 8.**

$$e \sim \mathsf{Categorical}(\pi^e), \pi^e > 0 \forall e \in \mathcal{E}_{tr}$$
$$Y^e = \gamma^\mathsf{T} \zeta_p^c(Z_1^e) + \varepsilon^e, \varepsilon^e \perp Z_1^e, \mathbb{E}[\varepsilon^e] = 0, \mathbb{E}[(\varepsilon^e)^2] = \sigma^2, |\varepsilon^e| \leq \varepsilon^\mathsf{sup} \qquad (89)$$
$$X^e = S(Z_1^e, Z_2^e)$$

*Assume that $Z_1^e$ component of $S$ is invertible, i.e. $\exists \tilde{S}$ such that $\tilde{S}(S(Z_1^e, Z_2^e)) = Z_1^e$ and also $\tilde{S}^t \gamma \neq 0$. In the above $\zeta_p^a$ is a polynomial feature map of degree $p$ defined as $\zeta_p^a : \mathbb{R}^a \to \mathbb{R}^{a'}$, where $a$ denotes the dimension of the input to the map $\zeta_p^a$, $\zeta_p^a(W) = [W, W \otimes W, \ldots, (W \otimes W \ldots p \text{ times} \otimes W)] = [(W^{\otimes i})_{i=1}^p]$ and $\otimes$ is the Kronecker product. Also, $a' = \sum_{i=1}^p a^i$. $\forall e \in \mathcal{E}_{tr}, \pi^e \geq \frac{\pi^\mathsf{min}}{|\mathcal{E}_{tr}|}$. The support of distribution of $Z^e = [Z_1^e, Z_2^e]$, $\mathbb{P}_{Z^e}^e$, is bounded and the operator norm of $S$, $\|S\| = \sigma_\mathsf{max}(S)$ ($\sigma_\mathsf{max}(S)$ is maximum singular value of $S$), is also bounded.*

Define $Z^e = (Z_1^e, Z_2^e)$ and say $Z_1^e \in \mathbb{R}^c$ and $Z_2^e \in \mathbb{R}^d$.

Can we directly use the analysis from the linear case? We cannot directly use the polynomial map for the features as we also need to find an appropriate transformation of the matrix $S$, which preserves the linear relationship between the transformed features and the transformed variables $Z$. We carry out this exercise below.

Define $\bar{S} = \mathsf{diag}\left[(S^{\otimes i})_{i=1}^p\right]$, where $\bar{S}$ is a block diagonal matrix with diagonal matrices defining it given as $\left\{ S, S^{\otimes 2} \ldots, S^{\otimes p} \right\}$.

Define $\bar{X}^e = \zeta_p^n(X^e)$, where $\zeta_p^n(X^e)$ is the polynomial feature map of degree $p$ of $n$ dimensional input $X^e$. Similarly, define $\bar{Z}_1^e = \zeta_p^c(Z_1^e)$, where $\zeta_p^c(Z_1^e)$ is the polynomial feature map of degree $p$ of $c$ dimensional input $Z_1^e$ and define $\bar{Z}^e = \zeta_p^{c+d}(Z^e)$, where $\zeta_p^{c+d}(Z^e)$ is the polynomial feature map of degree $p$ of $c + d$ dimensional input $Z^e = (Z_1^e, Z_2^e)$. Observe that each component of $\bar{Z}_1^e$ is also in $\bar{Z}$.

From the model we know that $X^e = SZ^e$. We would like to remind the reader of the mixed product property of tensors. Consider matrices $A \in \mathbb{R}^{i \times j}$, $B \in \mathbb{R}^{k \times l}$ $C \in \mathbb{R}^{j \times p}$, $D \in \mathbb{R}^{l \times q}$.

$$(A \otimes B)(C \otimes D) = (AC) \otimes (BD) \qquad (90)$$

In the expressions that follow, we exploit the mixed-product property of Kronecker product stated above.

$$\bar{S}\bar{Z}^e = \mathsf{diag}\left[(S^{\otimes i})_{i=1}^p\right]\left[(Z^{e,\otimes i})_{i=1}^p\right] = \left[(SZ^e)^{\otimes i})_{i=1}^p\right] = \left[(X^{e,\otimes i})_{i=1}^p\right] = \zeta_p^n(X^e) = \bar{X}^e \quad (91)$$

Define $\bar{\bar{S}} = \text{diag}\left[(\tilde{S}^{\otimes i})_{i=1}^{p}\right]$.

$$\bar{\bar{S}}X^e = \bar{\bar{S}}\bar{S}\bar{Z}^e = \text{diag}\left[(\tilde{S}^{\otimes i})_{i=1}^{p}\right]\text{diag}\left[(S^{\otimes i})_{i=1}^{p}\right]\left[(Z^{e,\otimes i})_{i=1}^{p}\right]$$

$$\text{diag}\left[((\tilde{S}S)^{\otimes i})_{i=1}^{p}\right]\left[(Z^{e,\otimes i})_{i=1}^{p}\right] = \left[((\tilde{S}SZ^e)^{\otimes i})_{i=1}^{p}\right] = \left[((Z_1^e)^{\otimes i})_{i=1}^{p}\right] = \zeta_p^c(Z_1^e) = \bar{Z}_1^e \tag{92}$$

The dimensionality of $\bar{X}^e$ is $n^{'} = \sum_{i=1}^{p} n^i = \frac{n^{p+1}-n}{n-1}$.

**Assumption 9.** *Inductve bias. $\mathcal{H}_{\Phi}$ is a finite set of linear models (bounded) parametrized by $\Phi \in \mathbb{R}^{n^{'}}$. $\bar{\bar{S}}^{\mathsf{T}}\gamma \in \mathcal{H}_{\Phi}$. $\exists\, \omega > 0, \Omega > 0, \forall \Phi \in \mathcal{H}_{\Phi}, \omega \leq \|\Phi\|^2 \leq \Omega$ and $2\omega \leq \|\bar{\bar{S}}^{\mathsf{T}}\gamma\|^2 \leq \frac{2}{3+2\sqrt{2}}\Omega$,*

We next compute the norm of $\bar{S}$ in terms of the norm of $S$. Recall we are using operator norm defined as $\|S\| = \sigma_{\max}(S)$. $\|\bar{S}\| = \left\|\text{diag}\left[(S^{\otimes i})_{i=1}^{p}\right]\right\|$. Since $\bar{S}$ is a diagonal matrix $\|\bar{S}\| = \max_{i=1,...,p}\{\|S^{\otimes i}\|\}$. Also, note that$\|S^{\otimes i}\| = \|S\|^i$ (Laub, 2005). Therefore, $\|\bar{S}\| = \max_{i=1,...,p}\{\|S\|^i\}$. Hence, if $\|S\|$ is bounded, $\|\bar{S}\|$ is also bounded.

Also, $\|\bar{Z}^e\|^2 = \sum_i \|Z^{e,\otimes i}\|^2$. Observe that $\|Z^{e,\otimes i}\| = \|Z^e\|^i$. Hence, if $\|Z^e\|$ is bounded, $\|\bar{Z}^e\|$ is also bounded. Since $\bar{X}^e = \bar{S}\bar{Z}^e$. We can conclude that $\|\bar{X}^e\|$ is also bounded. We can now follow the same line of reasoning as in equation 74, equation 75,equation 76, equation 77 to conclude that the loss and the gradient of the loss are bounded.

We rewrite the above model in Assumption 8 as a linear model in terms of the transformed features.

$$e \sim \text{Categorical}(\pi^e), \pi^e > 0 \forall e \in \mathcal{E}$$
$$Y^e = \gamma^t \bar{Z}_1 + \varepsilon^e, \varepsilon^e \perp \bar{Z}_1^e, \mathbb{E}[\varepsilon^e] = 0, \mathbb{E}[(\varepsilon^e)^2] = \sigma^2, |\varepsilon^e| \leq \varepsilon^{\text{sup}} \tag{93}$$
$$\bar{X}^e = \bar{S}\bar{Z}^e$$

We showed above in equation 92 that $\bar{Z}_1^e$ defined in equation 91 component of $\bar{Z}^e$ is invertible, $\bar{\bar{S}}\bar{S}\bar{Z}^e = \bar{Z}_1^e$. We have also shown above that support of $\bar{Z}^e$ is bounded and the norm of $\bar{S}$ is bounded and as a result $\|\bar{X}^e\|$ and the loss and the gradient of the loss (conditions in Assumption 3 are satisfied) are bounded. We adapt the linear general position assumption (Assumption 6) for the polynomial case below.

**Assumption 10.** *Linear general position of training environments. A set of training environments $\mathcal{E}_{tr}$ is said to lie in a linear general position of degree $r$ for some $r \in \mathbb{N}$ if $|\mathcal{E}_{tr}| > n^{'} - r + n^{'}/r$ and for non-zero $x \in \mathbb{R}^{n^{'}}$ and*

$$\text{dim}\left(\text{span}\left\{\mathbb{E}^e[\bar{X}^e\bar{X}^{e,\mathsf{T}}]x - \mathbb{E}^e[\bar{X}^{e,\mathsf{T}}\varepsilon^e]\right\}_{e \in \mathcal{E}_{tr}}\right) > n^{'} - r \tag{94}$$

We denote $\mathbb{E}[\bar{X}^e\bar{X}^{e,\mathsf{T}}] = \bar{\Sigma}_e$

**Assumption 11.** *For all the environments $e \in \mathcal{E}_{tr}$, $\bar{\Sigma}_e$ is positive definite.*

Define the minimum eigenvalue over all the matrices $\bar{\Sigma}_e$ as $\bar{\lambda}_{\min} = \min_{e \in \mathcal{E}_{tr}} \lambda(\bar{\Sigma}_e)$. Define $\bar{\epsilon}_{\text{th}} = \frac{24-16\sqrt{2}}{3}\frac{\pi^{\min}}{|\mathcal{E}_{tr}|}(\omega\bar{\lambda}_{\min})^2$.

From the analysis in this section, we see that we have been able to construct a linear model identical to 5, where the role of $X^e, Z^e, S, \tilde{S}, \Sigma^e, \lambda_{\min}, \epsilon_{\text{th}}$ is taken by $\bar{X}^e, \bar{Z}^e, \bar{S}, \bar{\bar{S}}, \bar{\Sigma}^e, \bar{\lambda}_{\min}, \bar{\epsilon}_{\text{th}}$. Now we are ready to use the result already proven for linear model and state the next Proposition in terms of the parameters for the polynomial model.

**Proposition 13.** *Let $\ell$ be the square loss. For every $\epsilon \in (0, \bar{\epsilon}^{\text{th}})$, $\delta \in (0, 1)$, if Assumptions 8, 9, 10 (with $r = 1$), 11 hold and if the number of data points $|D|$ is greater than $\frac{16L^{'4}}{\epsilon^2}\log\left(\frac{2|\mathcal{H}_{\Phi}|}{\delta}\right)$, then with a probability at least $1 - \delta$, every solution $\hat{\Phi}$ to EIRM (equation 6) satisfies $\hat{\Phi} = (\bar{\bar{S}}^{\mathsf{T}}\gamma)\alpha$, where $\alpha \in [\frac{1}{1+\tau\sqrt{\epsilon}}, \frac{1}{1-\tau\sqrt{\epsilon}}]$.*

### 7.4.2 INFINITE HYPOTHESIS CLASSES

In the work so far we have assumed that the hypothesis class $\mathcal{H}_\Phi$ is finite. In this section, we discuss infinite hypothesis class extensions. Before we do that we state an important result on covering numbers that we will use soon.

**Lemma 5.** *(Shalev-Shwartz & Ben-David, 2014) Define a set $\mathcal{A} = \{a \in \mathbb{R}^k, \|a\|^2 \leq A^{\mathsf{sup}}\}$. Covering number for $\eta$-cover of $\mathcal{A}$ given as $N_\eta(\mathcal{A})$ is bounded as*

$$N_\eta(\mathcal{A}) \leq \Big(\frac{2\sqrt{A^{\mathsf{sup}}k}}{\eta}\Big)^k \tag{95}$$

**A. Infinite Hypothesis Class: Confounders and Anti-causal variables**

In this section, we seek to extend Proposition 5 to infinite hypothesis classes.

We restate the Assumption 7 for linear models.

**Assumption 12.** *Inductve bias.* $\mathcal{H}_\Phi$ *is a set of linear models (bounded) parametrized by $\Phi \in \mathbb{R}^n$ $\mathcal{H}_\Phi = \{\Phi \in \mathbb{R}^n,\ 0 < \omega \leq \|\Phi\|^2 \leq \Omega\}$. $\tilde{S}^{\mathsf{T}}\gamma \in \mathcal{H}_\Phi$. $2\omega \leq \|\tilde{S}^{\mathsf{T}}\gamma\|^2 \leq \frac{2}{3+2\sqrt{2}}\Omega$,*

Note that the only difference between Assumption 7 and Assumption 12 is that the hypothesis class is not required to be finite anymore.

We already established in equation 75, equation 76, equation 77 that from Assumptions 5 and 7, we can show that the conditions in Assumption 3 hold, i.e., loss and the gradient of the loss are bounded, and also $X^e, Y^e$ are bounded . The same conclusion follows from Assumptions 5 and Assumption 12. Hence, for the rest of this section, we can state that $\|X^e\| \leq X^{\mathsf{sup}}$, $|Y^e| \leq K$, the square loss $\ell(\Phi(\cdot), \cdot)$ is bounded by $L$ and $\frac{\partial \ell(w \cdot (\Phi(\cdot), \cdot)}{\partial w}\Big|_{w=1.0}$ is bounded by $L'$. In the next lemma, we aim to show that if Assumption 5 and 12 hold, then $R'(\Phi)$ is Lipschitz continuous.

**Lemma 6.** *If Assumption 5 and 12 hold, then $R'(\Phi)$ is Lipschitz continuous in $\Phi$.*

*Proof.* The output of the model $|\Phi^{\mathsf{T}}X| \leq \|\Phi\|\|X\| \leq \sqrt{\Omega}X^{\mathsf{sup}}$ (From Cauchy-Schwarz).

$$R'(\Phi) = \sum_e \pi^e \mathbb{E}^e\Big[\Phi^{\mathsf{T}}X^e\big(\Phi^{\mathsf{T}}X^e - Y^e\big)\Big]^2$$

$$|R'(\Phi_1) - R'(\Phi_2)| = \Big|\sum_e \pi^e\Big(\mathbb{E}^e\big[\Phi_1^{\mathsf{T}}X^e\big(\Phi_1^{\mathsf{T}}X^e - Y^e\big)\big]^2 - \mathbb{E}^e\big[\Phi_1^{\mathsf{T}}X^e\big(\Phi_2^{\mathsf{T}}X^e - Y^e\big)\big]^2\Big)\Big|$$

$$\leq \sum_e \pi^e\Big|\mathbb{E}^e\big[\Phi_1^{\mathsf{T}}X^e\big(\Phi_1^{\mathsf{T}}X^e - Y^e\big)\big]^2 - \mathbb{E}^e\big[\Phi_2^{\mathsf{T}}X^e\big(\Phi_2^{\mathsf{T}}X^e - Y^e\big)\big]^2\Big| \tag{96}$$

We bound each term in the summation in the equation 96 above

$$\left| \mathbb{E}^e \left[ \Phi_1^\mathsf{T} X^e (\Phi_1^\mathsf{T} X^e - Y^e) \right]^2 - \mathbb{E}^e \left[ \Phi_2^\mathsf{T} X^e (\Phi_2^\mathsf{T} X^e - Y^e) \right]^2 \right|$$

$$= \left| \left( \mathbb{E}^e \left[ \Phi_1^\mathsf{T} X^e (\Phi_1^\mathsf{T} X^e - Y^e) \right] - \mathbb{E}^e \left[ \Phi_2^\mathsf{T} X^e (\Phi_2^\mathsf{T} X^e - Y^e) \right] \right) \left( \mathbb{E}^e \left[ \Phi_1^\mathsf{T} X^e (\Phi_1^\mathsf{T} X^e - Y^e) \right] + \right. \right.$$
$$\left. \left. \mathbb{E}^e \left[ \Phi_2^\mathsf{T} X^e (\Phi_2^\mathsf{T} X^e - Y^e) \right] \right) \right|$$

$$\leq \left| \mathbb{E}^e \left[ \Phi_1^\mathsf{T} X^e (\Phi_1^\mathsf{T} X^e - Y^e) \right] - \mathbb{E}^e \left[ \Phi_2^\mathsf{T} X^e (\Phi_2^\mathsf{T} X^e - Y^e) \right] \right| 2\sqrt{\Omega} X^{\mathsf{sup}} (\sqrt{\Omega} X^{\mathsf{sup}} + K)$$

$$\leq \left| \mathbb{E}^e \left[ \Phi_1^\mathsf{T} X^e (\Phi_1^\mathsf{T} X^e - Y^e) \right] - \mathbb{E}^e \left[ \Phi_2^\mathsf{T} X^e (\Phi_1^\mathsf{T} X^e - Y^e) \right] + \mathbb{E}^e \left[ \Phi_2^\mathsf{T} X^e (\Phi_1^\mathsf{T} X^e - Y^e) \right] - \right.$$
$$\left. \mathbb{E}^e \left[ \Phi_2^\mathsf{T} X^e (\Phi_2^\mathsf{T} X^e - Y^e) \right] \right| 2\sqrt{\Omega} X^{\mathsf{sup}} (\sqrt{\Omega} X^{\mathsf{sup}} + K)$$

$$= \left| \mathbb{E}^e \left[ (\Phi_1^\mathsf{T} X^e - \Phi_2^\mathsf{T} X^e)(\Phi_1^\mathsf{T} X^e - Y^e) \right] + \mathbb{E}^e \left[ \Phi_2^\mathsf{T} X^e (\Phi_1^\mathsf{T} X^e - \Phi_2^\mathsf{T} X^e) \right] \right| 2\sqrt{\Omega} X^{\mathsf{sup}} (\sqrt{\Omega} X^{\mathsf{sup}} + K)$$

$$\leq \left| \mathbb{E}^e \left[ (\Phi_1^\mathsf{T} X^e - \Phi_2^\mathsf{T} X^e)(\Phi_1^\mathsf{T} X^e - Y^e) \right] \right| 2\sqrt{\Omega} X^{\mathsf{sup}} (\sqrt{\Omega} X^{\mathsf{sup}} + K) +$$
$$\left| \mathbb{E}^e \left[ \Phi_2^\mathsf{T} X^e (\Phi_1^\mathsf{T} X^e - \Phi_2^\mathsf{T} X^e) \right] \right| 2\sqrt{\Omega} X^{\mathsf{sup}} (\sqrt{\Omega} X^{\mathsf{sup}} + K)$$

$$\tag{97}$$

We bound each term in the last line in equation 97

$$\left| \mathbb{E}^e \left[ (\Phi_1^\mathsf{T} X^e - \Phi_2^\mathsf{T} X^e)(\Phi_1^\mathsf{T} X^e - Y^e) \right] \right|$$
$$\leq \mathbb{E}^e \left[ \left| (\Phi_1^\mathsf{T} X^e - \Phi_2^\mathsf{T} X^e) \right| \left| (\Phi_1^\mathsf{T} X^e - Y^e) \right| \right]$$
$$\leq (\sqrt{\Omega} X^{\mathsf{sup}} + K) \mathbb{E}^e \left[ \left| (\Phi_1 - \Phi_2)^\mathsf{T} X^e) \right| \right]$$
$$\leq (\sqrt{\Omega} X^{\mathsf{sup}} + K) \mathbb{E}^e \left[ \|\Phi_1 - \Phi_2\| X^{\mathsf{sup}} \right] \quad \text{(Cauchy-Scwarz)}$$
$$\leq (\sqrt{\Omega} X^{\mathsf{sup}} + K) X^{\mathsf{sup}} \|\Phi_1 - \Phi_2\|$$

$$\tag{98}$$

$$\left| \mathbb{E}^e \left[ \Phi_2^\mathsf{T} X^e (\Phi_1^\mathsf{T} X^e - \Phi_2^\mathsf{T} X^e) \right] \right|$$
$$\leq \mathbb{E}^e \left[ \left| \Phi_2^\mathsf{T} X^e \right| \left| (\Phi_1^\mathsf{T} X^e - \Phi_2^\mathsf{T} X^e) \right| \right]$$
$$\leq \sqrt{\Omega} X^{\mathsf{sup}} \mathbb{E}^e \left[ \left| (\Phi_1 - \Phi_2)^\mathsf{T} X^e) \right| \right] \quad \text{(Cauchy-Scwarz)}$$
$$\leq \sqrt{\Omega} X^{\mathsf{sup}} \mathbb{E}^e \left[ \|\Phi_1 - \Phi_2\| X^{\mathsf{sup}} \right]$$
$$\leq \sqrt{\Omega} (X^{\mathsf{sup}})^2 \|\Phi_1 - \Phi_2\|$$

$$\tag{99}$$

Substituting equation 98 and equation 99 in equation 97 to get

$$|R'(\Phi_1) - R'(\Phi_2)| \leq 2\sqrt{\Omega}(X^{\mathsf{sup}})^2(\sqrt{\Omega} X^{\mathsf{sup}} + K)(2\sqrt{\Omega} X^{\mathsf{sup}} + K)\|\Phi_1 - \Phi_2\| \tag{100}$$

Therefore, $R'$ is Lipschitz with a constant $C' \leq 2\sqrt{\Omega}(X^{\mathsf{sup}})^2(\sqrt{\Omega} X^{\mathsf{sup}} + K)(2\sqrt{\Omega} X^{\mathsf{sup}} + K)$.

$$\square$$

We just showed that $R'$ is Lipschitz continuous and we set its Lipschitz constant as $C'$.

**Proposition 14.** *Let $\ell$ be the square loss. For every $\epsilon \in (0, \epsilon^{\mathsf{th}})$ and $\delta \in (0, 1)$, if Assumptions 5, 6 (with $r = 1$), 12 hold and if the number of samples $|D|$ is greater than $\frac{32L'^4}{\epsilon^2} \left[ n \log \left( \frac{16C' \sqrt{\Omega} n}{\epsilon} \right) + \log \left( \frac{2}{\delta} \right) \right]$, then with a probability at least $1 - \delta$ every solution $\hat{\Phi}$ to EIRM (equation 6) satisfies $\hat{\Phi} = \tilde{S}^\mathsf{T} \gamma(\alpha)$, where $\alpha \in [\frac{1}{1 + \tau \sqrt{\epsilon}}, \frac{1}{1 - \tau \sqrt{\epsilon}}]$.*

*Proof.* Following the proof of Proposition 5, our goal is to compute the probability of event $A$: $\{\forall \Phi \in \mathcal{H}_\Phi, |\hat{R}'(\Phi) - R'(\Phi)| \leq \frac{\epsilon}{2}\}$. We construct a minimum cover of size $N_\eta(\mathcal{H}_\Phi)$ (See Lemma 5) with points $\mathcal{C} = \{\Phi_j\}_{j=1}^b$.

Compute the probability of failure at one point $\Phi_j$ in the cover

$$\mathbb{P}\Big[\Big\{\tilde{D} : |R'(\Phi_j) - \hat{R}'(\Phi_j)| > \frac{\epsilon}{4}\Big\}\Big] < 2e^{-\frac{\epsilon^2|D|}{32L'^4}} \tag{101}$$

We use union bound to bound the probability of failure over the cover $\mathcal{C}$ as follows

$$\mathbb{P}\Big[\Big\{\tilde{D} : \max_{\Phi_j \in \mathcal{C}} |R'(\Phi_j) - \hat{R}'(\Phi_j)| > \frac{\epsilon}{4}\Big\}\Big] < 2N_\eta(\mathcal{H}_\Phi)e^{-\frac{\epsilon^2|D|}{32L'^4}} \tag{102}$$

Now consider any $\Phi \in \mathcal{H}_\Phi$ and suppose $\Phi_j$ is nearest point to it in the cover.

$$|R'(\Phi) - \hat{R}'(\Phi)| = |R'(\Phi) - R'(\Phi_j) + R'(\Phi_j) - \hat{R}'(\Phi_j) + \hat{R}'(\Phi_j) - \hat{R}'(\Phi)|$$
$$\leq |R'(\Phi) - R'(\Phi_j)| + |R'(\Phi_j) - \hat{R}'(\Phi_j)| + |\hat{R}'(\Phi_j) - \hat{R}'(\Phi)| \leq |R'(\Phi_j) - \hat{R}'(\Phi_j)| + 2\eta C' \tag{103}$$

In the above simplification, we exploited the Lipschitz continuity of $R'$. Therefore, for each $\Phi \in \mathcal{H}_\Phi$

$$|R'(\Phi) - \hat{R}'(\Phi)| \leq \max_{\Phi_j \in \mathcal{C}} |R'(\Phi_j) - \hat{R}'(\Phi_j)| + 2\eta C'$$
$$\max_{\Phi \in \mathcal{H}_\Phi} |R'(\Phi) - \hat{R}'(\Phi)| \leq \max_{\Phi_j \in \mathcal{C}} |R'(\Phi_j) - \hat{R}'(\Phi_j)| + 2\eta C' \tag{104}$$

Set $\eta = \frac{\epsilon}{8C'}$ in equation 104 and from equation 102 with probability at least $1 - N_\eta(\mathcal{H}_\Phi)2e^{-\frac{\epsilon^2|D|}{32L'^4}}$

$$\max_{\Phi \in \mathcal{H}_\Phi} |R'(\Phi) - \hat{R}'(\Phi)| \leq \epsilon/2 \text{ (since } \max_{\Phi_j \in \mathcal{C}} |R'(\Phi_j) - \hat{R}'(\Phi_j)| \leq \epsilon/4 \text{ )} \tag{105}$$

We bound $N_\eta(\mathcal{H}_\Phi)2e^{-\frac{\epsilon^2|D|}{32L'^4}} \leq \delta$ and solve for bound on $|D|$ to get

$$|D| \geq \frac{8L'^4}{\frac{\epsilon^2}{4}} \log\Big(\frac{2N_\eta(\mathcal{H}_\Phi)}{\delta}\Big) \text{ (Use Lemma 5)}$$
$$|D| \geq \frac{32L'^4}{\epsilon^2}\Big[n \log\Big(\frac{16C'\sqrt{\Omega n}}{\epsilon}\Big) + \log\Big(\frac{2}{\delta}\Big)\Big] \tag{106}$$

Therefore, if condition in equation 106 holds, then event $A$ occurs and following the same argument as in the proof of Proposition 5 the proof is complete.

$\square$

## B. Infinite hypothesis class: Lipschitz continuous functions

In this section, we seek to extend Proposition 2 and 4 to infinite hypothesis class of Lipschitz continuous functions that we formally define next. Define a map $\Phi : \mathcal{P} \times \mathcal{X} \to \mathbb{R}$ from the parameter space $\mathcal{P}$ and the feature space $\mathcal{X}$ to reals. Each $p \in \mathcal{P}$ is a possible choice for the representation $\Phi(p, \cdot)$. Consider neural networks as an example, $\mathcal{P}$ represents the set of the values the weights of the network can take.

**Assumption 13.** $\Phi : \mathcal{P} \times \mathcal{X} \to \mathbb{R}$ *is a a Lipschitz continuous function (with Lipschitz constant say $Q$).*

$\mathcal{P} \subset \mathbb{R}^k$ *is closed and bounded, thus there exists a $P < \infty$ such that $\forall p \in \mathcal{P}$, $\|p\|^2 \leq P$.*

$\mathcal{X} \subset \mathbb{R}^n$ *is closed and bounded, thus there exists a $X^{\mathsf{sup}} < \infty$ such that $\forall x \in \mathcal{X}$, $\|x\| \leq X^{\mathsf{sup}}$.*

$\mathcal{Y} \subset \mathbb{R}$ *is closed and bounded, thus there exists a $K < \infty$ such that $\forall y \in \mathcal{Y}$, $|y| \leq K$*

**Assumption 14.** *Lipschitz loss and gradient of loss.* $R(\Phi)$ *is Lipschitz with a constant* $C$*,* $R^{'}(\Phi)$ *is Lipchitz with a constant* $C^{'}$*.*

**From Assumption 13 derive the conditions in Assumption 3 and Assumption 14** $\Phi : \mathcal{P} \times \mathcal{X} \to \mathbb{R}$ is a continuous function defined over closed and bounded domain $\mathcal{P} \times \mathcal{X}$ (domain is compact) and as a result $\Phi$ is bounded say by $M$.

Consider square loss $\ell(\Phi(p, X), Y) = (Y - \Phi(p, X))^2 \leq (M + K)^2$. Hence, there exists an $L$ such that $|\ell(\Phi(\cdot), \cdot)| \leq L \leq (M + K)^2$.

$\left.\frac{\partial \ell(w.\Phi(\cdot),\cdot)}{\partial w}\right|_{w=1.0}$ is bounded: $\left|\left.\frac{\partial \ell(w.\Phi(\cdot),\cdot)}{\partial w}\right|_{w=1.0}\right| = |(Y - \Phi(p, X))\Phi(p, X)| \leq (K + M)M$.
Hence, there exists an $L'$ such that $\left|\left.\frac{\partial \ell(w.\Phi(\cdot),\cdot)}{\partial w}\right|_{w=1.0}\right| \leq L' \leq (K + M)M$

**Lemma 7.** *If Assumption 13 holds, then* $R(\Phi(p, \cdot))$ *and* $R^{'}(\Phi(p, \cdot))$ *are Lipschitz continuous in* $p$*.*

$R$ is Lipschitz:

$$
\begin{aligned}
|R(\Phi(p, \cdot) - R(\Phi(q, \cdot))| &= \left| \sum_e \pi^e \left( \mathbb{E}^e[(\Phi(p, X^e) - Y^e)^2] - \mathbb{E}^e[(\Phi(q, X^e) - Y^e)^2] \right) \right| \\
&\leq \sum_e \pi^e \left| \mathbb{E}^e \left[ (\Phi(p, X^e) - Y^e)^2 \right] - \mathbb{E}^e \left[ (\Phi(q, X^e) - Y^e)^2 \right] \right| \\
&= \sum_e \pi^e \left| \mathbb{E}^e \left[ \Phi(p, X^e) - \Phi(q, X^e) \right] \mathbb{E}^e \left[ \Phi(p, X^e) + \Phi(q, X^e) - 2Y \right] \right| \\
&\leq \sum_e \pi^e \mathbb{E}^e \left[ \left| \Phi(p, X^e) - \Phi(q, X^e) \right| \right] \mathbb{E}^e \left[ \left| \Phi(p, X^e) + \Phi(q, X^e) - 2Y \right| \right] \\
&\leq \sum_e \pi^e \mathbb{E}^e \left[ \left| \Phi(p, X^e) - \Phi(q, X^e) \right| \right] 2(M + K) \\
&\leq \|p - q\| 2(M + K)Q
\end{aligned}
$$

(107)

Therefore, $R$ is Lipschitz with a constant $C \leq 2(M + K)Q$

$R^{'}$ is Lipschitz:

$$
\begin{aligned}
|R^{'}(\Phi(p, \cdot)) - R^{'}(\Phi(q, \cdot))| &= \left| \sum_e \pi^e \left( \mathbb{E}^e \left[ \Phi(p, X^e)(\Phi(p, X^e) - Y^e) \right]^2 - \mathbb{E} \left[ \Phi(q, X^e)(\Phi(q, X^e) - Y^e) \right]^2 \right) \right| \\
&\leq \sum_e \pi^e \left| \mathbb{E}^e \left[ \Phi(p, X^e)(\Phi(p, X^e) - Y^e) \right]^2 - \mathbb{E}[\Phi(q, X^e)(\Phi(q, X^e) - Y^e)]^2 \right|
\end{aligned}
$$

(108)

We bound each term in the summation in the equation 96 above

$$\left|\left(\mathbb{E}^e\Big[\Phi(p,X^e)\big(\Phi(p,X^e)-Y^e\big)\Big]-\mathbb{E}^e[\Phi(q,X^e)\big(\Phi(q,X^e)-Y^e\big)]\right)\left(\mathbb{E}^e\Big[\Phi(p,X^e)\big(\Phi(p,X^e)-Y^e\big)\Big]+\right.\right.$$
$$\left.\left.\mathbb{E}^e\Big[\Phi(q,X^e)\big(\Phi(q,X^e)-Y^e\big)\Big]\right)\right|$$
$$\leq\left|\left(\mathbb{E}^e\Big[\Phi(p,X^e)\big(\Phi(p,X^e)-Y^e\big)\Big]-\mathbb{E}^e\Big[\Phi(q,X^e)\big(\Phi(q,X^e)-Y^e\big)\Big]\right)\right|2M(M+K)$$
$$\leq\left|\left(\mathbb{E}^e\Big[\Phi(p,X^e)\big(\Phi(p,X^e)-Y^e\big)\Big]-\mathbb{E}^e\Big[\Phi(q,X^e)\big(\Phi(p,X^e)-Y^e\big)\Big]+\mathbb{E}^e\Big[\Phi(q,X^e)\big(\Phi(p,X^e)-Y^e\big)\Big]-\right.\right.$$
$$\left.\left.\mathbb{E}^e\Big[\Phi(q,X^e)\big(\Phi(q,X^e)-Y^e\big)\Big]\right)\right|2M(M+K)$$
$$=\left|\left(\mathbb{E}^e\Big[\big(\Phi(p,X^e)-\Phi(q,X^e)\big)\big(\Phi(p,X^e)-Y^e\big)\Big]\right)+\mathbb{E}^e\Big[\Phi(q,X^e)\big(\Phi(p,X^e)-\Phi(q,X^e)\big)\Big]\right|2M(M+K)$$
$$\leq\left|\left(\mathbb{E}^e\Big[\big(\Phi(p,X^e)-\Phi(q,X^e)\big)\big(\Phi(p,X^e)-Y^e\big)\Big]\right)\right|2M(M+K)+$$
$$\left|\mathbb{E}^e\Big[\Phi(q,X^e)\big(\Phi(p,X^e)-\Phi(q,X^e)\big)\Big]\right|2M(M+K)$$
$$\tag{109}$$

We bound each term in the last line in equation 109

$$\left|\big(\mathbb{E}^e\big[(\Phi(p,X^e)-\Phi(q,X^e))(\Phi(p,X^e)-Y^e)]\big)\right|$$
$$\leq\mathbb{E}^e\big[\big|(\Phi(p,X^e)-\Phi(q,X^e))\big|\big|(\Phi(p,X^e)-Y^e)\big|\big]$$
$$\leq(M+K)Q\|p-q\|$$
$$\tag{110}$$

$$\big|\mathbb{E}\big[\Phi(q,X^e)(\Phi(p,X^e)-\Phi(q,X^e))\big]\big|$$
$$\leq\mathbb{E}\big[\big|\Phi(q,X^e)\big|\big|(\Phi(p,X^e)-\Phi(q,X^e))\big|\big]$$
$$\leq MQ\|p-q\|$$
$$\tag{111}$$

Substituting equation 110 and equation 111 in equation 109 to get

$$|R^{'}(\Phi(p,\cdot))-R^{'}(\Phi(q,\cdot))|\leq 2M(M+K)(2M+K)Q\|p-q\|\tag{112}$$

Therefore, $R^{'}$ is Lipschitz with a constant $C^{'}\leq 2M(M+K)(2M+K)Q$.

### C. EIRM: Sample complexity with no distributional assumptions

In this section, we discuss the extension of Proposition 2 to the infinite hypothesis class case. Consider the problem in equation 6 and replace the $\epsilon$ with $\epsilon+\kappa$. Define distance between the two sets $\mathcal{S}^{\mathsf{IV}}(\epsilon)$ and its approximation $\mathcal{S}^{\mathsf{IV}}(\epsilon+\kappa)$ as follows

$$\mathsf{dis}(\kappa)=\max_{g\in\mathcal{S}^{\mathsf{IV}}(\epsilon+\kappa)}\min_{h\in\mathcal{S}^{\mathsf{IV}}(\epsilon)}\mathsf{d}(g,h)\tag{113}$$

where $\mathsf{d}(g,h)$ is some metric that measures the distance between functions $g$ and $h$. Observe that if $a\leq b$, $\mathsf{dis}(a)\leq\mathsf{dis}(b)$.

**Assumption 15.** $\lim_{k\to 0}\mathsf{dis}(\kappa)=0$

Define

$$D^*=\max\left\{\frac{32L^2}{\nu^2}\Big[k\log\Big(\frac{16C\sqrt{Pk}}{\nu}\Big)+\log\Big(\frac{2}{\delta}\Big)\Big],\frac{8L^{'4}}{\kappa^2}\Big[k\log\Big(\frac{8C'\sqrt{Pk}}{\kappa}\Big)+\log\Big(\frac{2}{\delta}\Big)\Big]\right\}$$

**Proposition 15.** *For every $\nu>0$ and $\delta\in(0,1)$, if Assumption 13, 15 hold, then $\exists\,\kappa>0$ such that if the number of samples $|D|$ is greater than $D^*$, then with a probability at least $1-\delta$, every solution $\hat{\Phi}$ to EIRM (replace $\epsilon$ with $\epsilon+\kappa$ inequation 6) in $\mathcal{S}^{\mathsf{IV}}(\epsilon+2\kappa)$ and $|R(\hat{\Phi})-R(\Phi^*)|\leq\nu$, where $\Phi^*$ is a solution of IRM in equation 5.*

*Proof.* We divide the proof in two parts.

Define an event $A$: $\{\tilde{D} : \forall p \in \mathcal{P}, |R^{'}(\Phi(p, \cdot)) - \hat{R}^{'}(\Phi(p, \cdot))| \leq \kappa\}$.

In the first half, we will show that if event $A$ occurs, then $\mathcal{S}^{\text{IV}}(\epsilon) \subseteq \hat{\mathcal{S}}^{\text{IV}}(\epsilon + \kappa) \subseteq \mathcal{S}^{\text{IV}}(\epsilon + 2\kappa)$ and then bound the probability of $A$ not occuring.

$$
\begin{aligned}
R^{'}(\Phi(p, \cdot)) \leq \epsilon &\implies R^{'}(\Phi(p, \cdot)) - \hat{R}^{'}(\Phi(p, \cdot)) + \hat{R}^{'}(\Phi(p, \cdot)) \leq \epsilon \implies \\
\hat{R}^{'}(\Phi(p, \cdot)) &\leq \epsilon + |R^{'}(\Phi(p, \cdot)) - \hat{R}^{'}(\Phi(p, \cdot))| \implies \hat{R}^{'}(\Phi(p, \cdot)) \leq \epsilon + \kappa
\end{aligned}
\tag{114}
$$

Therefore, $\mathcal{S}^{\text{IV}}(\epsilon) \subseteq \hat{\mathcal{S}}^{\text{IV}}(\epsilon + \kappa)$

$$
\begin{aligned}
\hat{R}^{'}(\Phi(p, \cdot)) \leq \epsilon + \kappa &\implies \hat{R}^{'}(\Phi(p, \cdot)) - R^{'}(\Phi(p, \cdot)) + R^{'}(\Phi(p, \cdot)) \leq \epsilon + \kappa \implies \\
R^{'}(\Phi(p, \cdot)) &\leq \epsilon + \kappa + |R^{'}(\Phi(p, \cdot)) - \hat{R}^{'}(\Phi(p, \cdot))| \implies R^{'}(\Phi(p, \cdot)) \leq \epsilon + 2\kappa
\end{aligned}
\tag{115}
$$

Therefore, $\hat{\mathcal{S}}^{\text{IV}}(\epsilon + \kappa) \subseteq \mathcal{S}^{\text{IV}}(\epsilon + 2\kappa)$

We bound the probability of event $A$ not occurring. Using the covering number (from Lemma 5) we construct a minimum cover of size $b = N_\eta(\mathcal{P})$ with points $\mathcal{C}_1 = \{p_j\}_{j=1}^b$.

Compute the probability of failure at one point $p_j$ in the cover

$$
\mathbb{P}\left[\left\{\tilde{D} : |R^{'}(\Phi(p_j, \cdot)) - \hat{R}^{'}(\Phi(p_j, \cdot))| > \frac{\kappa}{2}\right\}\right] < 2e^{-\frac{\kappa^2 |D|}{8L'^4}}
\tag{116}
$$

We use union bound to bound the probability of the failure over the entire cover $\mathcal{C}_1$ as

$$
\mathbb{P}\left[\left\{\tilde{D} : \max_{p_j \in \mathcal{C}_1} |R^{'}(\Phi(p_j, \cdot)) - \hat{R}^{'}(\Phi(p_j, \cdot))| > \frac{\kappa}{2}\right\}\right] < N_\eta(\mathcal{P})2e^{-\frac{\kappa^2 |D|}{8L'^4}}
\tag{117}
$$

Now consider any $p \in \mathcal{P}$ and suppose $p_j$ is nearest point to it in the cover.

$$
\begin{aligned}
&|R^{'}(\Phi(p, \cdot)) - \hat{R}^{'}(\Phi(p, \cdot))| = \\
&|R^{'}(\Phi(p, \cdot)) - R^{'}(\Phi(p_j, \cdot)) + R^{'}(\Phi(p_j, \cdot)) - \hat{R}^{'}(\Phi(p_j, \cdot)) + \hat{R}^{'}(\Phi(p_j, \cdot)) - \hat{R}^{'}(\Phi(p, \cdot))| \\
&\leq |R^{'}(\Phi(p_j, \cdot)) - \hat{R}^{'}(\Phi(p_j, \cdot))| + 2\eta C^{'}
\end{aligned}
\tag{118}
$$

In the above simplficiation, we exploit the Lipschitz continuity of $R^{'}$.

Therefore

$$
\begin{aligned}
\forall p \in \mathcal{P} \ |R^{'}(\Phi(p, \cdot)) - \hat{R}^{'}(\Phi(p, \cdot))| &\leq \max_{p_j \in \mathcal{C}_1} |R^{'}(\Phi(p_j, \cdot)) - \hat{R}^{'}(\Phi(p_j, \cdot))| + 2\eta C^{'} \\
\max_{p \in \mathcal{P}} |R^{'}(\Phi(p, \cdot)) - \hat{R}^{'}(\Phi(p, \cdot))| &\leq \max_{p_j \in \mathcal{C}_1} |R^{'}(\Phi(p_j, \cdot)) - \hat{R}^{'}(\Phi(p_j, \cdot))| + 2\eta C^{'}
\end{aligned}
\tag{119}
$$

Set $\eta = \frac{\kappa}{4C^{'}}$ in equation 119 and from equation 117 with probability at least $1 - N_\eta(\mathcal{P})2e^{-\frac{\kappa^2 |D|}{8L'^4}}$

$$
\max_{p \in \mathcal{P}} |R^{'}(\Phi(p, \cdot)) - \hat{R}^{'}(\Phi(p, \cdot))| \leq \kappa \text{ (since } \max_{p_j \in \mathcal{C}_1} |R^{'}(\Phi(p_j, \cdot)) - \hat{R}^{'}(\Phi(p_j, \cdot))| \leq \kappa/2 \text{ )}
\tag{120}
$$

We bound $N_\eta(\mathcal{P})2e^{-\frac{\kappa^2 |D|}{8L'^4}} \leq \delta$ and solve for bound on $|D|$ to get

$$
\begin{aligned}
|D| &\geq \frac{8L'^4}{\kappa^2} \log\left(\frac{2N_\eta(\mathcal{P})}{\delta}\right) \\
|D| &\geq \frac{8L'^4}{\kappa^2}\left[k \log\left(\frac{8C'\sqrt{Pk}}{\kappa}\right) + \log\left(\frac{2}{\delta}\right)\right]
\end{aligned}
\tag{121}
$$

Therefore, if condition in equation 121 holds, then event $A$ occurs. If event $A$ occurs, then $\mathcal{S}^{\text{IV}}(\epsilon) \subseteq \hat{\mathcal{S}}^{\text{IV}}(\epsilon + \kappa) \subseteq \mathcal{S}^{\text{IV}}(\epsilon + 2\kappa)$.

$\Phi(p^*, \cdot)$ is a solution to IRM equation 5 and it satisfies $\forall \Phi(p, \cdot) \in \mathcal{S}^{\text{IV}}(\epsilon) \ R(\Phi(p^*)) \leq R(\Phi(p, \cdot))$.

Define an event $B$: $\{D : \forall p \in \mathcal{P}, |R(\Phi(p, \cdot)) - \hat{R}(\Phi(p, \cdot))| \leq \frac{\nu}{2}\}$.

If event $B$ occurs, then for a solution $\Phi(\hat{p}, \cdot)$ of equation 6 (where $\epsilon$ is replaced with $\epsilon + \kappa$) satisfies

$$
\begin{aligned}
R(\Phi(\hat{p}, \cdot)) - \frac{\nu}{2} &\leq \hat{R}(\Phi(\hat{p}, \cdot) \leq \hat{R}(\Phi(p^*, \cdot)) \leq R(\Phi(p^*, \cdot)) + \frac{\nu}{2} \\
R(\Phi(\hat{p}, \cdot)) &\leq R(\Phi(p^*, \cdot)) + \nu
\end{aligned}
\tag{122}
$$

Using the covering number (from Lemma 5) we construct a minimum cover of size $b = N_\eta(\mathcal{P})$ with points $\mathcal{C}_1 = \{p_j\}_{j=1}^b$.

Let us bound the probability of failure at one point $p_j$ in the cover

$$
\mathbb{P}\Big[|R(\Phi(p_j, \cdot)) - \hat{R}(\Phi(p_j, \cdot))| > \frac{\nu}{4}\Big] < 2e^{-\frac{\nu^2 |D|}{32L^2}}
\tag{123}
$$

We use union bound to bound the probability defined as

$$
\mathbb{P}\Big[\Big\{D : \max_{p_j \in \mathcal{C}_1} |R(\Phi(p_j, \cdot)) - \hat{R}(\Phi(p_j, \cdot))| > \frac{\nu}{4}\Big\}\Big] < N_\eta(\mathcal{P})2e^{-\frac{\nu^2 |D|}{32L^2}}
\tag{124}
$$

Now consider any $p \in \mathcal{P}$ and suppose $p_j$ is nearest point to it in the cover.

$$
\begin{aligned}
&|R(\Phi(p, \cdot)) - \hat{R}(\Phi(p, \cdot))| = \\
&|R(\Phi(p, \cdot)) - R(\Phi(p_j, \cdot)) + R(\Phi(p_j, \cdot)) - \hat{R}(\Phi(p_j, \cdot)) + \hat{R}(\Phi(p_j, \cdot)) - \hat{R}(\Phi(p, \cdot))| \\
&\leq |R(\Phi_j) - \hat{R}(\Phi_j)| + 2\eta C
\end{aligned}
\tag{125}
$$

Therefore, for each $p \in \mathcal{P}$

$$
\begin{aligned}
|R(\Phi(p, \cdot)) - \hat{R}(\Phi(p_j, \cdot))| &\leq \max_{p_j \in \mathcal{C}_1} |R(\Phi(p_j, \cdot)) - \hat{R}(\Phi(p_j, \cdot))| + 2\eta C \\
\max_{p \in \mathcal{P}} |R(\Phi(p, \cdot)) - \hat{R}(\Phi(p, \cdot))| &\leq \max_{p_j \in \mathcal{C}_1} |R(\Phi(p_j, \cdot)) - \hat{R}(\Phi(p_j, \cdot))| + 2\eta C
\end{aligned}
\tag{126}
$$

Set $\eta = \frac{\nu}{8C}$ in equation 124 and from equation 126 with probability at least $1 - N_\eta(\mathcal{P})2e^{-\frac{\nu^2 |D|}{32L'^4}}$

$$
\max_{p \in \mathcal{P}} |R(\Phi(p, \cdot)) - \hat{R}'(\Phi(p, \cdot))| \leq \nu/2 \text{ (since } \max_{p_j \in \mathcal{C}_1} |R'(\Phi(p_j, \cdot)) - \hat{R}'(\Phi(p_j, \cdot))| \leq \nu/4 \text{)}
\tag{127}
$$

We bound $N_\eta(\mathcal{P})2e^{-\frac{\nu^2 |D|}{32L^2}} \leq \delta$ and solve for bound on $|D|$ to get

$$
\begin{aligned}
|D| &\geq \frac{32L^2}{\nu^2} \log\Big(\frac{2N_\eta(\Phi)}{\delta}\Big) \\
|D| &\geq \frac{32L^2}{\nu^2} \Big[k \log\Big(\frac{16C\sqrt{Pk}}{\nu}\Big) + \log(\frac{2}{\delta})\Big]
\end{aligned}
\tag{128}
$$

Therefore, if condition in equation 128 holds, then with probability at least $1 - \frac{\delta}{2}$ event $A$ occurs.

Also, if

$$
|D| \geq \max\Big\{\frac{32L^2}{\nu^2}\Big[k \log\Big(\frac{16C\sqrt{Pk}}{\nu}\Big) + \log\Big(\frac{2}{\delta}\Big)\Big], \frac{8L'^4}{\kappa^2}\Big[k \log\Big(\frac{8C'\sqrt{Pk}}{\kappa}\Big) + \log\Big(\frac{2}{\delta}\Big)\Big]\Big\}
\tag{129}
$$

then the event $A \cap B$ occurs with at least $1 - \delta$ probability.

Project $\Phi(\hat{p}, \cdot) \in \hat{\mathcal{S}}^{\mathsf{IV}}(\epsilon + \kappa)$ on $\mathcal{S}^{\mathsf{IV}}(\epsilon)$, i.e., find the closest function in terms of the metric dis, to obtain $\Phi(\tilde{\hat{p}}, \cdot)$. If event $A$ occurs, then $\hat{p} \in \mathcal{S}^{\mathsf{IV}}(\epsilon + 2\kappa)$. The distance $\|\hat{p} - \tilde{\hat{p}}\| \leq \mathsf{dis}(2\kappa)$.

$$|R(\Phi(\hat{p}, \cdot)) - R(\Phi(\tilde{\hat{p}}, \cdot))| \leq C\|\hat{p} - \tilde{\hat{p}}\| \leq C\mathsf{dis}(2\kappa) \tag{130}$$

We choose $\kappa_0$ such that $\kappa < \kappa_0$ (use Assumption 15) such that $C\mathsf{dis}(2\kappa) \leq \nu$.

$$R(\Phi(p^*, \cdot) \leq R(\Phi(\tilde{\hat{p}}, \cdot)) \leq R(\Phi(\hat{p}, \cdot)) + \nu \tag{131}$$

Therefore, by combining equation 122 amd equation 131, we can conclude that if event $A \cap B$ occurs, then $\Phi(\hat{p}, \cdot) \in \hat{\mathcal{S}}^{\mathsf{IV}}(\epsilon + 2\kappa)$ and $|R(\Phi(p^*, \cdot)) - R(\Phi(\hat{p}, \cdot))| \leq \nu$. From the conditions on $|D|$, we know that $A \cap B$ occurs with probability $1 - \delta$. We substitute $\Phi(p^*, \cdot)$ as $\Phi^*$ and $\Phi(\hat{p}, \cdot)$ as $\hat{\Phi}$ and this completes the proof. $\qquad \square$

## D. OOD Performance: Covariate shift case

In this section, we discuss the extension of Proposition 4 to the infinite hypothesis class case.

Define $D_1^* = \left\{ \frac{32L^2}{\nu^2} \left[ k \log \left( \frac{32C\sqrt{Pk}}{\nu} \right) + \log \left( \frac{4}{\delta} \right) \right], \frac{16L'^4}{\epsilon^2} \log \left( \frac{4}{\delta} \right) \right\}$.

Define $D_2^* = \frac{32L^2}{\nu^2} \left[ k \log \left( \frac{16C\sqrt{Pk}}{\nu} \right) + \log \left( \frac{2}{\delta} \right) \right]$.

**Proposition 16.** *If Assumptions 4, 13 hold, $m \in \mathcal{H}_\Phi$ and if the number of samples $|D|$ is greater than $D_1^*$, then with a probability at least $1 - \delta$, every solution $\hat{\Phi}$ to EIRM satisfies $R(m) \leq R(\hat{\Phi}) \leq R(m) + \nu$.*

*If Assumptions 4, 13 hold, $m \in \mathcal{H}_\Phi$ and if the number of samples $|D|$ is greater than $D_2^*$, then with a probability at least $1 - \delta$ every solution $\Phi^\dagger$ of ERM satisfies $R(m) \leq R(\Phi^\dagger) \leq R(m) + \nu$.*

*Proof.* We begin with the first part. Following the proof of Proposition 5, our goal is to compute the probability of event $A$: $\{\forall p \in \mathcal{P}, |\hat{R}(\Phi(p, \cdot)) - R(\Phi(p, \cdot))| \leq \frac{\nu}{2}\}$. Using the covering number (from Lemma 5) we construct a minimum cover of size $b = N_\eta(\mathcal{P})$ with points $\mathcal{C} = \{p_j\}_{j=1}^b$.

Compute the probability of failure at one point $p_j$ in the cover

$$\mathbb{P}\left[ \left\{ D : |R(\Phi(p_j, \cdot)) - \hat{R}(\Phi(p_j, \cdot))| > \frac{\nu}{4} \right\} \right] < 2e^{-\frac{\nu^2 |D|}{32L^2}} \tag{132}$$

We use union bound to bound the probability of failure over the cover $\mathcal{C}$

$$\mathbb{P}\left[ \left\{ D : \max_j |R(\Phi(p_j, \cdot)) - \hat{R}(\Phi(p_j, \cdot))| > \frac{\nu}{4} \right\} \right] < N_\eta(\mathcal{P}) 2e^{-\frac{\nu^2 |D|}{32L^2}} \tag{133}$$

Now consider any $p \in \mathcal{P}$ and suppose $p_j$ is nearest point to it in the cover.

$$\begin{aligned} &|R(\Phi(p, \cdot)) - \hat{R}(\Phi(p, \cdot))| = \\ &|R(\Phi(p, \cdot)) - R(\Phi(p_j, \cdot)) + R(\Phi(p_j, \cdot)) - \hat{R}(\Phi(p_j, \cdot)) + \hat{R}(\Phi(p_j, \cdot)) - \hat{R}(\Phi(p, \cdot))| \\ &\leq |R(\Phi(p_j, \cdot) - \hat{R}(\Phi(p_j, \cdot))| + 2\eta C \end{aligned} \tag{134}$$

In the above simplification, we used the Lipschitz continuity of $R$. Therefore

$$\begin{aligned} \forall p \in \mathcal{P}, |R(\Phi(p, \cdot)) - \hat{R}(\Phi(p, \cdot))| &\leq \max_{p_j \in \mathcal{C}} |R(\Phi(p_j, \cdot)) - \hat{R}(\Phi(p_j, \cdot))| + 2\eta C \\ \max_{p \in \mathcal{P}} |R(\Phi(p, \cdot)) - \hat{R}(\Phi(p, \cdot))| &\leq \max_{p_j \in \mathcal{C}} |R(\Phi(p_j, \cdot)) - \hat{R}(\Phi(p_j, \cdot))| + 2\eta C \end{aligned} \tag{135}$$

Set $\eta = \frac{\nu}{8C}$ in equation 135 and from equation 133 with probability at least $1 - N_\eta(\mathcal{P})2e^{-\frac{\nu^2 n}{32L'^4}}$

$$\max_{p \in \mathcal{P}} |R^{'}(\Phi(p, \cdot)) - \hat{R}^{'}(\Phi(p, \cdot))| \leq \nu/2 \text{ (since } \max_{p_j \in \mathcal{C}} |R^{'}(\Phi(p_j, \cdot)) - \hat{R}^{'}(\Phi(p_j, \cdot))| \leq \nu/4 \text{ )} \quad (136)$$

We bound $N_\eta(\mathcal{P})2e^{-\frac{\nu^2 |D|}{32L^2}} \leq \delta$ and solve for bound on $|D|$ to get

$$|D| \geq \frac{8L^2}{\frac{\nu^2}{4}} \log\left(\frac{N_\eta(\mathcal{P})}{\delta}\right) \text{ (Use Lemma 5)}$$

$$|D| \geq \frac{32L^2}{\nu^2}\left[k \log\left(\frac{16C\sqrt{Pk}}{\nu}\right) + \log\left(\frac{2}{\delta}\right)\right] \quad (137)$$

Therefore, if condition in equation 137 holds, then event $A$ occurs. Observe that the optimal solution $p^*$ for expected risk minimization $p^* \in \arg\min_{p \in \mathcal{P}} R(\Phi(p, \cdot))$ satisfies $\Phi(p^*, \cdot) = m(\cdot)$ (From Lemma 4 and $m \in \mathcal{H}_\Phi$). If event $A$ occurs, then from same argument used in equation 48, a solution $p^+ \in \mathcal{P}$ of ERM satisfies $R(\Phi(p^*, \cdot)) \leq R(\Phi(p^+, \cdot)) \leq R(\Phi(p^*, \cdot)) + \nu$ is true.

We now move to the second part. From the first part of the proof, we conclude that when

$$|D| \geq \frac{32L^2}{\nu^2}\left[k \log\left(\frac{16C\sqrt{Pk}}{\nu}\right) + \log(\frac{4}{\delta})\right] \quad (138)$$

with a probability at least $1 - \frac{\delta}{2}$ event $A$ occurs.

Define an event $B$: $\tilde{D}$ is such that $|\hat{R}^{'}(m) - R^{'}(m)| \leq \frac{\epsilon}{2}$. Since $R^{'}(m) = 0$, $|\hat{R}^{'}(m) - R^{'}(m)| \leq \frac{\epsilon}{2} \implies |\hat{R}^{'}(m)| \leq \frac{\epsilon}{2} \implies \hat{R}^{'}(m) \leq \frac{\epsilon}{2}$. Therefore, $m \in \hat{\mathcal{S}}^{\mathsf{IV}}(\epsilon)$.

We write $\mathbb{P}(B) = \mathbb{P}(\{\tilde{D} : |\hat{R}^{'}(m) - R^{'}(m)| \leq \frac{\epsilon}{2}\}) = 1 - \mathbb{P}(\{\tilde{D} : |\hat{R}^{'}(m) - R^{'}(m)| > \frac{\epsilon}{2}\})$. The gradient of loss function is bounded $|\frac{\partial \ell(\Phi(\cdot), \cdot)}{\partial w}|_{w=1.0}| \leq L^{'}$. From Hoeffding's inequality in Lemma 1 it follows that

$$\mathbb{P}(|\tilde{D} : \hat{R}^{'}(h) - R^{'}(h)| > \frac{\epsilon}{2}) \leq 2\exp\left(-\frac{|D|\epsilon^2}{16L'^4}\right)$$

$$2\exp\left(-\frac{|D|\epsilon^2}{16L'^4}\right) \leq \frac{\delta}{2}$$

$$|D| \geq \frac{16L'^4}{\epsilon^2}\log\left(\frac{4}{\delta}\right) \quad (139)$$

Combining the two conditions equation 138 and equation 139,

$$|D| \geq \max\left\{\frac{32L^2}{\nu^2}\left[k \log\left(\frac{32C\sqrt{Pk}}{\nu}\right) + \log\left(\frac{4}{\delta}\right)\right], \frac{16L'^4}{\epsilon^2}\log\left(\frac{4}{\delta}\right)\right\}$$

This ensures $P(A \cap B) \geq 1 - \delta$. If event $A \cap B$ occurs, then we follow the same justification as in the proof of Proposition 4 to claim that a solution $\hat{p} \in \mathcal{P}$ to EIRM equation 6 satisfies $R(\Phi(p^*, \cdot)) \leq R(\Phi(\hat{p}, \cdot)) \leq R(\Phi(p^*, \cdot)) + \nu$

This completes the proof.

$\square$

### 7.4.3 EXTENSIONS TO BINARY CLASSIFICATION (CROSS-ENTROPY)

In the main body of the manuscript, we focused on regression (square-loss). In this section, we discuss the results that can be extended to binary classification (cross-entropy) loss. We will not go in the order in which the results were introduced in the manuscript but in an order that makes for easier exposition for the classification case.

We begin by showing how to extend Proposition 4 to binary classification (cross-entropy). Recall that the entropy of a distribution $\mathbb{P}_X$ is $H(\mathbb{P}) = -\mathbb{E}_{\mathbb{P}}\big[\log(d\mathbb{P})\big]$. Recall that the cross entropy of $\mathbb{Q}$ relative to $\mathbb{P}$ is $H(\mathbb{P}, \mathbb{Q}) = -\mathbb{E}_{\mathbb{P}}\big[\log(d\mathbb{Q})\big] = H(P) + \mathsf{KL}(d\mathbb{P}\|d\mathbb{Q})$. The cross entropy loss $\ell$ for binary classification when using a predictor $f : \mathcal{X} \to [0, 1]$ ($f(X^e)$ is the probability of label 1 conditional on $X^e$) is given as $\ell\big(f(X^e), Y^e\big) = Y^e \log\big(f(X^e)\big) + (1 - Y^e) \log\big(1 - f(X^e)\big)$. For the discussion below $\mathbb{Q}(Y^e|X^e)$ is defined in terms of $f$ as follows $\mathbb{Q}(Y^e = 1|X^e) = f(X^e)$ $\big(\mathbb{Q}(Y^e = 0|X^e) = 1 - f(X^e)\big)$.

$$
\begin{aligned}
R^e(f) &= \mathbb{E}^e\big[\ell(Y^e, f(X^e))\big] \\
&= \mathbb{E}^e\Big[Y^e \log\big(f(X^e)\big) + (1 - Y^e) \log\big(1 - f(X^e)\big)\Big] \\
&= \mathbb{E}^e\Big[\mathbb{E}^e\big[Y^e|X^e\big] \log\big(f(X^e)\big) + \big(1 - \mathbb{E}\big[Y^e|X^e\big]\big) \log\big(1 - f(X^e)\big)\Big] \\
&= \mathbb{E}^e\Big[\mathbb{P}(Y^e = 1|X^e) \log\big(f(X^e)\big) + \mathbb{P}(Y^e = 0|X^e) \log\big(1 - f(X^e)\big)\Big] \quad (140) \\
&= \mathbb{E}^e\Big[H\big(\mathbb{P}(Y^e|X^e), \mathbb{Q}(Y^e|X^e)\big)\Big] \\
&= \mathbb{E}^e\Big[H\big(\mathbb{P}(Y^e|X^e)\big) + \mathsf{KL}\big(\mathbb{P}(Y^e|X^e)\|\mathbb{Q}(Y^e|X^e)\big)\Big] \\
&= \mathbb{E}^e\Big[H\big(\mathbb{P}(Y^e|X^e)\big)\Big] + \mathbb{E}^e\Big[\mathsf{KL}\big(\mathbb{P}(Y^e|X^e)\|\mathbb{Q}(Y^e|X^e)\big)\Big]
\end{aligned}
$$

From the above it is clear that $\mathbb{Q}(Y^e|X^e) = \mathbb{P}(Y^e|X^e)$ minimizes the risk in an individual environment.

**Assumption 16.** *Invariance w.r.t all the features.* *For all $e, o \in \mathcal{E}_{all}$ and for all $x \in \mathcal{X}$, $\mathbb{E}[Y^e|X^e = x] = \mathbb{E}[Y^o|X^o = x]$. $X^e \sim \mathbb{P}^e_{X^e}$ and $\forall e \in \mathcal{E}_{all}$ support of $\mathbb{P}^e_{X^e}$ is equal to $\mathcal{X}$.*

Observe that in the binary-classification setting the above assumption amounts to equating the conditional probabilities $\mathbb{P}(Y^e|X^e)$ and $\mathbb{P}(Y^o|X^o)$.

Recall that map $m$ (from equation 2) simplifies to $\forall x \in \mathcal{X}$

$$
m(x) = \mathbb{E}^e[Y^e|X^e = x] = \mathbb{P}(Y^e = 1|X^e = x) \quad (141)
$$

If Assumption 16 holds, then from cross-entropy decomposition in equation 140 it is clear that $m$ solves the OOD problem (as it is optimal w.r.t each environment). It is also the unique minimizer. We can justify it based on the same argument presented in equation 56. Suppose there was another optimizer which was different from $m$ over a set with a non-zero measure. Over such a set the the KL divergence term inside equation 140 will be greater than zero, thus making the second term in equation 140 positive thus contradicting the optimality. This shows $m$ is the unique optimizer. The rest of the arguments presented in the proof of Proposition 4 carry over to this case. Therefore, Proposition 4 extends to the cross-entropy loss.

Note that Proposition 2's proof was agnostic to loss type and only used boundedness, which holds for both cross-entropy as long as the probability output are in the strict interior of $[0, 1]$ defined by $[\mathsf{p_{min}}, \mathsf{p_{max}}] \subset [0, 1]$. We could not generalize Proposition 5 to cross-entropy loss and that is left as future work.

Next, we move to showing how Proposition 1 can be generalized to binary classification.

**Assumption 17.** *Existence of an invariant representation.* $\exists \Phi^* : \mathcal{X} \to \mathcal{Z}$ *such that* $\forall e, o \in \mathcal{E}_{all}$ *and* $\forall x \in \mathcal{X}$, $\mathbb{E}[Y^e|\Phi^*(x)] = \mathbb{E}[Y^o|\Phi^*(x)]$.

Recall $m$ defined in equation 2, $\forall z \in \Phi^*(\mathcal{X})$

$$
m(z) = \mathbb{E}^e[Y^e|Z^e = z] = \mathbb{P}(Y^e = 1|X^e = z) \quad (142)
$$

Define a composite predictor $w \circ \Phi^*$. Substituting $f = w \circ \Phi^*$ in equation 140 we get the following. For the discussion below, a distribution $\mathbb{R}(Y^e|X^e)$ is defined in terms of $w \circ \Phi^*$ as follows $\mathbb{R}(Y^e = 1|X^e) = w \circ \Phi^*(X^e)$ $(\mathbb{R}(Y^e = 0|X^e) = 1 - w \circ \Phi^*(X^e))$.

$$
R^e(w \circ \Phi^*) = \mathbb{E}^e\Big[H\big(\mathbb{P}(Y^e|Z^e)\big)\Big] + \mathbb{E}^e\Big[\mathsf{KL}\big(\mathbb{P}(Y^e|Z^e)\|\mathbb{R}(Y^e|(Z^e))\big)\Big] \quad (143)
$$

If all the data is transformed by $\Phi^*$, then from the above decomposition equation 143 it is clear that $\mathbb{R}(Y^e|Z^e) = \mathbb{P}(Y^e|Z^e)$ is the optimal predictor for each environment. Hence, $w^*(Z^e) = \mathbb{P}(Y^e = 1|Z^e)$ is the best choice for $w$.

**Assumption 18.** *Existence of an environment where the invariant representation is sufficient.* $\exists$ *an environment $e \in \mathcal{E}_{all}$ such that $Y^e \perp X^e | Z^e$, where $Z^e = \Phi^*(X^e)$.*

We derive a relationship as follows for the environment $q$ satisfying Assumption 18.

$$\mathbb{P}(Y^q|Z^q, X^q) = \mathbb{P}(Y^q|Z^q) \text{ (follows from conditional independence in Assumption 18)} \quad (144)$$

Also, note that since $Z^q = \Phi^*(X^q)$ we have

$$\mathbb{P}(Y^q|Z^q, X^q) = \mathbb{P}(Y^q|X^q) \quad (145)$$

From equation 144 and equation 145 we have

$$\mathbb{P}(Y^q|X^q) = \mathbb{P}(Y^q|Z^q) \quad (146)$$

We use equation 146 in the cross entropy decomposition from equation 140

$$R^q(f) = \mathbb{E}^q\Big[H\big(\mathbb{P}(Y^q|Z^q)\big)\Big] + \mathbb{E}^q\Big[\mathsf{KL}\big(\mathbb{P}(Y^q|Z^q)\|\mathbb{Q}(Y^q|X^q)\big)\Big] \quad (147)$$

Recall $\mathbb{Q}(Y^q = 1|X^q) = f(X^q)$ ($\mathbb{Q}(Y^q = 0|X^q) = 1 - f(X^q)$). Also, recall $w^*(Z^q) = \mathbb{P}(Y^q = 1|Z^q)$. From the above it is clear that $f = w^* \circ \Phi^*$ is the optimal predictor for environment $q$.

$$R^q(w^* \circ \Phi^*) = \mathbb{E}^q\Big[H\big(\mathbb{P}(Y^q|X^q)\big)\Big] \quad (148)$$

The expected conditional entropy for environment $e$ is defined as $\bar{H}^e = \mathbb{E}^e\Big[H\big(\mathbb{P}(Y^e|Z^e)\big)\Big]$ is the risk achieved by $w^* \circ \Phi^*$. Also, $\bar{H}^e$ measures the amount of noise in the environment. This is much like the variance that remains in the least squares minimization. In the next assumption, we state that the noise in all the environments is bounded above. We also assume that one of the environments which achieves the maximum noise level is environment $q$, which satisfies Assumption 18.

**Assumption 19.** $\forall e \in \mathcal{E}_{all}, \bar{H}^e \leq \bar{H}^{\mathsf{sup}}, \bar{H}^q = \bar{H}^{\mathsf{sup}}$

Therefore,

$$R^q(w^* \circ \Phi^*) = \bar{H}^{\mathsf{sup}} \quad (149)$$

From equation 143 for all the environments

$$R^e(w^* \circ \Phi^*) = \bar{H}^e \quad (150)$$

Observe that $\max_{e \in \mathcal{E}_{all}} R^e(w^* \circ \Phi^*) = \bar{H}^{\mathsf{sup}}$. From the above assumption it is clear that for all predictors $f : \mathcal{X} \to [0,1]$.

$$\forall f, \max_{e \in \mathcal{E}_{all}} R^e(f) \geq R^q(f) \geq \bar{H}^{\mathsf{sup}}$$
$$\min_f \max_{e \in \mathcal{E}_{all}} R^e(f) \geq \bar{H}^{\mathsf{sup}} \quad (151)$$

Since $\max_{e \in \mathcal{E}_{all}} R^e(w^* \circ \Phi^*) = \bar{H}^{\mathsf{sup}}$, we conclude that $w^* \circ \Phi^*$ is the predictor that solves the OOD problem in equation 1. This completes the extension of Proposition 1 to cross-entropy.

### 7.4.4 ON THE BIASEDNESS OF ERM

Consider the model in Assumption 5. For each environment $e \in \mathcal{E}_{tr}$, define a vector $\rho^e = \mathbb{E}^e\big[\varepsilon^e X^e\big]$. Define a matrix $\bar{\rho}$ with $\rho^e$ as column vectors $\bar{\rho} = [\rho^1, \ldots, \rho^{|\mathcal{E}_{tr}|}]$. Define a vector $\bar{\pi} = [\pi^1, \ldots, \pi^{|\mathcal{E}_{tr}|}]$, where recall from Assumption 5 $\pi^o$ is probability a point comes from environment $o$.

**Proposition 17.** *If Assumption 5 holds, $\mathcal{H}_\Phi$ is a linear hypothesis class with parameter $\Phi$ and if the rank of $\bar{\rho}$ is at least one, then ERM is asymptotically biased, i.e., even with infinite data ERM will not achieve the desired solution $\tilde{\mathcal{S}}^\mathsf{T}\gamma$, except over a set of measure zero of probability distributions $\bar{\pi}$.*

*Proof.* Consider the case when ERM has access to infinite data, i.e., we are solving the expected risk minimization problem stated as $\min_{\Phi \in \mathcal{H}_\Phi} R(\Phi)$. We will consider the linear model in Assumption 5 and assume $\mathcal{H}_\Phi$ linear hypothesis class parametrized by $\Phi \in \mathbb{R}^n$. We simplify the $\nabla_\Phi R(\Phi)$ for the square loss below

$$\nabla_\Phi R(\Phi) = \sum_{e \in \mathcal{E}_{tr}} \pi^e \mathbb{E}^e \left[ \left( Y^e - \Phi^\mathsf{T} X^e \right) X^e \right] \tag{152}$$

We compute the gradient for $\Phi = \tilde{\mathcal{S}}^\mathsf{T} \gamma$ as

$$\nabla_{\Phi | \Phi = \tilde{\mathcal{S}}^\mathsf{T} \gamma} R(\Phi) = \sum_{e \in \mathcal{E}_{tr}} \pi^e \mathbb{E}^e \left[ \left( Y^e - \gamma^\mathsf{T} \tilde{\mathcal{S}} X^e \right) X^e \right] \text{ (Use Assumption 5)}$$
$$= \sum_{e \in \mathcal{E}_{tr}} \pi^e \mathbb{E}^e \left[ \varepsilon^e X^e \right] \tag{153}$$

Recall $\rho^e = \mathbb{E}^e \left[ \varepsilon^e X^e \right]$ and $\bar{\rho} = [\rho^1, \ldots, \rho^{|\mathcal{E}_{tr}|}]$. Recall $\bar{\pi} = [\pi^1, \ldots, \pi^{|\mathcal{E}_{tr}|}]$. Setting the gradient defined in equation 153 to zero and using the above matrix notation we get

$$\bar{\rho} \bar{\pi} = 0, \mathbf{1}^\mathsf{T} \bar{\pi} = 1, \bar{\pi} \geq 0 \tag{154}$$

If $\bar{\pi}$ satisfies equation 154, then ERM is unbiased, else it is not. Consider the set of vectors in the probability simplex $\{\bar{\pi} \mid \mathbf{1}^\mathsf{T} \bar{\pi} = 1, \bar{\pi} \geq 0\}$ and define a uniform probability distribution over it. Since rank of $\bar{\rho} > 0$ at least one of the columns of $\bar{\rho}$ is non-zero. As a result a uniform random draw from this set of probablity distributions would have zero probability of satisfying $\bar{\rho} \bar{\pi} = 0$. Therefore, $\tilde{\mathcal{S}}^\mathsf{T} \gamma$ is not the optimal solution to ERM and thus the solution of ERM would be biased away from $\tilde{\mathcal{S}}^\mathsf{T} \gamma$.

$\square$

In Proposition 5, we had assumed Assumptions 5, 6, 7 hold. If we also assume that rank of $\bar{\rho}$ is at least one, the Proposition 5 continues to hold. If for at least one $e \in \mathcal{E}_{tr}$, $\mathbb{E}^e \left[ \varepsilon^e X^e \right]$ is non-zero, then the rank of $\bar{\rho}$ is at least one. From proof of Theorem 10 in Arjovsky et al. (2019), linear general position continues to hold except over a set of covariance matrices with measure zero even when one of the $\mathbb{E}^e \left[ \varepsilon^e X^e \right]$ is non-zero.

Also, in the above Proposition 17, we only required that rank of $\bar{\rho}$ is at least one. However, if we make the additional assumptions 5, 6, 7, the result of the above Proposition continues to hold. Therefore, if Assumption 5, 6, 7 hold and rank of $\bar{\rho}$ is at least one, then ERM is asymptotically biased and IRM can be within $\sqrt{\epsilon}$ neighborhood of the ideal solution with the sample complexity shown in Proposition 5.

