# OpenReview forum: "Empirical or Invariant Risk Minimization? A Sample Complexity Perspective"
_ICLR.cc/2021/Conference — ICLR 2021 Poster_

### Official Review · AnonReviewer4 · 2020-10-25

**Rating:** 6
**Confidence:** 2

**Review:**

Summary:

The authors provide sample complexity study of the Invariant Risk Minimization (IRM) in the case of scalar predictions and one dimensional representations. Instead of considering the penalized version of IRMv1 proposed by **Arjovsky et al, 2019**, the authors study its constrained counterpart introducing a relaxation into the requirement of zero gradient. The main conclusion that I understood from the work is: the performance of IRM (provided the assumptions are satisfied) will never be much worse than that of ERM and will be superior to that of ERM in some particular cases. The authors support their theoretical discoveries empirically.

Review:

The paper is rather dense and the exposition could be improved. I had some troubles in Section 3.3.2 since the spaces of the variables and matrices are not introduced (but it *might* be standard in causality literature, which I am not familiar with).
It is difficult to judge the significance of the work. On the one hand, the proofs of most of the results follow fairly standard tools of empirical process theory and I am not sure how much more insights this work provides compared to that of Arjovsky et al, 2019. On the other hand,  the statements themselves seem to be novel and they fit well in the big picture.


Pros: The theoretical results on IRM are novel and the authors introduce new variants of colored MNIST.
Cons:  The proofs follow the same pattern as that for ERM. The fact that IRM is constrained does not bring much novelty in the proof. See for instance: **Woodworth et al, Learning Non-Discriminatory Predictors, 2017** and **Agarwal et al, A Reductions Approach to Fair Classification, 2018** for similar proof techniques. It is also unfortunate that the main body refers too often to supplementary material for some details that in my opinion are very relevant to the main body. For instance, empirical version of the gradient, additional discussion on why ERM is biased in Section 3.3.2, and the setup of the experiments are all postponed to appendix.

Questions:
1. On feasibility: I wonder whether there is an issue of feasibility for the problem in Eq. (5) and Eq. (6)? Is it obvious that if the set $\mathcal{H}_{\Phi}$ is finite then we can always find a feasible representation?
2. Slack variables: what is the reason to introduce slack variable $\epsilon$ in the formulation of Eq. (5)? It seems to me that it is sufficient to introduce it only for Eq. (6), since the authors only need to show that $\Phi^*$ is feasible (with high probability) for Eq. (6) in order to prove their upper bound. I also do not see why $\kappa$, introduced on p.5, cannot be zero.

Suggestion:
The phrase **''``For ease of exposition, we use the standard setting of finite hypothesis class and extend all the results to infinite hypothesis classes in the supplement''** is not convincing. Finite class is not that realistic in practice (correct me if I am wrong). If all the results of this work can be stated for infinite classes of bounded capacity (e.g. VC-class or totally bounded spaces), I suggest to completely erase the case of finite classes.


After rebuttal: I thank the authors for comprehensive rebuttal, which addressed most of my concerns. I update the score accordingly.

---

> ### Author Response · Authors · 2020-11-14
> **Response to Reviewer 4 (Part 1)**
>
> We are very thankful to Reviewer 4 for the time and effort. We have tried our best to incorporate your suggestions into the revised draft. We highlight all the changes in the revised paper in blue. Below we address the main concerns. We also look forward to hearing your feedback on the changes and our responses.
> 1. **On exposition:**  The reviewer raised concern about exposition. We hope that the following steps will alleviate those concerns.
>
>     - **A new illustrative example, figure and improved presentation:** We have added an illustrative example with a figure to describe the graphical model to describe the difference between Propositions 4 and 5. We also use this figure early on (Figure 1 in the revision) in the introduction to give a reader a simple example of the different types of distribution shifts. We have improved the presentation of the implications of Proposition 5, which we hope will help the reader.
>     - **A high-level dialogue:** At the beginning of the supplement, we have a high-level dialogue between two students (inspired from original work IRM) around existing works on IRM and how our work fits in the big picture.
> 2. **On the novelty of proofs:**  We would like to highlight key challenges that appear because of being in an IRM setting and how we dealt with them. Note these challenges are specific to the problem of IRM and do not appear in the papers on fairness that were pointed to.
>
>     **Key challenges and technical novelties**
>     - **IRM penalty is not separable:** The IRM penalty $R^{'}$ is a weighted sum of squares of norm of the gradient of the expected loss. Now since the population value (the gradient of the expected loss) is composed with a squared function, the penalty cannot be expressed as a separable sum over the data points. We need separability to be able to use standard concentration inequalities. We propose an alternate way of expressing the penalty that allows us to build a separable estimator.
>     -**Standard learning theory bounds deal with concentration around risk and not a target model itself.** It is common in learning theory to analyze the concentration of empirical risk around the expected risk. This does not guarantee that the two approaches -- empirical and expected -- actually find the same risk minimizer. In contrast in our case, we have a target model, $m\circ \Phi^{*}$ , that solves equation (1) and we want to be able to guarantee that empirical solutions concentrate around that. This being a new task in itself required us to do a new analysis (see the proof of Proposition 5).  This being a challenging task, we decided to pursue linear and polynomial generative models in this work.  We  exploit the linear general position assumption (which is a condition derived in Arjovsky et al 2019 especially for the IRM problem) along with new estimator we proposed for IRM penalty to connect concentration around risks with concentration around target model.
>
>     -**Significance and novelty of the results:** While we believe that there is sufficient novelty in our proof techniques (as described above), we also believe that the question that we are answering is very important with our work providing novel insights. Out-of-distribution generalization is one of the biggest challenges in machine learning and understanding issues of sample complexity for such setups is quite important. The comparison between ERM and IRM is actively debated in machine learning https://arxiv.org/pdf/2007.01434.pdf. We believe our work is timely and can serve as a guide for developing new methods and theory related to IRM and more generally out-of-distribution generalization.
> 3. **Referring to supplement:** We have added proof sketches to the key results thus reducing the reliance to read the supplement. These sketches help a reader understand the new technical ingredients that went into the proofs.  Additionally, we would be happy to include the discussion on the biasedness of ERM from supplement to the main text if the reviewer advises so.
> 4. **About feasibility:**  $m \circ \Phi^{ *}$ is the ideal predictor as it solves equation (1). Due to Assumption 1, we can see that if $m \circ \Phi^{ *}\in \mathcal{H}_{\Phi}$,  then $m\circ \Phi^{ *}$ is an invariant predictor and thus it is a feasible solution to the constraints in equation (5). Suppose we are using square loss. Another trivial invariant predictor is when $\Phi$ maps all points to zero (this follows from the fact that scalar classifier multiplied with representation will continue to return a zero value). Thus zero map guarantees feasibility trivially but it is not an invariant predictor we want the equation (5) to output. We hope solving equation (5) outputs $m \circ \Phi^{ *}$.

---

> > ### Author Response · Authors · 2020-11-14
> > **Response to Reviewer 4 (Part 2)**
> >
> > 5. **About slack in equation (5):**  Recall that the formulation in equation (4) involves equality constraints. If we construct a finite sample approximation of equation (4) directly, then in order to satisfy the constraint in equation (4) the finite sample approximation would have to be exactly equal to the true expected value. To alleviate this issue, we introduce an intermediate problem with a slack in the form of equation (5). In our proof we first show that by introducing the solution to the slacked version equation (5) lie in the neighborhood of equation (4). After that we connect the solutions of finite sample approximation of equation (5) given in equation (6) to equation (4).
> > 6. **Interpreting $\kappa$:**    Recall $\kappa = \min_{\Phi \in \mathcal{H}_{\Phi}}|R^{'}(\Phi)-\epsilon|$. It measures how close any penalty can get to the boundary  $\epsilon$. $\kappa$ quantifies how good the finite sample approximation $\hat{R}^{'}$ need to be in order to get $\hat{\mathcal{S}}^{\mathsf{IV}}(\epsilon) = \mathcal{S}^{\mathsf{IV}}(\epsilon)$. If we make $\kappa$ zero, then that would require the approximation $\hat{R}^{'}$ to be exactly equal to the expectation $R^{'}$, which is only guaranteed in the infinite sample limit. We now describe how $\kappa>0$ in many cases maps exactly to having a non-zero slack $\epsilon>0$. As described above $m \circ \Phi^{ *}$  is an invariant predictor and thus it is a feasible solution (satisfies the constraints) to equation (3). Suppose $m \circ \Phi^{ *} \in \mathcal{H}\Phi$. $m\circ \Phi^{ *}$ is also is a feasible solution (satisfies the constraints) to equation (4) and equation (5).  Therefore, $R^{'}(m\circ \Phi^{ *})=0$.  We order the representations based on the values of $R^{'}(\Phi)$ and select the representation with the smallest non-zero $R^{'}(\Phi)$ and call it $\tilde{\Phi}$. Let us say $\epsilon$ is set to a value less than $\frac{R^{'}(\tilde{\Phi})}{2}$. In this case, $\kappa=\epsilon$ and the assumption $\kappa>0$ changes to $\epsilon>0$, which translates to saying we allow the slack in equation (5) to be arbitrarily close zero but we just do not let it become exactly zero. To summarize, for sufficiently small $\epsilon$, $\kappa$ is equal to the slack $\epsilon$ and thus we only require $\epsilon$ to be greater than zero.
> > 7. **Replacing finite with infinite hypothesis class results:** Given that our paper is already perceived as being dense, we believe that keeping the finite hypothesis results improves readability. The proofs for infinite hypothesis classes follow the same general strategy as the finite case with many more technicalities leading to more notation, which we fear will make exposition harder if moved to the main paper. However, if you feel strongly about this we will try to do it.
> > 8. **Constrained vs. Penalized version:** We present the analysis for the constrained version in the main paper. Our results also extend to the penalized version from Arjovsky et al. IRMv1 as we show the supplement.
> >
> >
> > **Reference**
> > Arjovsky, Martin, et al. "Invariant risk minimization." arXiv preprint arXiv:1907.02893 (2019).

---

### Official Review · AnonReviewer3 · 2020-10-26
**interesting yet very technical insights on ERM/IRM**

**Rating:** 7
**Confidence:** 4

**Review:**

Summary: The paper investigates the choice of learning paradigms to reach out-of-distribution generalization, namely IRM vs ERM under different scenarios of domain generalization. Technically, generalization bounds and rates are calculated to be able to compare theoretically how each paradigm fares in the different scenarios. Proposed analysis shows that IRM generalizes better than ERM in a number of important cases: presence of confounders and/or anti--causal variables. Numerical experiments on variants of the Colored MNIST benchmark (for each scenario) verify the theoretical findings.

Good points:
- focus on an important problem, and answering elements for recent debate in the scientific community e.g. https://arxiv.org/abs/2010.05761 [scientific importance]
- general insights clear and easy to grasp (Table 1) [clarity of results]
- remarkable effort to explain technical results (e.g. paragraph "implications of Proposition 4") [readability effort]

Questions:
- it seems section 3.3.2 and generally results on the confounder and anti-causal scenarios are restricted to polynomial mechanisms ($g$) ?
- in experiments with the confounder case CF-MNIST (Fig. 1b) it seesm IRM only achieves chance-level performance (=~ .45 test error) ?

Points limiting the relevance of the paper:
- overly technical paper: it is hard to follow for readers interested in general (or deep) ML but not literate in generalization theory
- density of the paper: most of the space is dedicated to technical results which albeit interesting are seemingly "shoehorned" into 8 pages. I wonder if the conference format makes justice to such a deeply technical work... 47 pages total with appendix
- difficult to check the proofs: all are in appendix (no sketch) but there are so much of them. As a reviewer I find it discouraging as the time needed to check correctness would be much more than what I can afford in a short cycle for a conference such as ICLR
- result a little over-sold in abstract/conclusion: it seems that the theoretical results are somewhat limited (see question on sec. 3.3.2) and that is omitted in both abstract and conclusion, maybe giving a false sense of generality to the results. Maybe the studied case - which is already impressive - is more general than I think but I'd like a comment of the authors about it.

Points that would improve the paper (and my score):
- clarify the meaning of limitation of result in section 3.3.2 + add it in abstract/conclusion
- comment the CF-MNIST (Fig. 1b) experiment
- add/rewrite if possible "interpretation for the layman" paragraphs (e.g. last paragraph of sec. 3)

Finally, I'd like the authors to think about the best venue to make justice to their work. It seems to me that a general ML journal would be a good option that would 1) allow more space to make the text more self-contained 2) allow more time for proof checking and discussion/optimization 3) allow more interpretative comments to be included in the work and make it accessible to a wider audience. In the mean time a preprint could also circulate to share the current research results.

also: small typo on title of Assumption 7

---

> ### Author Response · Authors · 2020-11-14
> **Response to Reviewer 3**
>
> We are very thankful to Reviewer 3 for the time and effort. We have tried our best to incorporate your suggestions into the revised draft. We highlight all the changes in the revised paper in blue. Below we address the main concerns. We also look forward to hearing your feedback on the changes and our responses.
> 1. **About polynomial $g$:** Yes, we agree that our current results are only for anti-causal and (or) confounded settings work only for the polynomial mechanisms. We believe these results can be generalized to reproducing kernel Hilbert spaces and beyond and that forms an interesting direction for future work.  We now clearly state we only solve the polynomial case in abstract, conclusion and  Section 3. We also state that other non-linear models are left to future work.  Also, we believe that the linear and polynomial structural equation models that  we consider provide useful insights into lot of forces that co-occur in real data settings.
> 2. **About CF-CMNIST:** On CF-CMNIST we achieve an error of 0.45 which is 5 percent better than a chance. However, at the same time, the error of ERM is around 0.70. In CF-CMNIST it is more challenging to resolve spurious correlations than in AC-CMNIST where IRM is able to achieve around 0.30 error.  This suggests that confounder induced spurious correlations are harder to mitigate and may need more samples than needed in AC-CMNIST. Also, at this point, it might be important to ask how to improve from $0.45$ and get to the lowest possible value of $0.25$. Recall that are two more sources of approximation. We use IRMv1 from Arjovsky et al. to generate the solutions of IRM. IRMv1 recall is a penalized approximation of equation (6). IRMv1 itself is a non-convex loss and the SGD based procedure might not be guaranteed to converge to a global minimum of IRMv1 loss. Thus if these sources of approximation can be corrected that might help further improve IRM. We have added a comment on CF-CMNIST.
> 3. **Overly technical paper:**   The reviewer raised concern that the paper is overly technical. We hope the following measures help alleviate those concerns.
>     - **A new illustrative example, figure and improved presentation:** We have added an illustrative example with a figure to describe the graphical model to describe the difference between Propositions 4 and 5. We also use this figure early on (Figure 1 in the revision) in the introduction to give a reader a simple example of the different types of distribution shifts. We have improved the presentation of the implications of Proposition 5, which we hope will help the reader.
>     - **A high-level dialogue:** At the beginning of the supplement, we have a high-level dialogue between two students (inspired from original work IRM) around existing works on IRM and how our work fits in the big picture.
>
> 4. **About proof sketches:** The reviewer would have liked to have proof sketches in the main body. We have added short proof sketches for two of the main propositions to do justice to the technical contributions of the paper.
>
> 5. **On the total page length:**  The total length of the paper maybe long but that is due to the proofs of extensions and the other supporting experiments and discussion. Currently the proofs of the main results in the paper without taking into account the extensions are 13 pages, which we believe is not an outlier for ICLR (Small Nonlinearities in Activation Functions Create Bad Minima for Neural Networks, Yun et al. ICLR 2019). Other proofs and extensions are for a reader more interested in the general case.  Also, in the final version we can also try to reduce the length of the supplement.
>
> 6. **Other venues:**  We appreciate the reviewers suggestion on a possible more technical venue for this submission without page restrictions such as a journal. However, we feel that this question of ERM and IRM comparison is of a very broad interest as many people are developing out-of-distribution generalization methods. We feel the work is very timely and ICLR audience who is working actively on IRM and other out-of-distribution generalization would benefit from this work. Moreover, ICLR in the past has published math heavy papers some of which in fact have even appeared as orals (eg. Yilun Xu et, al. A Theory of Usable Information under Computational Constraints, ICLR 2020; Keyulu Xu et, al. How Powerful are Graph Neural Networks?, ICLR 2019).
>
>
> **Reference**
> Arjovsky, Martin, et al. "Invariant risk minimization." arXiv preprint arXiv:1907.02893 (2019).

---

### Official Review · AnonReviewer1 · 2020-10-29
**Interesting theoretical insights about the good performance of IRM in the presence of distributional shifts**

**Rating:** 7
**Confidence:** 3

**Review:**

SUMMARY
#######

The present paper proposes a sample complexity study to compare Empirical Risk Minimization (ERM) and Invariant Risk Minimization (IRM).

IRM is a framework developed by Arjovsky et al. 2019 to avoid learning a predictor based on spurious correlations (e.g. color background of a photo).

Data are assumed to be sampled from different environments, and it is assumed that there exists a transformation $\Phi^*$ of the input data $X$ such that the expected value (and the variance) of the output $Y$ given $\Phi^*(X)$ is the same among all environments. If $\Phi^*$ is the identity, this assumption encompasses covariate shift.

Under the additional assumption that $\Phi^*(X)$ is sufficient on at least one environment, authors show that for the square loss $x \mapsto \mathbb{E}[Y \mid \Phi^*(X) = \Phi^*(x)]$ is a solution to the Out-of-Distribution (OOD) generalization problem (prop. 1), that aims at minimizing the maximum environment risk.

IRM is obtained by restricting the search space to invariant predictors, i.e. predictors that write as $w \circ \Phi$ such that $w$ minimizes over all $u$ the environment risk of $u \circ \Phi$ for all environments. This condition is further replaced with a gradient constraint. Empirical IRM (EIRM) is achieved by replacing expectations with empirical averages.

Authors first show the concentration of the solution to EIRM around that of IRM (prop. 2).

Then, under additional assumptions, authors study how EIRM approximates the OOD solution by exhibiting sample complexities under both covariate shift (prop. 4) and more involved distributional shifts (prop. 5). Results suggest the superiority of EIRM over ERM in the later setting.

Experiments on variants of the colored MNIST dataset are presented, matching the theoretical findings. Code is provided.



COMMENTS
########

The paper is globally well written and understandable (despite notation getting heavier and heavier with the assumptions). IRM is clearly introduced and the motivations well exposed.

The subject tackled is of great interest, and the theoretical angle adopted is novel and insightful. Experiments are eloquent.

About Asm. 1, how hard is it to check in practice? Does someone have to explicitly exhibit $\Phi^*$? Maybe an example where $\Phi^* \ne \mathsf{I}$ could be useful at this point.

About eq. (1) and (3), the maximum over all environments has been replaced by the sum. This can be seen as replacing a $\| \cdot \|_\infty$ with a $\| \cdot \|_1$, which has a very different behavior (tends to cancel components, while the infinite norm makes all components small). How can this change be justified and impact proofs? The same question could apply to $R'(\Phi)$ that is the sum instead of the max (but the $\epsilon$-relaxation makes this change more natural).

About eq. (3), how to be sure that invariant predictors exist?

Is the restriction to real-valued $\Phi$ necessary? I feel it makes the compositional architecture $m \circ \Phi$ a bit obsolete. It rather appears now as a trick to use the differential constraint (the $\nabla$ could be replaced by standard derivative since $w$ is one-dimensional). Also, is there a way to consider sub-differentiable, or non-differentiable loss functions here?

About Prop. 2, can authors elaborate a bit on the $\kappa$ constant, that seems to relate to $\epsilon$ in a complex way, and is likely to be very small?

About Prop. 4, as far as I understood, one of the main changes is that $\epsilon$ now replaces $\kappa$ (that I think is better to get an idea of dependences). Otherwise, the bound seems almost identical to that of Prop. 2, suggesting that the difference between EIRM and OOD solutions is "the same" as that between EIRM and IRM solutions?



MINOR COMMENTS
##############

p.3 Asm. 1, $\exists$ in the beginning of an english (i.e. non mathematic) sentence is a bit weird
P.3 we define *the/a* invariant map
p.4 Prop. 1, If $\ell$ is *the* square loss
p.4 for *the* cross-entropy loss
p.4 the parentheses two lines after seem unnecessary
p.4 the space*s* of representation and
p.4 eq. (4) *s.t.* omitted
p.5 the statement of Prop. 3 is a bit misleading, putting the reference in the title would be clearer (although authors clearly cite the reference)
p.6 Asm. 7, induct*i*ve bias and & symbol quite unusual



OVERALL EVALUATION
##################

This paper proposes a very interesting theoretical study to elucidate which one of ERM or IRM is better and when. Authors also present conclusive experiments matching the theoretical findings. For all these reasons I think this contribution deserves acceptance.


--- EDIT POST REBUTTAL ---

I thank the authors for their detailed answer(s) and the efforts they put in editing the revision (in particular I agree with other reviewers that Prop. 17 adds clarity). Overall my stance on the paper has not changed and I still recommend acceptance.

---

> ### Author Response · Authors · 2020-11-14
> **Response to Reviewer 1 (Part 1)**
>
> We are very thankful to Reviewer 1 for the time and effort. We have tried our best to incorporate your suggestions into the revised draft. We highlight all the changes in the revised paper in blue. Below we address the main concerns. We also look forward to hearing your feedback on the changes and our responses.
>
> 1.  **About Assumption 1:** Assumption 1 is based on the principle of invariance in causality. The intuition for Assumption 1 goes as follows. Consider the process of human labeling an image. The human extracts the relevant causal features from the image and uses them to come up with the label. If images are collected from different locations, then each location can be understood as one environment. Suppose there is an image of the exact same cow in the same pose on a green background in one location and a brown background in another location. The human labeler will ignore the background and extract the relevant features representing the cow to label it a cow.  Assumption 1 requires the existence of such a causal feature extractor, which we call $\Phi^{*}$ (In this example of labelling a cow $\Phi^{ *}\not=\mathsf{I}$.).   Therefore, Assumption 1 in other words is very similar to assuming that only causal parents (non spurious features) are used to generate the labels in the `"true" data generating process and this generating mechanism is same across environments. The assumption itself is hard to verify mathematically. However, this is a very common assumption and is the bedrock for many works in causality and in causally inspired machine learning.  For example, please see the latest survey https://arxiv.org/abs/1911.10500 where invariance is argued to be the signature of a causal relationship. In fact, it is shown that "invariance condition" is equivalent to the standard definition of a Causal Bayesian Networks - a central object of interest in Pearl's causal theories. Please see https://ftp.cs.ucla.edu/pub/stat_ser/r384.pdf
>
> 2. **About sum vs. max in equation 3/4:** The reviewer correctly points out that it may seem odd that equation (4) has a sum in it, while in equation (1), we have a max. The formulation in equation (4) was proposed by Arjovsky et al. and the authors chose sum as the criterion. The formulation in equation (4) is able to recover or get close to the solution $m\circ \Phi^{ *}$ to equation (1) in many cases as we show in Proposition 4, 5, and 6. What would happen if we replaced sum with max in equation (4)?  We can adapt Proposition 4, 5, and 6 and continue to show that this formulation is able to recover the solution to equation (1). Why does changing sum to max not change the insights from Proposition 4,5, and 6?  Consider the square loss function. If the linear general position assumption is satisfied, then there are only two feasible solutions to the constraints in equation (4) $m\circ \Phi^{ *}$ and zero map. Therefore, both sum and max criterion will prefer  $m\circ \Phi^{ *}$ over the zero map.
>
> 3.  **About real valued  $\Phi$:** In Arjovsky et al. the authors provide a detailed justification as to why the real-valued $\Phi$ is both good in theory and practice. However, we understand and appreciate the concern of the reviewer. Can we generalize the sample complexity findings to a more general setup say when $\Phi$ is vector-valued? For more general loss functions we can reformulate the constraints and require the norm of the gradient w.r.t the parameters of the classifier composed on top to be close to zero. The proof techniques that we developed, can be adapted to these more general cases, and insights similar to the current results can be obtained. We stuck with real-valued  $\Phi$ as that  is standard.
>
> 4.  **About feasibility:** $m \circ \Phi^{ *}$ is the ideal predictor as it solves equation (1). Due to Assumption 1, we can see that if  $m\in \mathcal{H}w$ and $\Phi^{ *}\in \mathcal{H}_{\Phi}$,  $m\circ \Phi^{ *}$ is an invariant predictor and thus a feasible solution to the constraints in equation (3). Suppose we are using square loss and we are restricted to linear classifiers $\mathcal{H}_w$. In such a case, there is another trivial invariant predictor. Let $\Phi$ maps all the points to zero.  Any linear classifier on top of the zero representation is optimal. Thus zero predictor is an invariant predictor and a feasible solution to constraints in equation (3). Since zero is not a good predictor, we do not want the equation (3) to output the zero predictor. We hope solving equation (3) outputs $m \circ \Phi^{ *}$.
>
> For the remaining responses refer to part 2 of the review.

---

> > ### Author Response · Authors · 2020-11-14
> > **Response to Reviewer 1 (Part 2)**
> >
> >  5. **Interpreting $\kappa$:** Recall $\kappa = \min_{\Phi \in \mathcal{H}\Phi} |R^{'}(\Phi)-\epsilon|$.  It measures how close any penalty can get to the boundary  $\epsilon$. $\kappa$ quantifies how good the finite sample approximation $\hat{R}^{'}$ need to be in order to get $\hat{\mathcal{S}}^{\mathsf{IV}}(\epsilon) = \mathcal{S}^{\mathsf{IV}}(\epsilon)$. As described above $m \circ \Phi^{*}$ is an invariant predictor and thus it is a feasible solution (satisfies the constraints) to equation (3). Suppose  $m\circ \Phi^{ *} \in \mathcal{H}_{\Phi}$. $m\circ \Phi^{ *}$ is also a feasible solution (satisfies the constraints) to equation (4) and equation (5). Therefore, $R^{'}(m\circ \Phi^{ *})=0$.   We order the representations based on the values of $R^{'}(\Phi)$ and select the representation with the smallest non-zero $R^{'}(\Phi)$ and call it $\tilde{\Phi}$. Let us say $\epsilon$ is set to a value less than $\frac{R^{'}(\tilde{\Phi})}{2}$. In this case, $\kappa=\epsilon$ and the assumption $\kappa>0$ changes to $\epsilon>0$, which translates to saying we allow the slack in equation (5) to be arbitrarily close zero but we just do not let it become exactly zero. To summarize, for sufficiently small $\epsilon$, $\kappa$ is equal to the slack $\epsilon$ and thus we only require $\epsilon$ to be greater than zero.
> >
> > 6. **Similarities and differences between bound in Proposition 4 and 2:** There are a couple of important differences we would like to clarify. The second term  in Proposition 4, $\frac{16L'^2}{\epsilon^2} \log(\frac{2}{\delta})$, as rightly pointed out depends on $\epsilon$ as oppposed to $\kappa$ in Proposition 2. There is one crucial difference. The second term in Proposition 4 does not depend on the size of $\mathcal{H}_{\Phi}$ unlike the corresponding term in Proposition 2. Thus in Proposition 4, only the first term depends on the size of the hypothesis class and not the second term.
> >
> >
> > **Reference**
> > Arjovsky, Martin, et al. "Invariant risk minimization." arXiv preprint arXiv:1907.02893 (2019).

---

### Official Review · AnonReviewer2 · 2020-11-08
**Sample complexities for empirical invariant risk minimization that may show an edge over ERM in certain regimes of distributional change. Overall good paper.**

**Rating:** 7
**Confidence:** 4

**Review:**

### Summary

This paper considers a learning scenario where training data $(X,Y)$ comes from a mixture, where membership in each mixture component (“environment”) is clearly labeled. Generalization is required not just under the same mixture, but potentially under changing mixing distributions. This is captured via several alternative formulations. The key underlying assumption that makes the extrapolation possible is the existence of a a representation $\Phi(X)$ that leads to an environment-invariant statistics (by focusing on quadratic loss and linear predictors, the paper asks for invariant $E[Y|\Phi(X)]$ and constant $var[Y|\Phi(X)]$). A range of other assumptions are made to enable analysis, such as sufficiency of the representation in at least one environment and boundedness of the loss and its gradient.

The main object of study is empirical invariant risk minimization (EIRM), which minimizes the empirical risk under the further constraint that the predictor is (approximately) optimal in each environment. The main result is that ERM and EIRM have the same sample complexity when the environments are related via a covariate shift, but that under other situations (namely confounding variables and anti-causal relationships), EIRM continues to enjoy guarantees, while informal arguments are offered as to why ERM doesn’t. Experiments are given to illustrate this disparity, and thus potentially justify the use of EIRM.

### Strengths
+ The paper provides a very methodical and detailed reasoning to try and identify the benefits of invariant risk minimization and provides several compelling arguments.
+ The main strength of the paper is in placing the finger on the situation where ERM and IRM appear to differ. It doesn’t complete the story here, but at least identifies the reasons behind prior accounts that seemed to conflict on the surface.
+ The math is laid out very clearly and the gradual introduction of detail from the main text to the supplements keeps the presentation of the ideas manageable.
+ The new colored MNIST datasets proposed in this paper is a very useful contribution, which more realistically captures the kinds of shifts and dependences that may arise in realistic ML applications.

### Weaknesses
- In the quest for detail, some of the high-level perspective is lost. The first place where this happens is when the paper claims that Assumption 1 is equivalent to covariate shift when $\Phi$ is the identity map. But the constant-variance requirement of A1 does not follow from covariate shift, so in fact A1 is a stronger assumption in that case.  And this is not mild: it is the glue that binds together the proof of Proposition 1. (Incidentally, the argument for the latter can be given more simply as: in the sufficient environment, the optimal estimator enjoys the constant variance as MSE; if the objective is higher then, in all other environments, we can reduce it by reverting to the conditional expectation; thus the latter must be the optimal predictor.)
- The second place where the high-level perspective is lost is when tying together the ideal OOD predictor (1) and the IRM formulation (3). There are two facts that distinguish (1) and (3). First, (1) does not require the predictor to be optimal in each environment unlike (3). In fact, in the manner of most minimax results, the solution might sacrifice even in a “good” environment by being worse than optimal, so that some of the “bad” environments fare better. Second, (3)’s average objective may result in a choice of $\Phi$ far from the ideal representation. To see this, imagine two environments that need a more elaborate representation and a third that doesn’t but is very noisy. (1) would favor the latter and give a simple representation that would perform well in the worst case, while (3) may ignore the third environment and focus on the first two and thus optimize the average case.
- This blurred high-level perspective makes it so that the rest of the paper analyzes the procedure in (3) (or rather its gradient-constrained formulation) while striving to recover the solution of (1) (as given by Proposition 2). It’s a conceptual disconnected which unfortunately only becomes evident upon reading through, and is not adequately highlighted by the paper. It’s also possible that the bulk of additional assumptions introduced, as well as the loss and model structures adopted, are really a result of enforcing the reconciliation between these two different paths.
- The entire methodology of EIRM hinges on a perfect awareness of the environment (the constraint cannot be calculated without it). In that sense, it is quite an unfair comparison to plain ERM. In fact, the awareness can be thought of as adding a particular inductive bias to ERM, so it is not much of a surprise that the analysis presented is not more novel than regularized ERM.
- ~~What the paper claims a lot but does *not* deliver is a lower bound for ERM under the cases where new upper bounds for EIRM are given. In Table 1 there is a “No” under OOD, on page 7 we read “[t]his shows that EIRM works in more settings than ERM”, and in the conclusion we read “we proved that […] ERM can be asymptotically biased”. But apart from informal arguments and experiments, this is not in fact shown.~~ In particular, considering the great number of assumptions leading to Proposition 5, it is not at all clear how ERM behaves under the very same assumptions (note that the experiments do not satisfy all these assumptions, e.g. $\pi^\min=0$, so we cannot treat them as counterexamples).

### Comments
* The $1/\kappa^2$ dependence on $\kappa$ is a bit weird. $\kappa$ is the smallest $|R’(\Phi)-\epsilon|$ over all $\Phi$. So in principle there maybe a $\Phi$ that makes $R’(\Phi)=\epsilon$ and Proposition 2 vacuous. I think the way it is used is the same as the reason behind why $\kappa<\epsilon$: it’s because we assume a feasible solution of the optimization exists, so $0 < R’(\hat \Phi) < \epsilon$, thus $|R’(\hat\Phi)-\epsilon|<\epsilon$. But $\kappa$ is *not* defined as $|R’(\hat\Phi)-\epsilon|$, perhaps it should be?
* After Assumption 5, the paper should make it clear that the initial focus is on $S$ being linear.
* Some things should be moved up from the supplements for completeness. For example, the definition of your constraint, the $\theta$ for Figure 1d.
* Typos: Assumption 7, *Inductive* bias. $[\psi^1,\psi^e,\psi^3]$: change $e$ to $2$. Figure 1, CMNIST for all.

### Overall
The paper considers interest settings of distributional change and corresponding learning formulations. It makes a lot of assumptions to obtain sample complexities that justify the use of empirical invariant risk minimization, and falls a bit short by not giving a formal converse for the inadequacy of plan empirical risk minimization, despite making the claim. Nevertheless, the contributions are insightful, and the paper may be worth sharing with the community.

_[Edit: I'm glad the author clarified some of the tensions between the different definitions. The ERM bias proof seems also okay, but it's still not clear how ERM performs with all the additional assumptions of the IRM result. I'm still not on the same page regarding $\kappa$, but I'm willing to take a leap of faith on the bits that I'm uneasy about. I would suggest this paper is shared with the community, since I believe it has enough good insights.]_

---

> ### Author Response · Authors · 2020-11-14
> **Response to Reviewer 2 (Part 1)**
>
> We are very thankful to Reviewer 2 for the time and effort. We have tried our best to incorporate your suggestions into the revised draft. We highlight all the changes in the revised paper in blue. Below we address the main concerns. We also look forward to hearing your feedback on the changes and our responses.
> 1. **Assumption 1 and connection to covariate shift:** We only meant to connect the covariate shift assumption with the first part of Assumption 1, which requires the existence of an invariant expectation, and not with the second part, which requires a constant variance. Thanks for pointing this out. We have made this correction.
> 2. **Connection between equation 1 and equation 3:** Below we describe the connection between equation (3) and equation (1) and why equation (3) is really needed to solve equation (1). We have utilized some of the extra space from one page to highlight these points better in the paper (See pages 3 and 4).
>
>     -  **Equation (1) is challenging to solve directly and equation (3) is needed:** Equation (1)  is the ideal problem we want to solve. However, it is extremely challenging for the following reasons. In equation (1), we solve a min-max optimization where the max operation occurs over all the environments -- these include the seen train environments and the unseen test environments. Since we don't have access to the test environments, equation (1) is very challenging to solve. The formulation in equation (3) was proposed in Arjovsky et al. to solve equation (1). But before describing equation (3) one may wonder why not solve a simple min-max optimization over the training environments itself. A standard min-max optimization on training environments will fail to solve equation (1) for the model discussed in Section 3.2.2 (See Arjovsky et al. for a theoretical justification) but the formulation in equation (3) succeeds. Our Propositions 4-6 show several different settings when solving equation (3) solves equation (1) under assumptions that are very similar to the ones in Arjovsky et al. We do acknowledge that if we make minimal or no modeling assumptions such as in Proposition 2, it is not possible to guarantee that equation (3) always solves equation (1).
>
>      - **Difference between equation (3) and equation (1):** The reviewer pointed out that in equation (3), we find a classifier that is optimal in each environment, while this is not the case for equation (1). We want to clarify an important point here. In equation (3), we first transform the data using a representation (also a part of the optimization variable) and then find a classifier that is simultaneously optimal across all the environments. On the other hand, we do not transform the data using such representations before finding the optimal classifier in equation (1). This process of transforming data and finding one classifier that works well across all the environments forces the representation to be conservative and rely on fewer features. In the example that the reviewer gives with the three environments, it is not very clear to us why equation (3) would fail. If a more elaborate representation is selected, then the classifier on top of it would not be optimal for the third environment, which would disqualify it as a possible invariant predictor.
>
>     - **What is the reason for first transforming and then searching an optimal classifier in equation (3)?**
> Recall that in Assumption 1, we defined a representation $\Phi^*$, which was subsequently used to define a predictor $m\circ \Phi^*$ that was shown to solve equation (1). Note that after transforming the data by a representation $\Phi^*$, the best classifier in each environment is the same model $m$. This very property that selecting a representation $\Phi^*$ leads to the same optimal predictor in each environment is the motivation for what is called an invariant predictor in equation (3). Equation (3) searches over invariant predictors. Since $m\circ\Phi^*$ is an invariant predictor itself, equation (3) hopes to find it and solve equation (1).
>
>  3. **Addressing issue of getting lost in details:**
>     - **A new illustrative example, figure and improved presentation:** We have added an illustrative example with a figure to describe the graphical model to describe the difference between Propositions 4 and 5. We also use this figure early on (Figure 1 in the revision) in the introduction to give a reader a simple example of the different types of distribution shifts. We have improved the presentation of the implications of Proposition 5, which we hope will help the reader.
>
>      -**A high-level dialogue:** At the beginning of the supplement, we have a high-level dialogue between two students (inspired from original work IRM) around existing works on IRM and how our work fits in the big picture.
>
> For the remaining responses refer to part 2 of the response below.

---

> > ### Author Response · Authors · 2020-11-14
> > **Response to Reviewer 2 (Part 2)**
> >
> > 4. **Our analysis vs.  regularized ERM:** Below we explain the key challenges that differentiate our setup from a standard regularized ERM.
> >
> >       **Key challenges and technical novelties:**
> >      - **IRM penalty is not separable** The IRM penalty $R^{'}$ is a weighted sum of squares of norm of the gradient of the expected loss. Now since the population value (the gradient of the expected loss) is composed with a squared function, the penalty cannot be expressed as a separable sum over the data points. We need separability to be able to use standard concentration inequalities. We propose an alternate way of expressing the penalty that allows us to build a separable estimator.
> >     - **Standard learning theory bounds deal with concentration around risk and not a target model itself.** It is common in learning theory to analyze the concentration of empirical risk around the expected risk. This does not guarantee that the two approaches -- empirical and expected -- actually find the same risk minimizer. In contrast in our case, we have a target model, $m\circ \Phi^{*}$ , that solves equation (1) and we want to be able to guarantee that empirical solutions concentrate around that. This being a new task in itself required us to do a new analysis (see the proof of Proposition 5).  This being a challenging task, we decided to pursue linear and polynomial generative models in this work.  We  exploit the linear general position assumption (which is a condition derived in Arjovsky et al 2019 especially for the IRM problem) along with new estimator we proposed for IRM penalty to connect concentration around risks with concentration around target model.
> > 5. **Proving biasedness of ERM:** In the supplement of the submitted version, we had a  discussion on why ERM is biased. In the revision, we provide a formal Proposition (Proposition 17) in the supplement and proof of biasedness of ERM for the same setup as in Proposition 5. We can move this new proposition from the supplement to the main part of the paper if the reviewer advises so.
> > 6. **About environment information:**  The reviewer makes an interesting point. ERM does not use the information about the environment index while the IRM method does. Even if we were to augment the environment index to the features in ERM, our arguments in the proof of Proposition 17 in the supplement still apply to show that ERM will continue to be biased.
> > 7. **Interpreting $\kappa>0$:** Here we provide a simple justification of why $\kappa>0$ is reasonable.  Recall $\kappa = \min_{\Phi \in \mathcal{H}_{\Phi}} |R^{'}(\Phi)-\epsilon|$.  It measures how close any penalty can get to the boundary  $\epsilon$. $\kappa$ quantifies how good the finite sample approximation $\hat{R}^{'}$ need to be in order to get $\hat{\mathcal{S}}^{\mathsf{IV}}(\epsilon) = \mathcal{S}^{\mathsf{IV}}(\epsilon)$. As described above $m \circ \Phi^{*}$ is an invariant predictor and thus it is a feasible solution (satisfies the constraints) to equation (3). Suppose  $m\circ \Phi^{ *} \in \mathcal{H}\Phi$. $m\circ \Phi^{ *}$ is also a feasible solution (satisfies the constraints) to equation (4) and equation (5). Therefore, $R^{'}(m\circ \Phi^{ *})=0$.   We order the representations based on the values of $R^{'}(\Phi)$ and select the representation with the smallest non-zero $R^{'}(\Phi)$ and call it $\tilde{\Phi}$. Let us say $\epsilon$ is set to a value less than $\frac{R^{'}(\tilde{\Phi})}{2}$. In this case, $\kappa=\epsilon$ and the assumption $\kappa>0$ changes to $\epsilon>0$, which translates to saying we allow the slack in equation (5) to be arbitrarily close zero but we just do not let it become exactly zero. To summarize, for sufficiently small $\epsilon$, $\kappa$ is equal to the slack $\epsilon$ and thus we only require $\epsilon$ to be greater than zero.
> >
> > 8. **Proof sketches added:** The reviewer suggested to move somethings from the supplement. We have added key proof sketches in the main paper now.
> >
> > **Reference**
> > Arjovsky, Martin, et al. "Invariant risk minimization." arXiv preprint arXiv:1907.02893 (2019).

---

### Decision · Program_Chairs · 2021-01-07
**Final Decision**

**Decision:**

Accept (Poster)

**Comment:**

The paper considers learning settings with distributional change. It makes a lot of assumptions to obtain sample complexities that justify the use of empirical invariant risk minimization, and falls a bit short by not giving a formal converse for the inadequacy of plan empirical risk minimization, despite making the claim. Nevertheless, the contributions are insightful, and the paper may be worth sharing with the community.
The grading were overall positive from the reviewers, though particularly critical, and I doubt the whole paper could be fully double-checked: one could question the ability of the reviewers to perform a deep analysis on a 48-pages theoretical paper in the time constraints imposed by a conference model...